# The CTLH ubiquitin ligase substrates ZMYND19 and MKLN1 negatively regulate mTORC1 at the lysosomal membrane

Yin Wang [1,2], Yifei Liao[1,2], Yizhe Sun [1,2], Bidisha Mitra[1,2], Rui Guo [1,2,5], Brenda Iturbide Piedras[1,2], Shaowen White[1,2], Hsin-Yao Tang [3], John M. Asara [4], Italo Tempera [3], Paul M. Lieberman [3] & Benjamin E. Gewurz [1,2] ✉

Most Epstein–Barr virus-associated gastric carcinoma (EBVaGC) harbor non-silent mutations that activate phosphoinositide 3 kinase (PI3K) to drive downstream metabolic signaling. To gain insights into PI3K/mTOR pathway dysregulation in this context, we perform a human genome-wide CRISPR/Cas9 screen for hits that synergistically blocked EBVaGC proliferation together with the PI3K antagonist alpelisib. Multiple subunits of carboxy terminal to LisH (CTLH) E3 ligase, including the catalytic MAEA subunit, are among top screen hits. CTLH negatively regulates gluconeogenesis in yeast, but not in higher organisms. The CTLH substrates MKLN1 and ZMYND19, which highly accumulated upon MAEA knockout, associate with one another and with lysosome outer membranes to inhibit mTORC1. Rather than perturbing mTORC1 lysosomal recruitment, ZMYND19 and MKLN1 block the interaction between mTORC1 and Rheb and also with mTORC1 substrates S6 and 4E-BP1. Thus, CTLH enables cells to rapidly tune mTORC1 activity at the lysosomal membrane via the ubiquitin/proteasome pathway.

Gastric carcinoma is the fourth leading cause of cancer associated death worldwide[1]. The Cancer Genome Atlas defined four distinct gastric carcinoma subtypes, of which Epstein–Barr virus-associated gastric carcinoma (EBVaGC) represents ~9% of cases. Several features distinguish EBVaGC from the other subtypes, including particularly high frequency of non-silent mutations in *PIK3CA*, which encodes the phosphoinositide 3-kinase (PI3K) p110α catalytic subunit. As compared with 3–42% mutation frequency observed in other gastric carcinoma subtypes, *PIK3CA* mutations are present in >80% of EBVaGC, suggesting important oncogenic driver roles[2,3]. PI3K negative regulator PTEN mutations are observed in other cases, raising the question of whether PI3K activation is a universal EBVaGC feature. Although curable if caught early, metastatic EBVaGC remains largely untreatable.

PI3K transduces signals from plasma membrane receptors to activate downstream anabolic metabolism pathways[4]. PI3K phosphorylates phosphatidylinositol 4,5-bisphosphate to generate the second messenger PIP3. Aberrant PI3K signaling is a cancer hallmark[5]. Major PI3K pathway targets include the kinases Akt and the mechanistic target of rapamycin complex 1 (mTORC1), comprised of mTOR, regulatory-associated protein of mTOR (Raptor), mammalian lethal with SEC13 protein 8 (mLST8), 40 kDa Proline-rich Akt substrate (PRAS40), and DEP domain-containing mTOR-interacting protein (DEPTOR) subunits. mTORC1 is recruited to lysosomal outer membrane activation sites by Ras-related GTPase (Rag) heterodimers in response to nutrient cues, including plentiful amino acid supply[6,7]. Since Rag heterodimers are comprised of a RagA or RagB subunit paired with a Rag C or D subunit, there are four Rag complexes. mTORC1 is then activated by the small

[1]Division of Infectious Diseases, Department of Medicine, Brigham and Women's Hospital, Boston, MA, USA. [2]Broad Institute, Cambridge, MA, USA. [3]The Wistar Institute, Philadelphia, PA, USA. [4]Division of Signal Transduction, Beth Israel Deaconess Medical Center and Department of Medicine, Harvard Medical School, Boston, MA, USA. [5]Present address: Department of Molecular Biology and Microbiology, Tufts University, Boston, USA. ✉e-mail: bgewurz@bwh.harvard.edu

GTPase Ras homolog enriched in brain (Rheb), which is tethered to lysosomal outer membrane sites[8–11].

The PI3K-AKT-mTOR pathway plays a critical role in cell proliferation, survival, and metabolism[12,13]. Activated mTORC1 promotes anabolic metabolic pathways, including protein synthesis via phosphorylation of translation initiation factor 4E binding protein 1 (4E-BP1) and S6 kinase (S6K), nucleotide synthesis via upregulation of the transcription factor ATF4, and regulate lysosomal biogenesis via phosphorylating TFEB and TFE3[14]. 4E-BP1 phosphorylation upregulates cap-dependent translation, whereas S6K phosphorylation promotes protein synthesis and anabolic metabolism[15,16]. Consequently, hyperactive PI3K activity is a major cancer therapeutic target[17]. However, drug resistance and dose-related toxicities limit the use of PI3K inhibitors[18], which has stimulated enthusiasm for synergistic therapeutic approaches[19,20]. For example, the Food and Drug Administration approved combination PIK3 p110α inhibitor alpelisib and estrogen receptor antagonist fulvestrant use for metastatic breast cancer[21].

To gain insights into factors that support EBVaGC PI3K/mTORC1 signaling, we performed a human genome-wide CRISPR/Cas9 screen for targets whose inhibition was synthetic lethal with alpelisib. Multiple subunits of the C-terminal to LisH (CTLH) E3 ligase were amongst the top screen hits. While CTLH negatively regulates gluconeogenesis in yeast[22–24], CTLH has pleotropic metabolism roles in higher organisms[25–32] and potential CTLH roles in gastric carcinoma are unstudied. Our multi-omic analyses highlighted that CTLH supports mTORC1 activity by controlling levels of substrates ZMYND19 and MKLN1. Upon perturbation of CTLH activity, ZMYND19/MKLN1 associate at lysosome membrane sites, where they bind to Raptor and RagA/C and interfere with mTORC1 activation at the level of its association with Rheb.

## Results

### CRISPR/Cas9 screen for alpelisib synthetic lethal targets

We characterized PI3K signaling in YCCEL1, one of only three available EBVaGC cell lines, which expresses the EBV-encoded proteins EBNA1 and LMP2A[33] and harbors a PIK3CA kinase domain histidine 1047 arginine gain-of-function mutation (Supplementary Fig. 1a). Alpelisib diminished PI3K target AKT threonine 308 and serine 473 phosphorylation in a dose-dependent manner, with most AKT phosphorylation lost at the 0.5 μM dose in YCCEL1 (Fig. 1a). At this dose, alpelisib restrained YCCEL1 proliferation by approximately 50% (Fig. 1b). Alpelisib treatment and CRISPR *PIK3CA* knockout (KO) produced concordant transcriptome-wide changes in YCCEL1, suggestive of on-target activity (Fig. 1c, Supplementary Fig. 1b, c, and Supplementary Data 1).

To identify EBVaGC targets that synergistically block EBVaGC proliferation together with 0.5 μM alpelisib, we performed a human genome-wide CRISPR-Cas9 screen. Cas9 + YCCEL1 were transduced with the Brunello lentiviral single guide RNA (sgRNA) library at multiplicity of infection of 0.3. Seven days post-transduction and following puromycin selection, the KO library was cultured either with 0.5 μM alpelisib or with DMSO vehicle for an additional two weeks. Using biological triplicate replicates, PCR-amplified sgRNA abundances in the surviving cell pools were quantitated by next-generation DNA sequencing. sgRNA abundances between alpelisib versus DMSO-treated cells were cross-compared, and hits were identified by the STARS algorithm (Fig. 1d and Supplementary Data 2)[34].

At a multiple hypothesis adjusted $q < 0.05$ cutoff, the screen identified 30 hits, in which independent sgRNAs targeting these human genes were depleted in the surviving alpelisib-treated cell pool relative to their levels in DMSO-treated cells (Fig. 1e). Top hits included genes encoding factors known to be highly related to PI3K biology. These included (1) the ubiquitin E3 ligase KBTBD2, which regulates PI3K p85α regulatory subunit abundance[35,36]; (2) GPX4, which uses glutathione to detoxify lipid free radicals and protect against ferroptosis[37]; and (3) the ubiquitin specific protease USP7, which removes a monoubiquitin group from PTEN to support its cytoplasmic subcellular localization[38] and which protects gastric cancer cells from ferroptosis[39] (Fig. 1e, f). These data are consistent with the observation that PI3K signaling protects cells against ferroptosis induction[34].

Multiple positive regulators of the RAF/MAP kinase pathway also scored strongly, including KRAS, which recruits RAF to plasma membrane activation sites, as well as the catalytic and regulatory components of the SHOC2 phosphatase complex, encoded by *SHOC2* and *PPP1CB* (Fig. 1e–g). SHOC2 dephosphorylates plasma membrane localized RAF phosphoserine 365 to drive RAF/MAP kinase pathway activation[40]. While PI3K/mTOR and RAS/MAPK pathways crosstalk[41,42], this has yet to be characterized in the gastric carcinoma setting. However, KRAS is targeted by deleterious mutations in multiple gastric carcinoma subtypes, including EBVaGC[2]. We validated that GPX4, USP7, or KRAS KO significantly reduced numbers of live YCCEL1 treated with alpelisib, relative to levels in cells treated with vehicle. We also validated that the ferroptosis antagonist Fer-1 rescued survival of cells treated with both alpelisib and with the GPX4 antagonist ML-210 (Supplementary Fig. 1d–f).

Multiple subunits of the E3 ubiquitin ligase C-terminal to LisH (CTLH) complex scored as top hits, including genes encoding the catalytic unit MAEA subunit, YPEL5, and WDR26 (Fig. 1e–g). The CTLH E2 ubiquitin ligase encoded by *UBE2H*[26,43] nearly also scored. CTLH is the mammalian homologue of the *Saccharomyces cerevisiae* glucose-induced degradation deficient (GID) complex, which ubiquitinates gluconeogenic enzymes when extracellular glucose is abundant[22,43–45]. This metabolic role does not appear to have been conserved in higher organism CTLH complexes[25–29].

The CRISPR screen also identified sgRNA targets whose knockout ameliorated effects of low dose alpelisib on YCCEL1 proliferation, as judged by significantly enriched sgRNA abundance in alpelisib versus DMSO-treated cells (Fig. 1h–j and Supplementary Data 2). Top hits included genes encoding the PI3K negative regulator PTEN and the PI3K p85α regulatory subunit (PIK3R1), as expected, p85α restrains PI3K catalytic activity and is a tumor suppressor[46,47]. Multiple mTOR negative regulators were also amongst top hits in this category, including TSC1 and components of the GATOR1 complex encoded by NPRL2 and NPRL3. Together with TSC2 and TBC1D7, TSC1 negatively regulates mTORC1 activation through GTPase-activating protein (GAP) activity towards the small GTPase Rheb[48], whereas GATOR1 GAP activity negatively regulates mTORC1 lysosomal recruitment by the Ragulator-Rag complex in response to amino acid depletion[8,49–51]. Multiple Cullin-3 (Cul3) E3 ligase-related genes also scored (Fig. 1h–j). The AKT pathway negative regulator KCTD5, which is a Cul3 substrate adaptor, scored strongly[52]. Similarly, the Cul3 substrate adaptor KEAP1 and the kelch-family member ENC1 scored, a target of which is the transcription factor Nrf2, which drives antioxidant responses[53].

### CTLH inhibition causes metabolic remodeling

CTLH is an unusually complex ubiquitin E3 ligase, comprised of at least nine components (Fig. 2a), but has not previously been linked to PI3K signaling or studied mechanistically in the gastric cancer context. We therefore investigated phenotypes of CRISPR KO of the RING domain-containing catalytic subunit and screen hit MAEA. MAEA KO and alpelisib together decreased YCCEL1 live cell numbers more than either alone (Fig. 2b–d). Similar results were obtained in EBV-infected SNU-719 gastric carcinoma cells, EBV-uninfected PIK3CA mutant HGC-27 gastric carcinoma, and in PIK3CA wildtype SNU-1 gastric carcinoma cells (Supplementary Fig. S2a–d). Notably, SNU-1 harbor an activating KRAS G12D mutation that activates downstream PI3K and MAPK pathways[54]. However, alpelisib and MAEA KO exerted milder effects on EBV-uninfected SNU-16 gastric carcinoma cells, perhaps due to lower levels of PI3K activity (Supplementary Fig. S2e). MAEA KO also did not exhibit synthetic lethal effects with alpelisib in HEK-293T

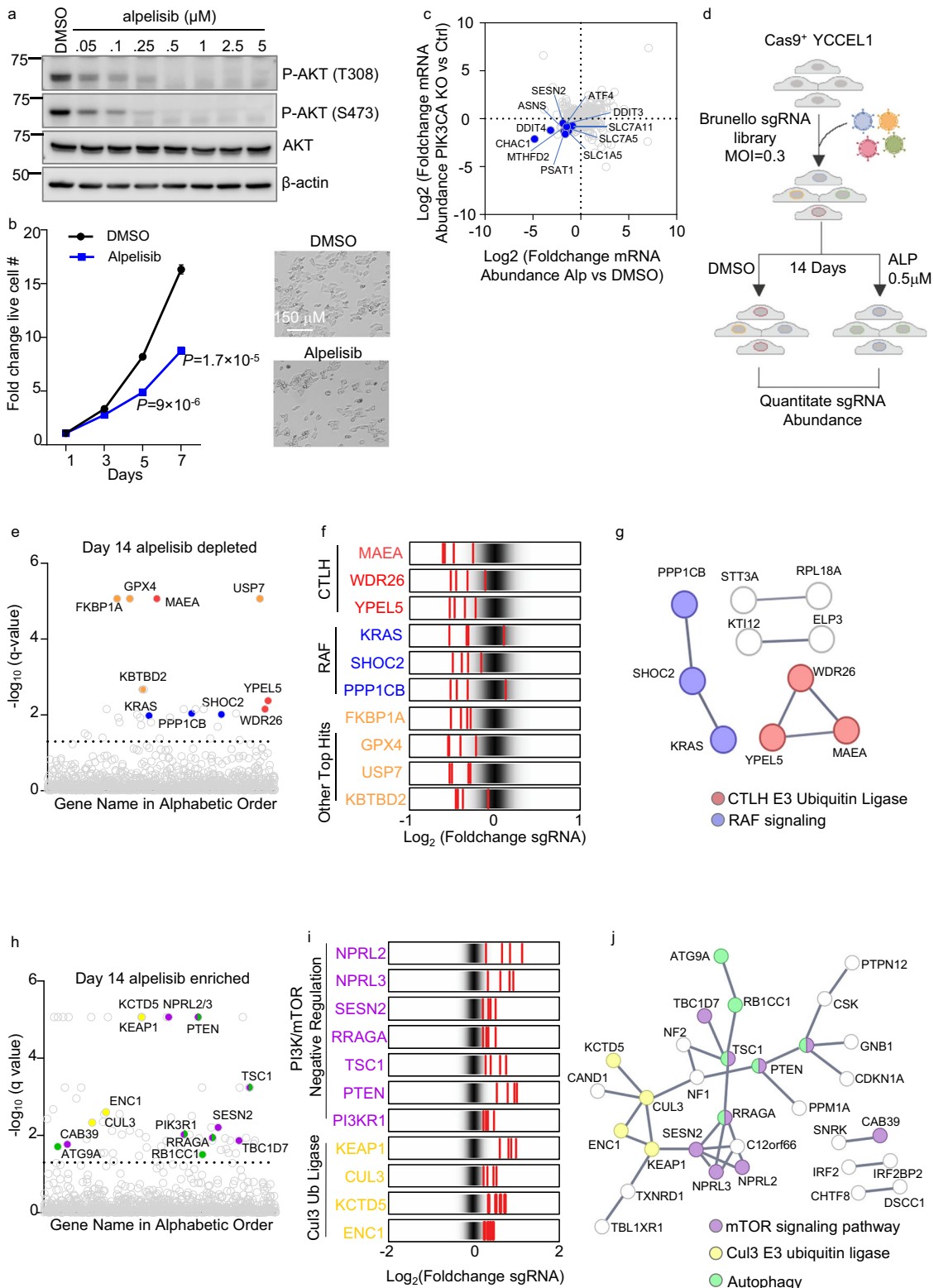

(Supplementary Fig. S2f). MAEA depletion did not alter EBNA1 expression or de-repress EBV lytic gene expression, as judged by immunoblot for EBV immediate early BZLF1 or early BMRF1 lytic cycle antigens in YCCEL1 or SNU-719 (Supplementary Fig. S2g, h). Furthermore, BZLF1 KO did not rescue viability of alpelisib-treated MAEA KO YCCEL1 (Supplementary Fig. S2i), suggesting that effects on EBV gene expression did not account for the observed synthetic lethality.

To identify how MAEA KO altered gastric carcinoma growth versus survival, alone or together with alpelisib, we performed cell cycle analysis on propidium iodide-stained cells. While MAEA KO or alpelisib alone diminished S-phase cell number, together they caused widespread cell death, as judged by the number of sub-G0 cells (Fig. 2e and Supplementary Data 3). CRISPR MAEA effects were on-target, since MAEA cDNA with a silent point mutation to abrogate Cas9 cutting

**Fig. 1 | EBVaGC human genome-wide CRISPR-Cas9 screen for knockouts synthetic lethal with PI3K antagonist alpelisib. a** Immunoblot analysis of alpelisib inhibition of PI3K substrate AKT phosphorylation, using whole cell lysates (WCL) from YCCEL1 treated with 0–5 µM alpelisib for 3 h, representative of $n = 3$ biologically independent replicates. **b** Mean ± standard deviation (SD) fold change of live YCCEL1 cells treated with 0.5 µM alpelisib vs DMSO from $n = 3$ of biologically independent replicates. *P*-values were calculated by two-tailed Student's *t*-test. **c** Volcano plot visualization of $Log_2$ (fold change) mRNA abundances from $n = 3$ RNA-seq replicates of YCCEL1 CRISPR *PIK3CA* knockout versus control cells (*y*-axis) and YCCEL1 treated with 5 µM alpelisib versus DMSO for 6 h (*x*-axis). Individual transcripts are shown as circles. PI3K/mTOR pathway target transcription factor ATF4 are highlighted in blue. **d** Schematic of YCCEL1 genome-wide CRISPR-Cas9 screen. Cas9 + YCCEL1 transduced with the Brunello sgRNA library were selected with puromycin and cultured in DMSO or 0.5 µM alpelisib for 14 days. PCR-amplified sgRNA was quantitated to identify differential hits. The screen was performed in biological triplicate. Created in BioRender. Guo, R. (2025) https://BioRender.com/bvqo7ck. **e** CRISPR screen Manhattan plot demonstrating -$Log_{10}$

adjusted *q*-values (STARs algorithm). Genes are arranged alphabetically along the *x*-axis. Lower values signify sgRNA depletion in Day 14 alpelisib versus DMSO-treated cells. CTLH genes are red, RAF signaling genes blue, and other top hits orange. **f** Rug plots showing the Log2 transformed Foldchange abundances in Day 14 alpelisib versus DMSO treated cells of the four sgRNAs targeting the indicated screen hit gene (shown in red), relative to the overall distribution of Brunello library sgRNAs. CTLH subunits, RAF-related, and other top screen hits are highlighted. **g** STRING network analysis showing selected high confidence score (> 0.75) connections between selected screen hits. **h** CRISPR screen Manhattan plot as shown in (**e**). Higher values signify sgRNA enrichment in Day 14 alpelisib versus DMSO treated cells. PI3K/mTOR negative regulator hits are purple, Cul3 ubiquitin ligase related genes yellow and autophagy related genes green. **i** Rug plots as shown in (**f**). PI3K/mTOR negative regulators and Cul3-related screen hits are highlighted. **j** STRING network analysis showing selected high confidence score (>0.75) connections between selected screen hits. Source data are provided as a Source Data file for (**a**) and (**b**).

rescued survival of alpelisib-treated YCCEL1 upon KO of endogenous MAEA (Fig. 2f).

Multiple CTLH metabolic roles have been described[28–32,55–57], including in negative regulation of glycolysis via non-degradative ubiquitination of both pyruvate kinase M2 (PKM2) and the L-lactate dehydrogenase A chain (LDHA) and in negative regulation of central carbon metabolism[28,57]. We therefore hypothesized that CTLH perturbation altered gastric carcinoma metabolism pathways, and that this underlay its strong screen phenotype. To gain insights, we performed targeted liquid chromatography-tandem mass spectrometry (LC-MS/MS) profiling[58] of control vs MAEA-depleted YCCEL1 cells early after CRISPR editing, in the absence or presence of 0.5 µM alpelisib for 30 h. MAEA editing most significantly perturbed glycolysis, pentose phosphate, riboflavin, pyrimidine, and purine metabolism pathways (Fig. 2g and Supplementary Data 4). Glycolysis and pyruvate metabolism where the most significantly altered metabolism pathways in alpelisib-treated MAEA KO vs control cells (Fig. 2h and Supplementary Data 4). Nucleotide metabolism and glycolysis were also the most significantly perturbed by alpelisib in both MAEA KO or control YCCEL1 cells (Supplementary Fig. 3a, b and Supplementary Data 4), further highlighting overlap in effects of MAEA perturbation and PI3K blockade. Notably, a WDR26-containing CTLH ligase complex was recently found to regulate turnover of the enzyme nicotinamide/nicotinic-acid-mononucleotide-adenylyltransferase 1[32].

Multiple glycolysis pathway metabolites were highly depleted by MAEA KO + alpelisib treatment, including dihydroxy-acetone-phosphate, D-glyceraldehyde-3-phosphate, fructose-1,6-biphosphate, pyruvate, and lactate (Fig. 2i and Supplementary Data 4). Similarly strong effects were observed on purine and pyrimidine pathway metabolites (Supplementary Fig. 3c). To further analyze effects of MAEA KO on glycolysis, alone or together with alpelisib, we performed FACS analysis of fluorescent glucose analog 2-NBDG uptake. MAEA KO diminished 2-NBDG levels, alone or in combination with alpelisib, suggesting that a MAEA-containing CTLH ligase supports glucose uptake, which could be either by direct or indirect mechanisms (Fig. 2j and Supplementary Data 5). Furthermore, Seahorse XF flux analysis demonstrated that MAEA editing reduced extracellular acidification (suggestive of impaired glycolysis), basal and maximum respiration and ATP production, either alone or additively with alpelisib (Fig. 2k, l and Supplementary Fig. 3d–g). Taken together, these data suggest that a MAEA-containing CTLH complex supports central carbon and anabolic metabolism pathways in gastric carcinoma cells.

## ZMYND19 and MKLN1 form a complex that negatively regulates mTOR

To gain further insights, we next performed RNAseq on control vs MAEA KO YCCEL1. mTORC1 and xenobiotic metabolism pathways

were the most significantly altered by MAEA depletion (Fig. 3a and Supplementary Data 6). Multiple ATF4-regulated genes were highly downmodulated in alpelisib-treated MAEA KO cells, even in comparison with alpelisib-treated control cell levels (Supplementary Fig. 4a and Supplementary Data 6). mTORC1 can induce purine synthesis through one-carbon metabolism via upregulation of methylenetetrahydrofolate dehydrogenase 2 (MTHFD2) in an ATF4-dependent manner[59,60]. Since ATF4 and MTHFD2 were highly downmodulated in alpelisib-treated MAEA KO cells (Supplementary Fig. 4a and Supplementary Data 6), we hypothesized that they may jointly block mTOR. Consistent with this, MAEA KO diminished translation rate, as judged by puromycin chase immunoblot analysis (Supplementary Fig. 4b). By contrast, MAEA KO, with or without alpelisib, did not increase EIF2α serine 51 phosphorylation, suggesting that an integrated stress response pathway was not responsible for effects on translation (Supplementary Fig. 4c).

We therefore next analyzed MAEA KO effects on mTOR activity. MAEA KO reduced phosphorylation of mTOR targets S6K kinase and 4E-BP1 in both YCCEL1 and SNU-719 EBV+ gastric carcinoma cells and in HEK-293T (Fig. 3b, c and Supplementary Fig. 4d, e). Further suggestive of a CTLH regulatory role in support of mTORC1, we observed additive effects of MAEA depletion and alpelisib on inhibition of S6K T389 and 4E-BP1 S65 phosphorylation. Although the GID complex regulates AMP kinase in *C. elegans*[29], we did not appreciate a significant change in AMPK T172 phosphorylation in MAEA-depleted cells (Fig. 3c and Supplementary Fig. 4f). Also suggestive of inhibitory effects at the level of mTORC1, MAEA KO and low dose rapamycin additively reduced S6K and 4E-BP1 phosphorylation levels (Supplementary Fig. 4g). Low dose rapamycin and MAEA KO also synergistically reduced live cell numbers (Supplementary Fig. 4h), consistent with the hypothesis that MAEA KO inhibitory effects on mTOR underlay the observed synthetic lethality with alpelisib in YCCEL1. MAEA KO effects on mTOR inhibition were not dependent on induction of EBV lytic gene expression, as we observed a similar result in control versus BZLF1-depleted cells (Supplementary Fig. 4i).

MAEA KO or amino acid starvation each increased LC3B-II levels, suggestive of autophagy induction downstream of mTOR inhibition (Supplementary Fig. 5a). Combined MAEA depletion and amino acid starvation further increased LC3B-II levels, which reached a similar magnitude as observed in cells treated with the mTOR inhibitor Torin 1[61] (Supplementary Fig. 5a). We also observed increased numbers of LC3B puncta in MAEA-depleted than control YCCEL1 under amino acid starvation conditions (Supplementary Fig. 5b, c). Furthermore, transmission electron microscopy demonstrated increased autophagosome-like double membrane structures[62] in MAEA-depleted cells, consistent with increased autophagy (Supplementary Fig. 5d, e). We also observed increased numbers of lipid droplets in MAEA-

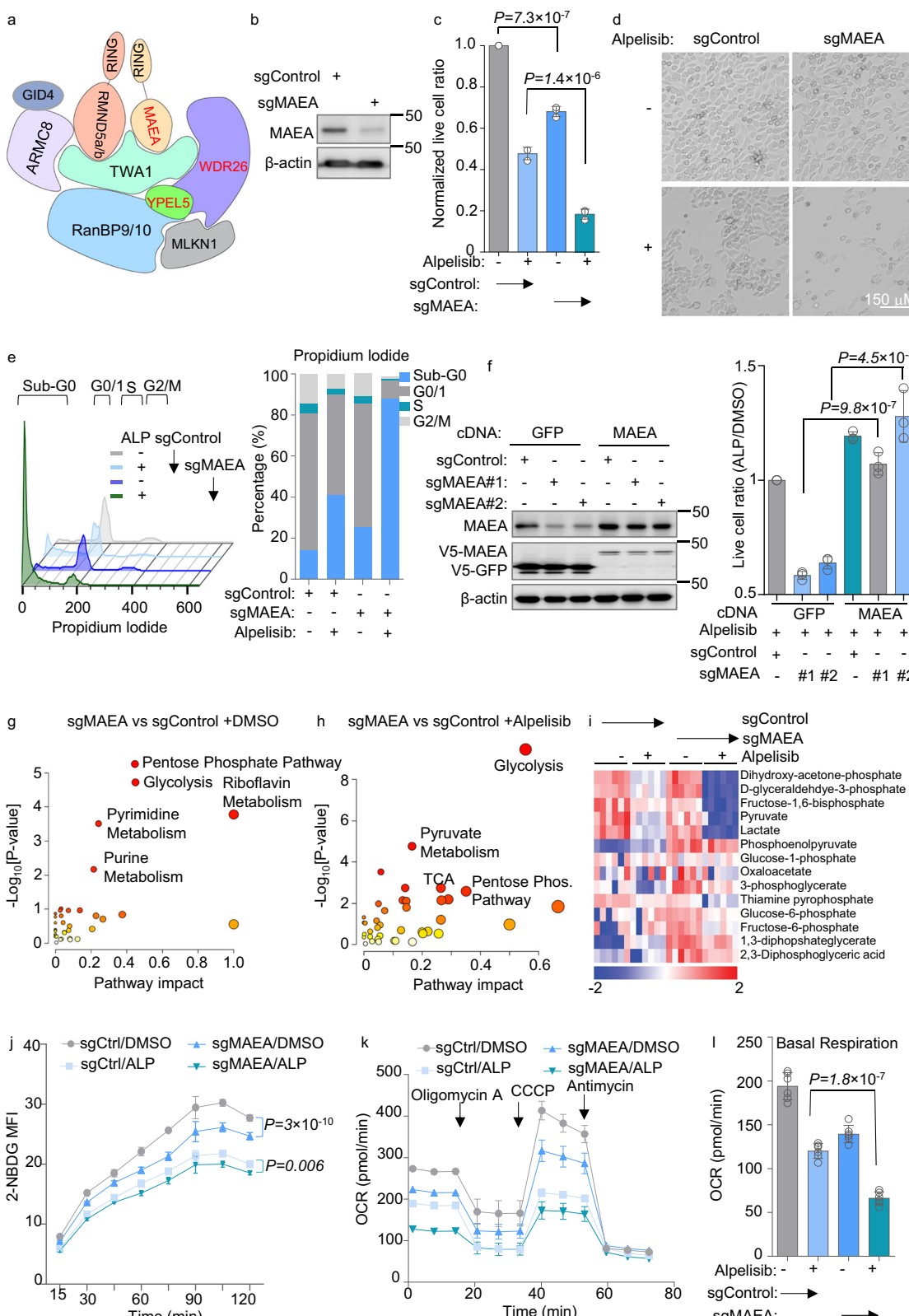

depleted cells (Supplementary Fig. 5f). Notably, mTORC1 inhibition can result in either lipid droplet biogenesis versus autophagic consumption in a cell-type specific manner[63–65], suggesting the former occurs in YCCEL1. However, given emerging CTLH lipid metabolism roles[30,31], it is also plausible that increased numbers of lipid droplets in MAEA-depleted cells may indicate CTLH roles in lipid droplet metabolism.

To screen for MAEA substrates that regulate mTOR activity, we performed whole cell proteomic analysis. Indicative of on-target CRISPR editing, MAEA itself was one of the most significantly depleted proteins in cells expressing MAEA sgRNA relative to cells expressing control sgRNA. The abundances of MKLN1 and ZMYND19 were amongst the most significantly increased by CTLH perturbation (Fig. 3d and Supplementary Data 7), in agreement with studies[28,43,57,66]

**Fig. 2 | Key CTLH E3 ubiquitin ligase roles in EBVaGC survival and metabolism regulation. a** Schematic of CTLH, adapted from Mohamed et al.[28,57]. Screen hits are in red. **b** Immunoblot of WCL from YCCEL1 expressing control vs. MAEA sgRNA. **c** Mean ± SD live cell ratios from $n = 3$ replicates of YCCEL1 in (**b**) treated with 0.5 μM alpelisib vs DMSO. *P*-values by two-tailed one-way ANOVA. **d** Representative $n = 3$ biologically independent replicate bright-field images of (**c**). **e** Cell cycle analysis of propidium iodide (PI) stained YCCEL1 in (**c**). Based on FACS analyses (left), percentages of cells (right) in cell cycle phases were calculated. Representative $n = 3$ biologically independent replicates. **f** MAEA cDNA rescue. Left, immunoblot analysis of WCL from YCCEL1 expressing GFP vs CRISPR-resistant MAEA cDNA with control or MAEA sgRNA. Right, mean ± SD live cell ratios from $n = 3$ replicates of YCCEL1 expressing control GFP vs MAEA cDNA and treated with 0.5 μM alpelisib for 7 days. *P*-values by two-tailed one-way ANOVA. Live cell values were normalized to DMSO-treated cells. **g, h** Metabolic pathway impact analysis from liquid chromatography mass spectrometry (LC/MS) analyses ($n = 6$ independent replicates) of YCCEL1 expressing MAEA vs control sgRNAs in (**g**) DMSO and (**h**) 0.5 μM alpelisib for 30 h. *x*-axis shows pathway impact values, *y*-axis shows the -log₁₀ *P*-value from MetaboAnalyst 3.0[109] Topology analysis. **i** Glycolysis heatmap. Row Z-scores from LC/MS analysis of YCCEL1 cells in (**g**) and (**h**). Z-scores show SDs from the mean value in each row. **j** 2-NBDG uptake in YCCEL1 expressing Control (Ctrl) or MAEA sgRNAs and treated with DMSO or alpelisib (ALP). Means ± SD mean fluorescence intensity (MFI) from $n = 3$ independent replicates. *P*-values are analyzed by two-tailed two-way ANOVA. **k** Oxygen consumption rates (OCRs) of YCCEL1 as in (**g**) and (**h**) following addition of oligomycin A, CCCP, and antimycin. Mean ± SEM from $n = 6$ independent replicates. **l** Basal OCR of YCCEL1 as in (**k**). Mean ± SD from $n = 6$ independent replicates. *P*-values by two-tailed one-way ANOVA. Immunoblot analysis is representative of at least $n = 3$ biologically independent replicates. Source data are provided as a Source Data file for (**b**), (**c**), (**e**), (**f**), and (**j**–**l**).

which previously identified each as CTLH substrates in other cell contexts. We validated by immunoblot that MAEA depletion increased MKLN1 and ZMNYD19 abundances in both YCCEL1 and SNU-719 EBV+ gastric carcinoma cells (Supplementary Fig. 6a). Likewise, ZMYND19, and to a somewhat lesser extent MKLN1 half-lives were 1.9 and 7.7 h, respectively in control cells, but each were significantly stabilized by MAEA KO (Fig. 3e, f). Bortezomib proteasome inhibition increased steady-state MKLN1 and ZMYND19 levels, and increased levels of inducibly expressed ZMYND19 or MKLN1, whose steady-state levels were otherwise low (Supplementary Fig. 6b–d). Interestingly, MKLN1 is not only a CTLH substrate, but is also a component of particular CTLH complexes[26,31,43,67], and MKLN1 depletion also increased steady-state ZMYND19 levels (Supplementary Fig. 6e). MAEA KO also significantly increased MKLN1 and ZMYND19 levels in alpelisib-treated cells. By contrast, alpelisib alone did not increase MKLN1 or ZMYND19 abundances, whereas alpelisib treatment did not further increase ZMYND19 or MKLN1 levels in MAEA-depleted cells (Supplementary Fig. 6f–h and Supplementary Data 7). Taken together, these results support a model in which MKLN1 is both a component of and substrate of CTLH in YCCEL1, as previously suggested in other cell types[26,31,43], and that MKLN1 also facilitates ZMYND19 degradation.

We next tested whether ZMYND19 and/or MKLN1 were necessary for YCCEL1 cell death triggered by MAEA KO plus alpelisib. We expressed control sgRNA or sgRNA targeting ZMYND19, MKLN1, or MAEA and then treated cells with vehicle or alpelisib for 7 days. As shown in Fig. 3g, while KO of either ZMYND19 or MKLN1 alone failed to rescue alpelisib-treated MAEA KO cell survival, ZMYND19/MKLN1 double KO significantly rescued viability of alpelisib-treated cells, suggesting a partially redundant or joint ZMYND19/MKLN1 role (Fig. 3g). Furthermore, electroporation of in vitro transcribed (IVT) ZMYND19 and MKLN1 mRNA (used to provide a burst of over-expression that compensated for their short half-lives) decreased mTOR target S6K T389 phosphorylation and 4E-BP1 S65 phosphorylation levels in MKLN1/ZMYND19 double KO 293T (Fig. 3h). However, in MKLN1/ZMYND19 double KO 293T, reconstitution of MKLN1 or ZMYND19 expression alone was not sufficient to block mTOR activity (Fig. 3h). In unedited YCCEL1, which had low endogenous MKLN1 and ZMYND19, MKLN1 overexpression was sufficient to downmodulate S6K and 4E-BP1 phosphorylation, likely in combination with endogenous ZMYND19, as overexpression of both MKLN1 and ZMYND19 more strongly blocked their phosphorylation (Supplementary Fig. 6i). Similarly, over-expression of both MKLN1 and ZMYND19 induced higher levels of cell death than MKLN1 over-expression did in alpelisib-treated YCCEL1 (Supplementary Fig. 6j, k). These data suggest that ZMYND19 and MKLN1 may jointly inhibit mTOR activity.

WDR26 is critical for nuclear but not cytoplasmic CTLH complex formation[67–69]. We therefore tested effects of WDR26 KO in YCCEL1. As suggested by the CRISPR screen, WDR26 KO and alpelisib treatment synthetically reduced YCCEL1 live cell numbers (Supplementary Fig. 7a). However, in contrast to MAEA KO, WDR26 KO did not increase MKLN1 or ZMYND19 levels (Supplementary Fig. 7b). Likewise, WDR26 KO did not reduce mTOR activity, as judged by immunoblot analysis of S6K and 4EBP1 phosphorylation (Supplementary Fig. 7c). Taken together, these results suggest that in the presence of PI3K inhibition by alpelisib, distinct nuclear and cytoplasmic CTLH complexes may have independent synthetic lethal roles.

**CTLH, MKLN1, and ZMYND19 localize to lysosomal membranes**
MKLN1 can support trafficking of cargo to lysosomes, suggesting that it can associate with lysosomes, likely at the cytoplasmic-facing outer membrane leaflet[70,71]. Since mTORC1 activity is highly regulated at the lysosome outer membrane[6,7], we therefore investigated CTLH sub-cellular distribution, using GFP-tagged MAEA as a readout in live YCCEL1. Consistent with previous studies[26,68], the majority of MAEA localized to the nucleus, but cytoplasmic subpopulations were also observed (Fig. 4a). To better characterize the cytoplasmic puncta, we stained lysosomes with LysoTracker Red in cells expressing MAEA-GFP. Interestingly, a subset of MAEA-GFP puncta dynamically abutted lysosomes (Fig. 4a, b, Supplementary Fig. 8a, and Supplementary Movie 1), suggesting that CTLH may be transiently associate with the lysosome outer membrane, potentially via protein-protein interactions.

To further characterize whether ZMYND19 and/or MKLN1 localize to lysosomal outer membrane regions, we used lysosomal immunopurification[72]. We established YCCEL1 with stable expression of integral lysosomal transmembrane protein 192 (TMEM192) tagged with three tandem HA-epitopes, which were exposed to the cytosol (HA-Lyso cells)[72] (Fig. 4c). Immunoblot analysis of whole cell lysate (WCL) versus lysosome immunoprecipitation (Lyso-IP) samples suggested appropriate fractionation, as lysosomal-associated membrane protein 1 (LAMP1) was enriched in Lyso-IP samples as compared with WCLs, whereas markers for other cellular compartments, including cytosolic GAPDH, Golgi Golgin97, ER calreticulin or mitochondrial VDAC1 were not detected in Lyso-IP samples (Fig. 4c, d). We detected MAEA in WCL, but not appreciably in the Lyso-IP fraction. However, MKLN1 was readily detectable in both WCL and Lyso-IP samples from YCCEL1 MAEA KO cells, in potential agreement with prior studies that identified MKLN1 localization to lysosomes in neuron cells[70,71]. ZMYND19 was preferentially detected in the Lyso-IP fraction. Alpelisib did not appreciably alter MKLN1 or ZMYND19 association with lysosomes (Fig. 4d). Furthermore, transiently expressed ZMYND19 and MKLN1 were highly enriched in Lyso-IP samples, whereas negative control GFP was not (Fig. 4e).

To further investigate ZMYND19 subcellular localization, we performed confocal microscopy imaging, which identified overlap between ZMYND19 and lysosomal marker LAMP2 signals, suggestive of co-localization within the limits of detection of this approach (Fig. 4f and Supplementary Fig. 8b). Similarly, subpopulations of MKLN1 and ZMYND19 co-immunoprecipitated with lysosomes, both in control cells and in cells with CRISPR MAEA depletion (Supplementary Fig. 8c).

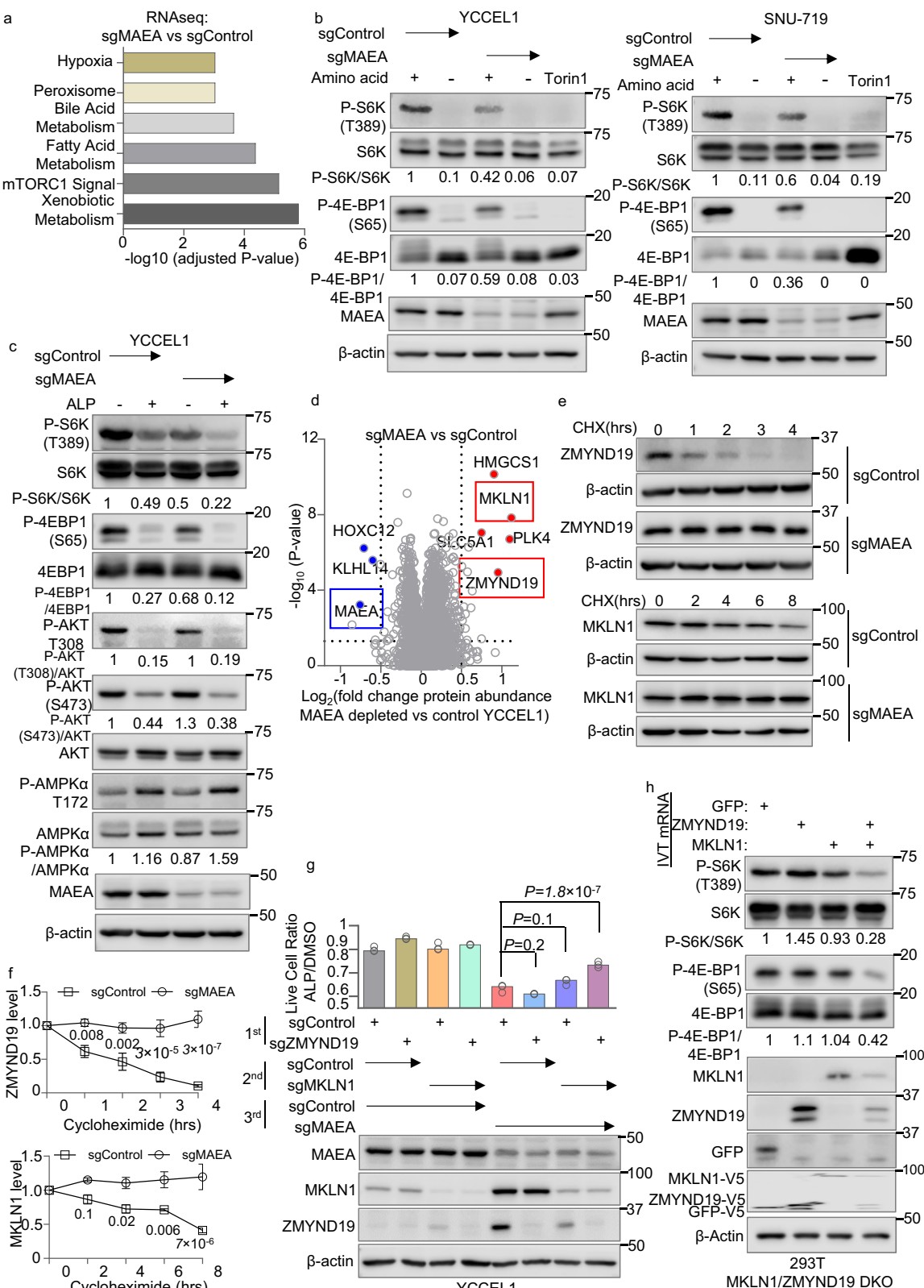

However, MAEA association with LysoIP material was difficult to detect (Fig. 4d and Supplementary Fig. 8c), suggesting that CTLH may only transiently associate with lysosomes, for instance as it targets ZMYND19. Interestingly however, CRISPR ZMYND19 KO reduced the amount of MKLN1 that co-purified with lysosomes (Fig. 4g). Further suggestive of their cytoplasmic subcellular localization, Proteinase K degraded both MKLN1 and ZMYND19 in YCCEL1 that were gently

Dounce homogenized to disrupt plasma membranes, but lysosome resident Cathepsin L was largely protected (Fig. 4h). As a positive control, Cathepsin L was degraded by Proteinase K addition to Dounce homogenized lysates that were also treated with the detergent Triton X-100 to disrupt lysosomal membranes (Fig. 4h). Taken together, these data support the model that ZMYND19 recruits MKLN1 to the cytoplasm-facing lysosomal outer membrane leaflet region (Fig. 4i).

**Fig. 3 | CTLH substrates ZMYND19 and MKLN1 inhibit mTOR. a** MSigDB hallmark pathway analysis of differentially expressed genes (log$_2$ foldchange mRNA abundance >0.5 or < −0.5 and with adjusted *P*-value < 0.05) between YCCEL1 expressing MAEA vs control sgRNAs, performed with two-tailed enrichment analysis in Enrichr[110]. **b** Immunoblot analysis of WCL from YCCEL1 (left) or SNU-719 (right) expressing control or MAEA sgRNA and then cultured in the absence or presence of amino acids for 50 min. Cells treated with Torin 1 (100 nM) for 3 h as positive control. **c** Immunoblot analysis of WCL from YCCEL1 expressing control or MAEA sgRNA and treated with alpelisib 0.5 μM for 8 h. **d** Volcano plot of whole cell proteomic analysis of MAEA-depleted vs control cells to identify candidate YCCEL1 substrates. Shown are -log10 (*P*-value, two-tailed *t*-test) *y*-axis vs Log2 (fold change in protein abundance) *x*-axis between YCCEL1 that expressed MAEA vs control sgRNA, from *n* = 3 biological independent replicates. **e** Immunoblot analysis of WCL from YCCEL1 expressing control or MAEA sgRNA and treated with cycloheximide (CHX) for the indicated number of hours. **f** Mean ± SD ZMYND19 (left) or

MKLN1 (right) abundances in YCCEL1 expressing MAEA vs control sgRNAs. Shown are β-actin normalized abundances from *n* = 3 immunoblots, analyzed with Licor Image Studio software. *P*-values by two-tailed two-way ANOVA. **g** Live cell ratios of YCCEL1 expressing the indicated control, ZMYND19 and/or MKLN1 sgRNAs and then treated with 0.5 μM alpelisib vs DMSO for 5 days. Shown are mean ± SD values from *n* = 3 independent replicates of alpelisib vs DMSO treated cells. *P*-values were calculated by two-tailed one-way ANOVA. Bottom, immunoblot analysis of WCL from cells expressing the indicated combinations of three sgRNAs, which were sequentially expressed by lentiviral transduction, just prior to alpelisib vs DMSO treatment. **h** Immunoblot analysis of WCL from HEK-293 MKLN1 and ZMYND19 double knockout (DKO) single cell clones electroporated with in vitro transcribed (IVT) mRNAs encoding GFP, ZMYND19, or MKLN1. Cells were collected 3 h post-electroporation. β-Actin was used as the loading control. Immunoblot analysis is representative of *n* = 3 biologically independent replicates. Source data are provided as a Source Data file for (**b**), (**c**), and (**f**–**h**).

## ZMYND19 and MKLN1 association and mTOR blockade domains

To gain further insights into the association between MAEA, MKLN1, and ZMYND19, we used co-immunoprecipitation analysis. MAEA and MKLN1 reciprocally co-immunoprecipitated one another, but not with control GFP (Supplementary Fig. 9a). Similarly, MAEA and MKLN1 co-immunoprecipitated with V5-tagged ZMYND19 (Supplementary Fig. 9b), consistent with prior studies[57,73]. To test if the ZMYND19 C-terminal zinc finger was required for association with MKLN1 or MAEA, we expressed V5-tagged full-length or zinc finger-deleted ZMYND19 (ΔZnF, Fig. 5a) in YCCEL1. Zinc finger deletion reduced, but did not complete abrogate MKLN1 co-immunoprecipitation with ZMYND19, whereas it strongly perturbed association between ZMYND19 and MAEA in both YCCEL1 and 293T cells (Fig. 5b and Supplementary Fig. 9c). However, ΔZnF ZMYND19 maintained the ability to associate with lysosomes, albeit perhaps somewhat less robustly (Supplementary Fig. 9d). These data suggests that the ZMYND19 zinc finger mediates association with both MAEA and MKLN1, perhaps when the latter of which is a component of CTLH.

To then test ZYMND19 and/or MKLN1 effects on mTOR blockade, we electroporated a HEK-293T single cell MKLN1/ZMYND19 double KO clone with control GFP, ZMYND19, MKLN1, or ZMYND19 and MKLN1 mRNAs. We used HEK-293T single cell KO so that we could cleanly add back at elated levels MKLN1 alone, ZMYND19 alone, or both together. Overexpression of both ZMYND19 and MKLN1 impaired mTOR activity, as judged by a reduction in S6K and 4E-BP1 phosphorylation, whereas overexpression of MKLN1 alone or in combination with ΔZnF ZMYND19 only mildly impaired their phosphorylation (Fig. 5c). Overexpression of ZMYND19 alone did not appreciably block S6K or 4E-BP1 phosphorylation (Fig. 5c).

We next examined MKLN1 domains important for mTOR blockade. MKLN1 is comprised of an N-terminal LisH domain, a CTLH domain, and six Kelch repeats (Fig. 5d, e), the latter of which can assemble β-propeller structures to support protein–protein interactions[74]. We therefore tested the effects of progressive C-terminal deletion of MKLN1 kelch repeats on its ability to inhibit mTOR, in combination with full-length ZMYND19. This analysis indicated that MKLN1 kelch repeats 2–6 were dispensable for mTOR blockade, as judged by inhibition of S6K or 4E-BP1 phosphorylation (Fig. 5e). Interestingly, LysoIP demonstrated that Δkelch2-6, and in particular Δkelch1-6, increased the fraction of MKLN1 associated with lysosomes (Supplementary Fig. 9e). However, expression of ZMYND19 together with MKLN1 C-terminal deletion mutants lacking all six Kelch repeats or lacking the CTLH domain and the six kelch repeats, resulted in S6K and 4E-BP1 hyper-phosphorylation. Notably, ZMYND19 steady-state levels were lower when co-expressed with MKLN1 constructs that diminished mTOR activity (full length, Δkelch3-6 or Δkelch2-6) (Fig. 5e). These data indicate that the first MKLN1 kelch repeat is needed for mTOR inhibition, but not for lysosomal association.

We next tested whether the MKLN1 CTLH domain was necessary for mTOR blockade, and whether MKLN1 kelch repeat 1, or rather any MKLN1 kelch repeat, was necessary for mTOR blockade. To do so, we electroporated in vitro transcribed RNAs encoding ZMYND19 together with full-length MKLN1 or with MKLN1 deletion mutants lacking the CTLH domain, kelch repeat 1, or kelch repeat 2 into 293T MKLN1/ZMYND19 double KO cells (Fig. 5f). This analysis indicated that the CTLH domain was necessary for mTOR inhibition. Likewise, since kelch 1 but not kelch repeat 2 was required for mTOR inhibition, our results suggest that the MKLN1 kelch repeat 1 plays a non-redundant role (Fig. 5g). As described above, steady-state ZMYND19 levels were higher when co-expressed with MKLN1 deletion mutants that fail to block mTOR (Fig. 5g, lanes 3–5). Interestingly however, each of these deletion mutants co-immunoprecipitated with ZMYND19-HA (Supplementary Fig. 9f). Taken together, these results are consistent with a model in which MKLN1 kelch repeats 1–6 downmodulate its recruitment to lysosomes, but that association with the ZMYND19 zinc finger domain relieves this inhibition, and that the ZMYND19/MKLN1 complex blocks mTOR in a manner dependent on the MKLN1 N-terminal CTLH and K1 domains, but to a lesser extent on the ZMYND19 zinc finger (Fig. 5h).

## ZMYND19 associates with Raptor and RagA/C

We hypothesized that ZMYND19 and/or MKLN1 interact with machinery that control mTOR activation at the outer lysosomal membrane. To test this, we performed co-immunoprecipitation analysis on lysates from cells that expressed FLAG-tagged GFP, TMEM192, MKLN1, or ZMYND19. Interestingly, FLAG-ZMYND19, but not any of the other baits, robustly co-immunoprecipitated Myc-tagged Raptor (Fig. 6a). In support, Alphafold Multimer 2 predicted that ZMYND19 associates with Raptor through its zinc finger domain (Fig. 6b and Supplementary Fig. 10a). Consistent with this model, Raptor co-immunoprecipitated with full length, but to a significantly lesser extent, ΔZnF ZMYND19, in both YCCEL1 and 293T cells (Fig. 6c and Supplementary Fig. 10b). Furthermore, Raptor co-immunoprecipitated with MKLN1 only when it was co-expressed with ZMYND19 (Fig. 6d), whereas ZMYND19 co-immunoprecipitated with Raptor and MAEA even in extracts from MKLN1 depleted cells (Supplementary Fig. 10c). Taken together, these data suggest that the ZMYND19 zinc finger supports association with MKLN1, Raptor and CTLH.

mTORC1 activity is tightly regulated by Rag proteins and Rheb[75–77]. Rag are obligate heterodimers, in which RagA or RagB pair with RagC or RagD to form four distinct complexes[78] that can each bind Raptor to recruit mTORC1 to lysosomes when amino acid supplies are abundant[79–81] (Fig. 6e). We therefore tested whether ZYMND19 and MKLN1 associate with RagA/C or Rheb. To do so, FLAG-tagged MKLN1 and ZMYND19 were co-expressed with Myc-tagged Raptor and either HA-tagged Rheb or HA-RagA/C. Whereas Rheb did not appreciably

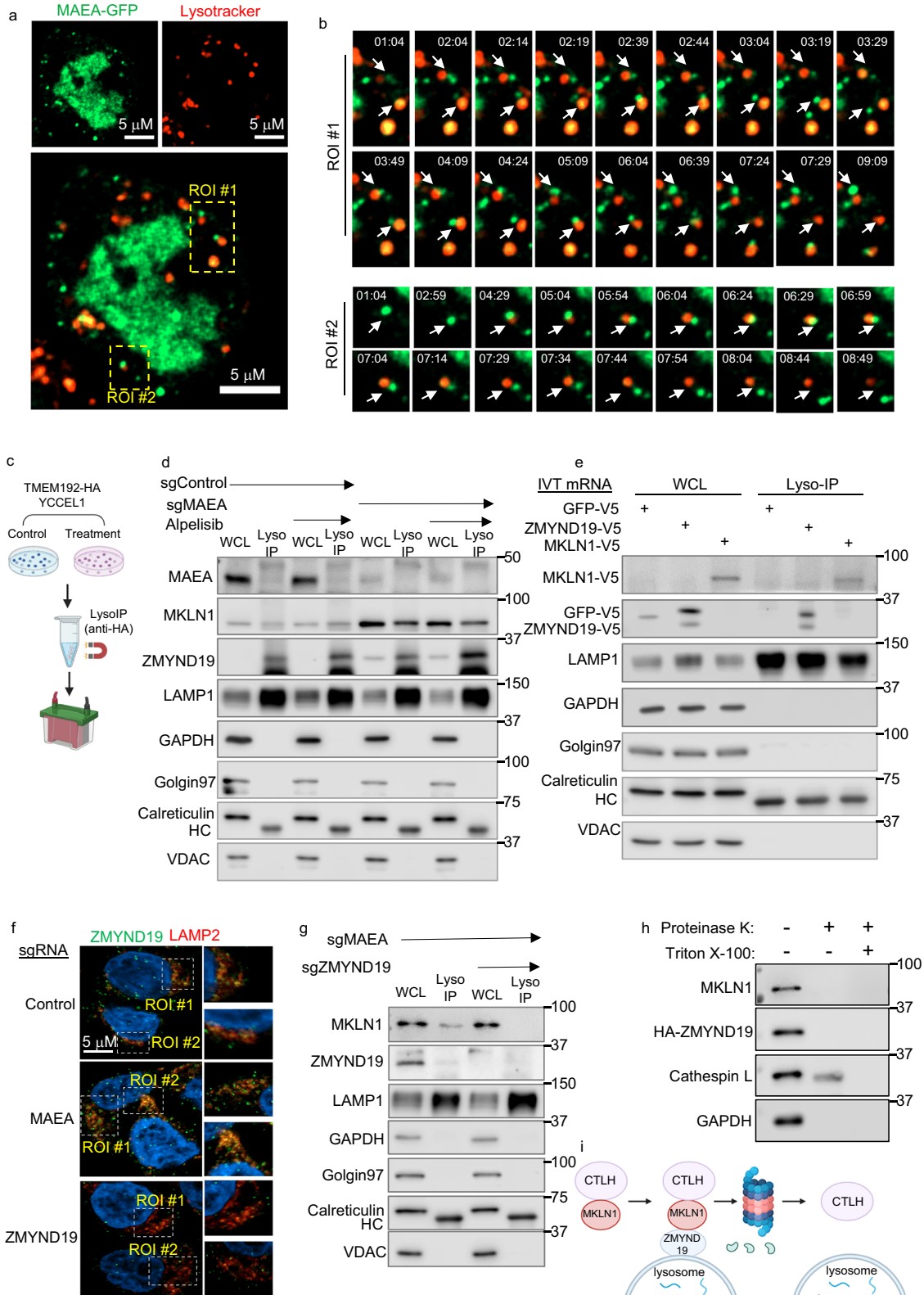

associate with ZMYND19 and MKLN1, RagC and to a lesser extent RagA co-immunoprecipitated (Fig. 6e).

GTP-bound RagA/B and GDP-bound RagC/D recruit mTORC1 to lysosome outer membrane sites, where mTORC1 is then activated by GTP-bound Rheb[79,80,82] (Fig. 6e). To examine whether ZMYND19 and MKLN1 preferentially associate with either GTP-loaded RagA/C state, we co-expressed RagA and RagC point mutants restricted to the GTP-

or GDP-bound conformations[80]. Interestingly, while these expressed at lower levels than wildtype RagA or C, we nonetheless found that the $RagC^{GTP}$ point mutant expressed with its $RagA^{GDP}$ counterpart co-immunoprecipitated to a somewhat higher level with ZMYND19/MKLN1 than $RagC^{GDP}$ co-expressed with $RagA^{GTP}$ (Fig. 6e). These data suggest that ZMYND19 and MKLN1 may preferentially associate with the inactive $Rag^{GTP}$ state, though we did not specifically test this

**Fig. 4 | CTLH substrates ZMYND19 and MKLN1 associate with lysosomes upon CTLH inhibition. a** Confocal microscopy analysis of CTLH subcellular distribution. YCCEL1 expressing GFP-tagged MAEA for 24 h were stained with LysoTracker red. **b** Two regions of interest (ROI) marked in (**a**) were quantitated over an ~8 min timecourse to analyze for dynamic association between MAEA-GFP and lysosomes. **c** Schematic diagram of lysosomal immunopurification (LysoIP)[72]. HA-epitope tagged lysosomal transmembrane protein TMEM192 was stably expressed in YCCEL1. TMEM192HA decorated lysosomes were immunopurified with anti-HA magnetic beads. Created in BioRender. Guo, R. (2025) https://BioRender.com/580vurf. **d** Immunoblots of WCL vs LysoIP samples from YCCEL1 TMEM192-HA expressing cells that also expressed control or MAEA sgRNAs and that were treated with 0.5 µM alpelisib for 24 h. Lysosomal marker LAMP1, Golgi marker Golgin97, ER marker calreticulin, mitochondrial marker VDAC1, and cytosol marker GAPDH are shown as controls. HC, immunoglobulin heavy chain. **e** Immunoblots of WCL vs LysoIP samples from YCCEL1 transfected with in vitro transcribed (IVT) mRNAs encoding V5-tagged GFP, ZMYND19, or MKLN1. **f** Analysis of ZMYND19 and LAMP2 co-localization. Representative confocal microscopy images of YCCEL1 expressing

control, MAEA, or ZMYND19 sgRNA. ZMYND19 and lysosomal marker LAMP2 were stained with Alexa Fluor 488 (green) or Alexa Fluor 595 (red) conjugated antibodies. **g** Immunoblot analysis of WCL vs LysoIP samples from YCCEL1 TMEM192-HA expressing cells that also expressed MAEA and ZMYND19 sgRNAs. HC: immunoglobulin heavy chain. **h** Analysis of MKLN1 and ZMYND19 subcellular localization. YCCEL1 with stable HA-ZMYND19 were gently lysed by Dounce homogenization and co-incubated with proteinase K (10 µg/ml) for 15 min on ice. As a positive control, Triton X-100, which disrupts lysosomal membranes, was added together with proteinase K. Shown are immunoblot analyses following the indicated proteinase K and Triton X-100 treatments, including for the intra-lysosomal resident protein Cathepsin L. **i** Schematic model. A cytosolic subpopulation of MKLN1-containing CTLH complexes targets lysosome-associated ZMYND19 and MKLN1 for proteasomal degradation. Immunoblot analysis and confocal microscopy are representative of at least *n* = 3 biologically independent replicates. Source data are provided as a Source Data file for (**d**), (**e**), (**g**), and (**h**). Created in BioRender. Guo, R. (2025) https://BioRender.com/jpdx7d4.

association under fed versus starved conditions. Since folliculin has GTPase-activating activity towards RagC/D[83], we also tested if it was important for mTORC1 inhibition by ZMYND19 and MKLN1. As expected, folliculin depletion strongly suppressed S6K and 4E-BP1 phosphorylation, consistent with its roles in support of mTORC1 lysosomal recruitment. Interestingly, combined CRISPR folliculin and MAEA depletion further reduced S6K and 4E-BP1 phosphorylation, suggesting that they may independently block mTORC1 (Supplementary Fig. 10d). However, MKLN1/ZMYND19 are unlikely to act through GATOR1, since MAEA KO inhibited mTORC1 in cells with combined KO of the GATOR1 NPRL2 catalytic subunit (Supplementary Fig. 10e).

## ZMYND19/MKLN1 disrupts mTORC1 binding to Rheb and TOS motifs

We tested if MAEA KO blocked mTORC1 activation upon amino acid stimulation of amino acid starved cells. Control or MAEA-depleted YCCEL1 or 293T were grown in amino acid-free media for 50 min and then stimulated by amino acid add back for 10 min. Immunoblot analysis demonstrated that MAEA KO diminished the magnitude of S6K and 4E-BP1 phosphorylation triggered by amino acid stimulation (Supplementary Fig. 11a). Next, we performed confocal microscopy to define whether MAEA knockout altered mTOR subcellular localization following amino acid stimulation. Interestingly, mTOR and the lysosomal resident protein LAMP2 co-localized in both control and MAEA-depleted cells following amino acid stimulation, suggesting that MKLN1/ZMYND19 do not block mTORC1 lysosomal recruitment (Fig. 7a, b).

To further test the model that ZMYND19/MKLN1 block mTORC1 activation at a step distal to lysosomal recruitment, we stably expressed either Raptor or a published Raptor-Rheb fusion protein (referred to hereafter as Raptor-Rheb), in which Raptor is fused to the Rheb C-terminal tail 15 amino acids. These Rheb residues serve as a lysosomal targeting sequence to dictate constitutive Raptor-Rheb lysosomal outer membrane localization, even upon amino acid starvation[84]. We validated that Raptor-Rheb, but not Raptor, drove mTOR lysosomal localization under amino acid starvation, as judged by co-localization with LAMP2 (Supplementary Fig. 11b). Importantly, MAEA depletion did not perturb mTOR co-localization with LAMP2 (Supplementary Fig. 11c, d), even though it did impair mTORC1 activity in cells that stably expressed Raptor-Rheb, as judged by S6K and 4E-BP1 phosphorylation levels (Supplementary Fig. 11e). Furthermore, MAEA depletion did not appreciably alter the amount of mTOR that co-purified with lysosomes in LysoIP analysis of cells with stable Raptor-Rheb expression (Supplementary Fig. 11f).

The tuberous sclerosis complex, comprised of TSC1, TSC2, and TBC1D7 components, negatively regulates mTORC1 through Rheb GAP activity[48]. We therefore asked whether mTORC1 inhibition by

ZMYND19 and MKLN1 was dependent on TSC2. However, CRISPR MAEA depletion inhibited S6K and 4E-BP1 phosphorylation in YCCEL1 that were also depleted for TSC2 by either of two sgRNA approaches (Fig. 7c). Furthermore, Rheb overexpression failed to rescue S6K and 4E-BP1 phosphorylation in MAEA KO YCCEL1 cells (Supplementary Fig. 12a). We therefore hypothesized that ZMYND19 and MKLN1 might instead block the association between mTORC1 and Rheb to prevent mTORC1 allosteric activation. To test this hypothesis, we immunopurified GFP-tagged Raptor complexes from cells that co-expressed v5-tagged Rheb. Rheb co-immunoprecipitated with Raptor in control cells, as expected, given their bridging by mTOR[85]. Importantly, MAEA KO precluded Raptor association with Rheb (Fig. 7d and Supplementary Fig. 12b). Interestingly, MAEA KO also impaired Raptor association with the mTORC1 substrates S6K and 4E-BP1 (Fig. 7d and Supplementary Fig. 12c), each of which contain a conserved five-amino acid TOR signaling (TOS) motif[85]. MKLN1 and ZMYND19 over-expression was sufficient to block Raptor association with Rheb, and also impaired Raptor association with S6K and 4E-BP1 (Fig. 7e). Consistent with generalized mTORC1 inhibition, MAEA KO also reduced phosphorylation levels of mTORC1 targets TFEB and TFE3, which do not contain a TOS motif (Supplementary Fig. 12d, e), suggesting that MAEA KO blocks mTOR more broadly than only by inhibiting its association with TOS-containing substrates such as 4E-BP1 and S6K. Taken together, these results are consistent with a model in which loss of CTLH complex activity leads to MKLN1 and ZMYND19 accumulation at lysosomes, which impede mTORC1 association with and activation by Rheb and also its interaction with S6K and 4E-BP1 (Fig. 7f).

## Discussion

Despite intensive study, much remains to be learned about mechanisms that control mTORC1 activity in distinct contexts. Here, we used a genome-wide CRISPR-Cas9 screen to identify targets whose depletion, together with PI3K inhibition by alpelisib, blocked proliferation of an EBV-associated gastric carcinoma model with hyperactive PI3K/mTOR signaling. This analysis revealed that the multi-subunit E3 ligase CTLH is a major regulator of mTORC1 activity by suppressing levels of ZMYND19 and MKLN1. Upon loss of CTLH activity, ZMYND19 and MKLN1 were stabilized and associated with one another and with Raptor at lysosomal outer membrane sites, where they impaired mTORC1 association with Rheb and with TOS motif-containing substrates.

ZMNYD19/MKLN1 together blocked mTORC1, downstream of its lysosomal recruitment by Ragulator-Rag, but without requiring TSC activity. Rather, ZMYND19 associated with Raptor and with GDP-loaded RagC, positioning ZMYND19/MKLN1 to block allosteric mTORC1 activation by Rheb. Indeed, MAEA KO or ZMYND19/MKLN1 overexpression each strongly impaired Raptor and Rheb association.

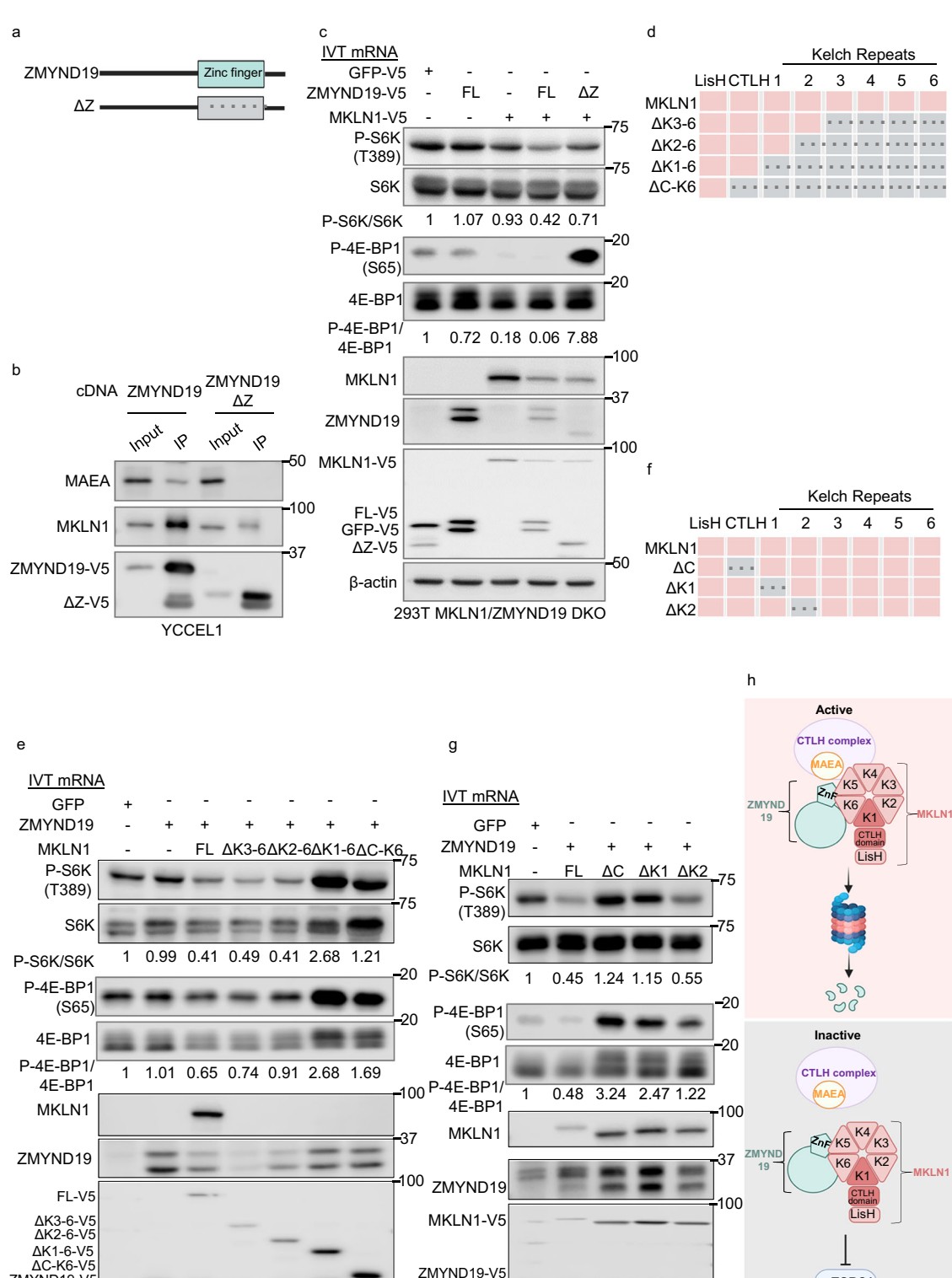

Since Raptor does not directly bind Rheb, but instead associates with Rheb through lysosomal-tethered mTORC1 complexes, our data suggest that ZMYND19 and MKLN1 block mTORC1 association with Rheb, potentially via steric or allosteric interference. Since MAEA knockout or ZMYND19/MKLN1 overexpression also blocked Raptor association with the TOS motif-containing substrates S6K and 4E-BP1[86–89], we anticipate that ZMYND19/MKLN1 may bind to a Raptor surface that occludes its association with TOS. Taken together, our data suggest that ZMYND19/MKLN1 associate with both Rag and Raptor in order position it to then block both mTORC1 association with Rheb and with TOS-containing substrates. Such a mechanism would be distinct from GID regulation in yeast, in which glucose availability induces GID4 expression to drive E3 activity[44,90], and to our knowledge from other cellular factors that negatively regulate mTORC1.

**Fig. 5 | Identification of ZMYND19 and MKLN1 domains important for mTOR inhibition. a** Schematic model of ZMYND19 full length and zinc finger domain deletion mutant ZMYND19 (ΔZ). Zinc finger deletion is indicated by gray box with dotted lines. **b** Analysis of ZMYND19 zinc finger role in association with MKLN1 and MAEA. Immunoblots of 5% input vs anti-V5 immunopurified full length versus ΔZ ZMYND19 from YCCEL1 that stably expressed the indicated cDNAs. **c** Analysis of ZMYND19 zinc finger role in mTOR inhibition. Immunoblot analysis of 293T MKLN1/ZMYND19 double knockout (DKO) single cell clines electroporated with the indicated in vitro transcribed (IVT) mRNAs encoding V5-tagged GFP, full length (FL) or ΔZ ZMYND19. WCL were prepared three hours after electroporation. **d** Model of full length MKLN1 and C-terminal truncation mutants, showing the LisH, CTLH, and six kelch domains. Domain deletions are indicated by gray boxes with dotted lines. **e** Analysis of MKLN1 kelch and CTLH (c) domain roles in mTOR blockade. Immunoblot analysis of 293T MKLN1/ZMYND19 DKO clones electroporated with the

indicated in vitro transcribed mRNAs. WCL were prepared three hours after electroporation. **f** Model of full length MKLN1 and CTLH, Kelch domain 1 (ΔK1) or 2 (ΔK2) deletion mutants. **g** Analysis of mTOR blockade by MKLN1 deletion mutants lacking the CTLH (ΔC), Kelch 1 (ΔK1), or Kelch 2 (ΔK2) domains. Immunoblot analysis of 293T MKLN1/ZMYND19 DKO clones electroporated with the indicated in vitro transcribed mRNAs. WCL were prepared three hours after electroporation. **h** Model of mTOR blockade by the ZMYND19/MKLN1 complex. A MKLN1-containing CTLH complex negatively regulates ZMYND19 and MKLN1. The ZMYND19 zinc finger (ZnF) is important for association with MKLN1. The MKLN1 Kelch domain 1 (K1) and CTLH domains are important for mTOR inhibition. Created in BioRender. Guo, R. (2025) https://BioRender.com/wc17q23. Immunoblot analysis is representative of at least *n* = 3 biologically independent replicates. Source data are provided as a Source Data file for (**b**), (**c**), (**e**), and (**g**).

MAEA was originally named Macrophage Erythroblast Attacher and found to have roles in erythrocyte development[91]. While our data agree with prior publications that MAEA is largely nuclear, we identified a cytoplasmic MAEA subpopulation that dynamically associated with lysosomes. In support, MKLN1 and ZMYND19 are mostly cytoplasmic, suggesting that they are regulated by a non-nuclear CTLH subpopulation. An interesting possibility is that a CTLH subpopulation may traffic to lysosomal regions in response to an environmental and likely nutritional cue. Notably, MKLN1 overexpression drives cytoplasmic trafficking of a CTLH subpopulation[45]. Thus, MKLN1 may direct CTLH to specific cytoplasmic targets[43], which we now suggest include ZMYND19. We speculate that MKLN1 incorporation, perhaps together with additional post-translational modification, drives a CTLH subpopulation to lysosomal outer membrane sites. MKLN1 also has roles in actin- and microtubule-dependent GABA(A) receptor trafficking[70], and MKLN1 may therefore play roles in trafficking a subpopulation of CTLH to lysosomal outer membrane sites.

Little has remained known about ZMYND19. Our data suggest that ZMYND19 constitutively localizes to lysosomal outer membrane sites, where it recruits MKLN1 through its zinc finger domain. As ZMYND19 does not contain transmembrane domains and is not known to have lipid anchoring post-translational modifications, we speculate that ZMYND19 association with Raptor and RagC/D positions it, together with MKLN1, to block mTORC1 activity. We hypothesize that ZMYND19-bound MKLN1 is the substrate for CTLH in a manner dependent upon MAEA and potentially also RMND5a RING activity. In this manner, MKLN1 may serve as a CTLH substrate adaptor, but interestingly also as a CTLH substrate, as suggested by prior studies[26,31,43,67], together with ZMYND19. In support, CRISPR MKLN1 depletion significantly increased ZMYND19 abundance, whereas ZMYND19 depletion did not increase MKLN1 abundance. Since the ubiquitin E2 enzyme UBE2H nearly scored in our screen and is the human homolog of yeast GID3 that supports GID E3 ligase activity[26,43], our data suggest that MAEA RING-dependent ligase activity underlies the observed proteasomal turnover of MKLN1 and ZMYND19.

The CTLH RING subunits RMND5A and MAEA can each target MKLN1 for proteasomal degradation in HeLa cells[43], where MKLN1's half-life is ~24 h, as opposed to ~8 h in YCCEL1, perhaps suggesting a degree of cell-type specific regulation. Notably, RMND5A did not score in our CRISPR screen, but MAEA KO also depletes RMND5A in HeLa[43]. Therefore, further studies will be required to determine whether RMND5A and MAEA have redundant activity towards MKLN1 and ZMYND19, or whether MAEA is uniquely able to target them. Notably, MKLN1 forms a tetramer, and four MKLN1 protomers can bind to two CTLH substrate receptor scaffolding modules in vitro to drive higher-order CTLH structures[67]. Whereas human CTLH assembles into complexes of 600–800 kDa, MKLN1 depletion shifts the complex towards 150–350 kDa[67]. We speculate that high molecular weight MKLN1-containing CTLH complexes target lysosome-bound ZMYND19 for

degradation. Such a mechanism might account for our observation that CTLH transiently associates with lysosomes, since MKLN1 degradation at the lysosome may then alter CTLH structure and localization.

The CTLH homolog GID complex has major metabolic roles that enable yeast to rapidly adapt to the presence of glucose in the extracellular environment. When glucose is abundant, GID ubiquitinates and triggers proteasomal degradation of the gluconeogenic enzymes fructose 1,6-bisphosphatase (FBP1), phosphoenolpyruvate carboxykinase (PCK1), cytoplasmic malate dehydrogenase (MDH2), and isocitrate lyase (ICL1)[22]. Interestingly, GID can assemble with different substrate recognition factors. For instance, GID4 expression is derepressed by the presence of extracellular glucose[92], and GID4 then binds the catalytically inactive core complex to serve as a substrate receptor that drives polyubiquitination of specific gluconeogenesis enzymes[44,55]. GID contains two RING finger domain-containing proteins, GID2 and GID9, which form a heterodimer and whose human homologs are RMND5A and MAEA, respectively.

We found that MAEA KO impaired glucose uptake and reduced extracellular acidification, a key indicator of glycolytic flux. In combination with alpelisib, MAEA KO reduced levels of multiple glycolysis pathway metabolites, including fructose-1,5-bisphosphate, glyceraldehyde-3-phosphate, dihydroxy-acetone-phosphate, pyruvate, and lactate more strongly than in YCCEL1 treated with alpelisib alone. These results contrast somewhat with a recent study, which identified that a RanBPM-containing CTLH complex restricted glycolytic flux in HeLa cells, which was attributed at least in part to non-degradative ubiquitin effects on PKM2 and the lactate dehydrogenase A subunit[28,57]. It is possible that distinct CTLH complexes have opposing effects on glycolysis, and we note that RanBPM did not score in our screen. Alternatively, CTLH may have cell context-specific roles that differ between HeLa cells[28,57], which express several human papillomavirus oncogenes versus in the EBV-infected gastric carcinoma cells with activated PI3K signaling used in this study. A third possibility is that CTLH may negatively regulate glycolytic flux in YCCEL1 by non-degradative ubiquitination, as reported in HeLa[28,57], but that inhibitory effects on mTOR may have predominated in the context of alpelisib treatment. Further studies will be required to differentiate between these interesting possibilities.

Our results highlight an interesting difference upon MAEA depletion in human cells versus RMND5A depletion in the *C. elegans* GID complex. KO of the other GID RING ligase subunit RMND5A increased AMP kinase expression and activation, which reduced mTOR activity, increased autophagic flux, and increased *C. elegans* lifespan[29]. While AMP kinase was identified as a *C. elegans* GID substrate, we did not detect increased AMPK kinase levels by our proteomic or immunoblot analyses of MAEA KO vs control human cells. Thus, it is possible that this GID role has not been conserved in human cells, or may instead require loss of both MAEA and RMND5A ubiquitin ligase activity. Similarly, CTLH can target the transcription repressor HBP1

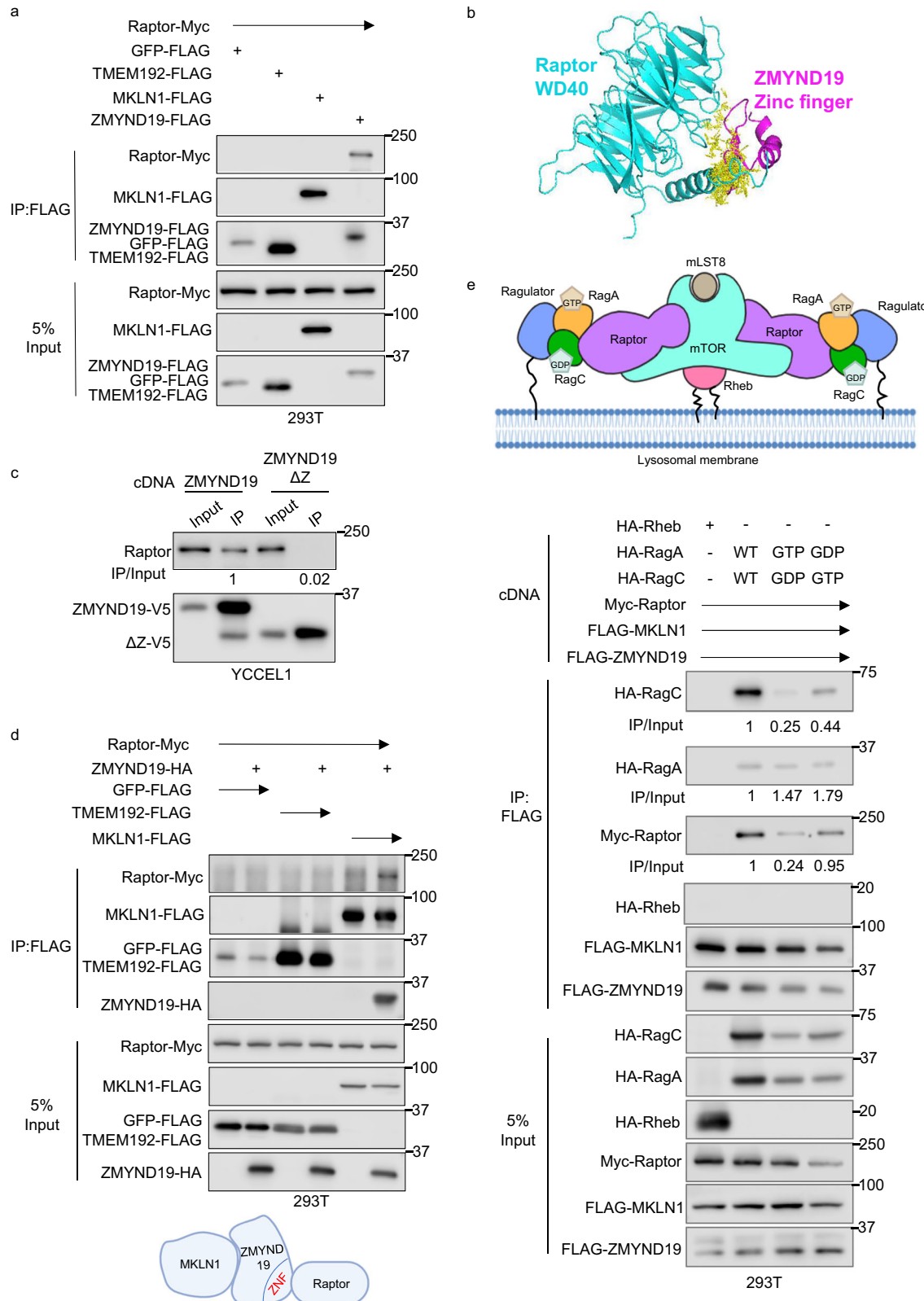

for degradation to support cell cycle[57], though we did not observe significantly increased HBP1 levels in YCCEL1 upon MAEA KO, perhaps indicative of a cell-type specific HBP1 regulatory role.

Two recent studies[30,31] found that CTLH targets 3-hydroxy-3-methylglutaryl (HMG)-coenzyme A (CoA) synthase 1 (HMGCS1), the initial enzyme in the mevalonate pathway for degradation. Interestingly, mTOR inhibition by Torin 1 triggers HMGCS1 degradation by a

MKLN1-containing CTLH complex[31]. Consistent with these studies, we observed strong HMGCS1 upregulation in MAEA-edited YCCEL1. However, in contrast with the Yi et al. study[30], we observed that MAEA KO impaired mTOR signaling in HEK-293T, as judged by immunoblot analysis of S6K and 4E-BP1 phosphorylation. We speculate that differences in the timing of experiments between the two studies may have accounted for this discrepancy. In particular, we performed

**Fig. 6 | Interaction of ZMYND19/MKLN1 and Raptor. a** Analysis of ZMYND19 and MKLN1 association with Raptor. Immunoblot analysis of 5% input vs anti-FLAG immunopurified complexes from HEK-293T transiently transfected with the indicated cDNAs for 24 h. GFP and TMEM192 were used as negative controls for cytosolic and lysosomal membrane proteins, respectively. **b** Alphafold multimer 2 model of ZMYND19 zinc finger and Raptor WD40 association at 3.5 Å. The Raptor WD40 domain is shown in teal, the ZMYND19 zinc finger in magenta, and the predicted interaction interface is detailed in yellow. **c** Analysis of ZMYND19 zinc finger domain role in Raptor association. Immunoblots of 5% input versus anti-V5 immunopurified complexes from YCCEL1 that stably expressed the indicated V5-tagged ZMYND19 full length or zinc finger deletion mutant (ΔZ). **d** Analysis of ZMYND19 roles in association between MKLN1 and Raptor. Immunoblots of 5% input versus ant-FLAG immuno-purified complexes from 293T cells that transiently expressed the indicated epitope-tagged constructs. Created in BioRender. Guo, R.

(2025) https://BioRender.com/do88x9t. **e** Analysis of MKLN1/ZMYND19 association with RagA, RagC, or RHEB. Top, schematic model of mTORC1 associated with Ragulator/RagA/C and RHEB complexes tethered to the lysosomal membrane, adapted from Rogala K. et al.[111]. GTP-bound RagA and GDP-bound RagC heterodimers, tethered to lysosomal membranes by Ragulator, associate with Raptor to recruit mTORC1 complexes, which can then be activated by membrane-associated GTP-loaded RHEB. Shown below is immunoblot analysis of 5% input vs. anti-FLAG immunopurified complexes from 293T transiently transfected with the indicated FLAG-tagged ZMYND19 and MKLN1, Myc-tagged Raptor, HA-tagged Rheb, and HA-tagged wildtype (WT) RagA and RagC or point mutants restricted to the GTP- or GDP-bound conformations[80]. Immunoprecipitation and Immunoblot analysis are representative of at least $n = 3$ biologically independent replicates. Source data are provided as a Source Data file for (**a**), (**c**–**e**).

analyses at a very early timepoint post-CRISPR editing (Day 4), whereas analyses were performed following expansion of single cell HEK-293 MAEA KO clones in the Yi et al.[30] study. It is therefore possible that 293T evolved resistance to MAEA KO-driven mTOR inhibition over the period of expansion of 293T single cell clones in the Yi et al. study. Nonetheless, our results, together with recently published studies[30,31] highlight cross-talk between CTLH and mTOR.

We have not observed changes in CTLH activity towards ZYMND19/MKLN1 upon glucose withdrawal, suggesting that CTLH may not function in a glucose-sensing pathway. Intriguingly, carbon stress regulates assembly of an anticipatory form of the GID E3 ligase, which then awaits expression of the substrate receptor for activation of its activity[55]. Thus, it will be of interest to test whether CTLH assembly might instead be negatively regulated by carbon stress. Alternatively, CTLH phosphorylation by casein kinase 2 regulates UBE2H/CTLH pairing[93], and this may enable cross-talk between receptor-driven kinase signaling, CTLH activity, and mTORC1. Given CTLH cross-talk with HMGCS1, it is also tempting to speculate that a component of the mevalonate or cholesterol biosynthesis pathway may instead regulate CTLH activity. We hypothesize that human CTLH activity can be rapidly toggled in response to an as yet unidentified environmental signal. In this manner, when the nutrient is replete, CTLH degrades the MKLN1/ZMYND19 complex in order to license mTORC1 activation. However, when the nutrient is depleted, we anticipate that a post-translational modification disrupts this activity, enabling ZMYND19/MKLN1 to accumulate, to bind to Raptor and RagA/C and to block mTORC1 association with Rheb and with TOS motif-containing substrates.

CTLH perturbation downregulated mTORC1 in two EBV+ gastric carcinoma models and also in several EBV-negative cell lines. Since CTLH is expressed in a wide range of human cell types and tissues, we suspect that it supports mTORC1 in diverse human cellular contexts, particularly with hyperactive PI3K signaling. This may relate to the observation that CLTH subunits are overexpressed in a variety of human tumors, and that elevated WDR26 levels correlate with poor prognosis[94]. Our results therefore, suggest CTLH may be a therapeutic target. For instance, GID4 small-molecule binders were recently identified[95], and it is likely that CTLH inhibitors can be developed. While germline MAEA KO causes myeloproliferative disease in mice[96], it is possible that CTLH inhibition in combination with low-dose FDA-approved PI3K inhibitors such alpelisib may not have this effect. Thus, CTLH inhibition together with PI3K blockade warrants further investigation as a potential therapeutic modality for EBVaGC and for other tumors dependent upon hyperactive PI3K/mTOR signaling. Interestingly, knockout of the CTLH subunit and screen hit WDR26 did not impair mTOR signaling and did not increase MKLN1 or ZMYND19 abundances. Since WDR26 is important for assembly of nuclear but apparently not cytoplasmic CTLH complexes[67,68], it will be of interest to determine how a nuclear CTLH complex exerts synthetic lethal effects with alpelisib, apparently by a distinct mechanism than by mTOR inhibition.

In summary, a human genome-wide CRISPR/Cas9 screen identified that KO of multiple CTLH E3 ligase subunits blocked EBVaGC proliferation, together with alpelisib. In the absence of degradation by CTLH, ZMYND19 and MKLN1 were stabilized, accumulated at lysosomal membrane regions, and inhibited both mTORC1 association with Rheb and with the TOS motif-containing substrates 4E-BP1 and S6K. These studies identify a EBVaGC therapeutic target whose inhibition may also be synergistic with PI3K blockade in tumors with aberrantly elevated PI3K activity, and identify a pathway by which CTLH-dependent proteasomal activity tunes mTORC1 activity at the lysosomal membrane.

## Methods

### Cell culture

The EBV+ gastric cancer cell line YCCEL1 was obtained from Elliott Kieff. The EBV+ cell line SNU-719 was obtained from Adam Bass. The EBV- gastric cancer cell line HGC-27 was obtained from Sigma, SNU-1 and SNU-16 were from ATCC. HEK-293T was obtained from ATCC. P3HR-1 with conditional ZTA and RTA immediate early alleles triggered by 4-hydroxytamoxifen (4-HT) were obtained from Elliott Kieff[97]. Cell lines with stable Streptococcus pyogenes Cas9 expression were generated by lentiviral transduction and blasticidin selection (5 mg/mL), as previously described[98]. Cells were cultured in a humidified incubator with 5% $CO_2$ at 37 °C. The cells were routinely tested and certified as mycoplasma-free using the MycoAlert kit (Lonza).

YCCEL1 cells were grown in Eagle's Minimum Essential Medium (EMEM) medium (ATCC) with 10% fetal bovine serum (FBS, Gibco). SNU719, SNU16, Daudi, and P3HR1 cells were grown in RPMI 1640 medium (Gibco, Life Technologies) with 10% FBS. 293T was grown in Dulbecco's Modified Eagle's Medium (DMEM) with 10% FBS. HGC-27 were grown in EMEM (ATCC) with 2 mM Glutamine (Gibco, Life Technologies), 1% Non-Essential Amino Acids (NEAA, Gibco, Life Technologies), and 10% FBS. BYL719 (Alpelisib, Cat#16986, Cayman chemical) was used at 0.5 μM. Antibodies used in the study are listed in the Reporting Summary.

### HEK-293T transfection

HEK-293T cells were transfected as follows. 2 million cells were seeded in 10 cm dishes (Corning). After 24 h, pRK5-based plasmids were transfected with the Effectene transfection reagent (Qiagen, 301425) according to manufacturer's protocol. Empty pRK vector was used to normalize the level of DNA to 2 μg across transfection conditions. Cells were collected 24–30 h post-transfection. The following amounts of cDNA were used in the indicated figures. Figure 6a, 500 ng Raptor-MYC, 50 ng GFP-FLAG, 50 ng TMEM192-FLAG, 300 ng ZMYND19-FLAG, or 300 ng MKLN1-FLAG. Figure 6c, 500 ng Raptor-MYC, 50 ng GFP-FLAG, 50 ng TMEM192-FLAG, 300 ng ZMYND19-FLAG or 300 ng MKLN1-FLAG. Fig. S5C 300 ng ZMYND19-HA, 50 ng GFP-FLAG, 50 ng TMEM192-FLAG, 300 ng MKLN1-FLAG or 300 MKLN1-FLAG truncation mutants. Fig. S5D 300 ng ZMYND19-HA, 50 ng GFP-FLAG, 50 ng

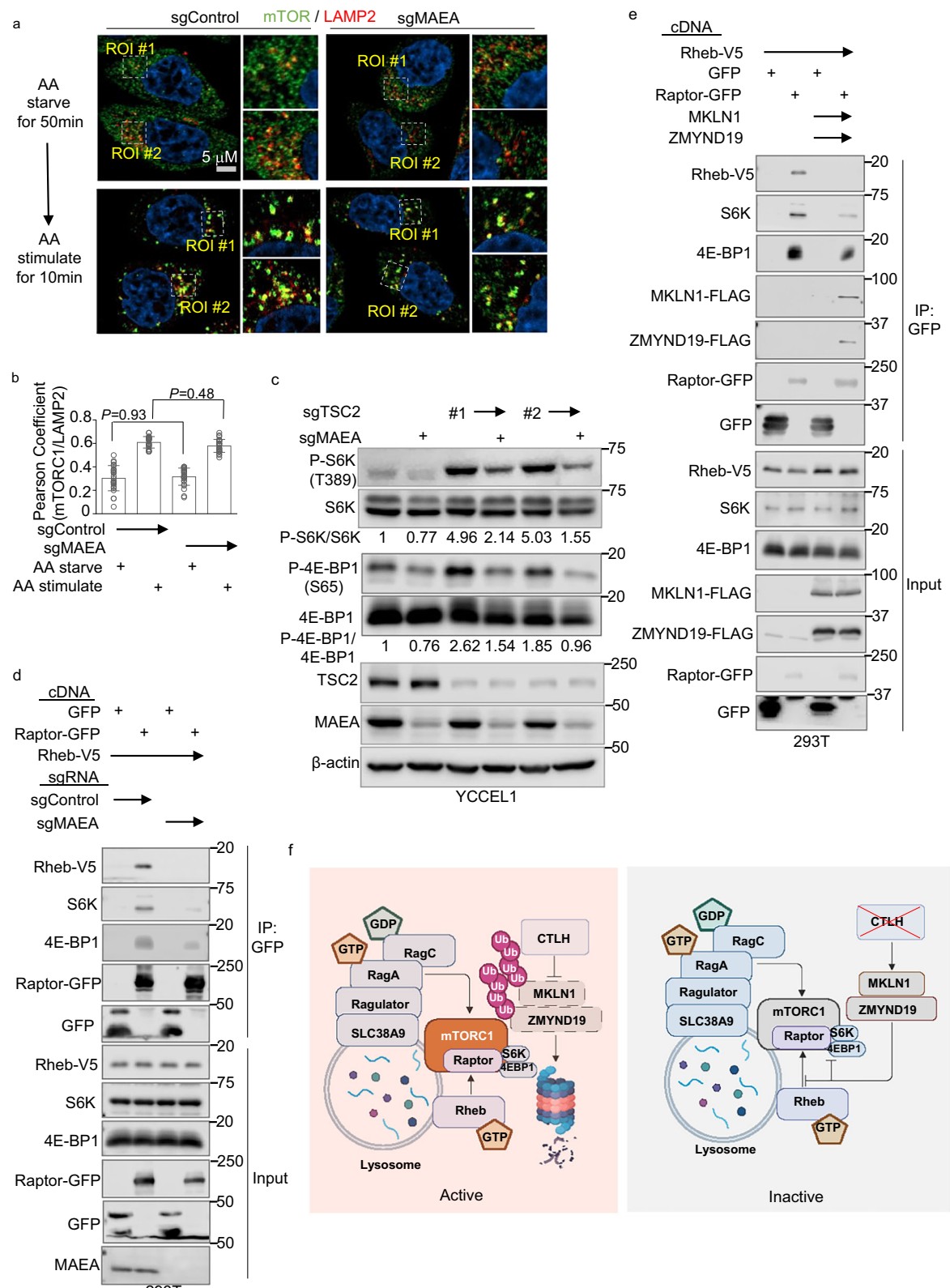

TMEM192-FLAG, 300 ng MKLN1-FLAG or 300 MKLN1 truncates with FLAG tag. Fig. S6C 500 ng Raptor-myc, 300 ng ZMYND19-FLAG, or 300 ng MKLN1-FLAG, 100 ng Rheb-HA, 100 ng RagA or 100 ng RagC. For Fig. 7d, 6 million cells were seeded in 10 cm dishes (Corning). After 24 h, 6 µg pLJM1-Raptor-GFP or pLJM1-GFP (generously provided by Brendan Manning) and 6 µg of PLI-TRC313-Rheb-V5 plasmids were transfected with the PEI reagent (Sigma, 919012). For Fig. 7e, 6 million cells with stable Rheb-V5 expression were seeded into 10 cm dishes (Corning). After 24 h, 6 µg pLJM1-Raptor-GFP or 6 µg of pLJM1-GFP, together with 6 µg of pRK-ZMYND19 and 6 µg of pRKMKLN1, were co-transfected with the PEI reagent. For Supplementary Fig. 12b, 6 million cells were seeded into 10 cm dishes (Corning). After 24 h, 6 µg PLI-TRC313-Rheb-V5 or 6 µg of PLI-TRC313-GFP-V5, together with 6 µg pLJM1-Raptor were co-transfected with the PEI reagent. For

**Fig. 7 | ZMYND19 and MKLN1 block a late stage in mTORC1 activation, distal to lysosomal membrane recruitment. a** Analysis of MAEA depletion effects on amino acid stimulation-driven mTOR lysosomal subcellular localization. Representative confocal microscopy images of YCCEL1 expressing control or MAEA sgRNA and amino acid starved for 50 min and then cultured in amino acid-free media (top) or with amino acid addback for 10 min (bottom). mTOR and lysosomal marker LAMP2 were stained with Alexa Fluor 488 (green) or Alexa Fluor 595 (red) conjugated antibodies, respectively. **b** Quantification of co-localization of mTORC1 and LAMP2 in immunofluorescence images in (**a**). Shown are mean ± SD values from $n = 25$ independent replicates of YCCEL1 expressing MAEA sgRNA vs control sgRNA cells. Pearson's correlation coefficients were analyzed with Fiji. *P*-values were calculated by two-tailed one-way ANOVA. **c** Analysis of whether TSC2 is necessary for mTORC1 inhibition upon MAEA depletion. Immunoblot analysis of WCL from YCCEL1 that expressed the indicated MAEA or TSC2 sgRNAs (#1 and #2). **d** Analysis of MAEA depletion effects on Raptor association with Rheb and with mTORC1 substrates S6K and 4E-BP1. Immunoblot analysis of 5% input versus anti-GFP immuno-purified complexes from 293T control or MAEA KO single cell clones

that were transiently transfected with Rheb-V5, GFP, or Raptor-GFP expression vectors. **e** Analysis of MKLN1 and ZMYND19 co-expression effects on Raptor association with Rheb and with mTORC1 substrates S6K and 4E-BP1. Immunoblots of 5% input versus anti-GFP immuno-purified complexes from 293T cells with stable Rheb-V5 expression and transiently transfected with GFP, Raptor-GFP, MKLN1, and ZMYND19 expression vectors. **f** Schematic model of CTLH, ZMYND19, and MKLN1 roles in mTORC1 regulation. A subpopulation of CTLH homes to lysosome regions. When MAEA is active, CTLH targets ZMYND19 and MKLN1 for proteasomal degradation. Upon loss of MAEA-dependent CTLH activity, ZMYND19 and MKLN1 associate at lysosome outer membrane sites, perhaps through ZMYND19 zinc-finger association with Raptor. Rather than blocking mTORC1 lysosomal recruitment, ZMYND19 and MKLN1 block mTORC1 association with Rheb and with its substrates S6K and 4E-BP1. Created in BioRender. Guo, R. (2025) https://BioRender.com/9oyxckk and https://BioRender.com/algs0ol. Immunoprecipitation and Immunoblot analysis are representative of at least $n = 3$ biologically independent replicates. Source data are provided as a Source Data file for (**b–e**).

Supplementary Fig. 12c, 6 million cells were seeded into 10 cm dishes (Corning). After 24 h, 6 µg prK-Metap2-Flag or 6 µg of pRK-S6K-Flag (generously provided by Kuang Shen), together with 6 µg pLJM1-Raptor-GFP were co-transfected with the PEI reagent.

## Chemicals
Alpelisib (BYL719) was purchased from Cayman Chemical (Cat#16986) and was used at 0.5 µM unless otherwise indicated. ML-210 was purchased from Cayman Chemicals (Cat#23282) and used at 5 µM. Rapamycin (Cat#AY-22989) was purchased from Selleckchem and used at 5 nM. Ferrostatin-1 (Fer-1) was purchased from Sigma-Aldrich (Cat#SML0583) and used at 10 µM. Propidium iodide was purchased from Invitrogen (Cat#P03566) and used at 5 µg/ml in cell cycle analysis, 2-NBDG from ThermoFisher (Cat#N13195) and used at 10 mg/mL in glucose uptake assays, puromycin from Thermofisher (CatA1113803) was used at 10 µg/mL in puromycin chase assays, and cycloheximide from R&D systems (Cat#0970/100) was used at 50 µg/mL for protein half-life and as a control for puromycin chase analysis. Doxycycline was used at 250 ng/ml. Torin 1 was purchased from MedChemExpress (Cat#HY-13003) and used at 100 nm for 3 h.

## Lentivirus production and transduction
0.3 million HEK-293T cells were seeded in the 6-well plates (Corning). After 24 h, cells were co-transfected with 500 ng of lentiviral plasmids (400 ng of psPAX2 and 150 ng of VSV-G) using TransIT-LT1 Transfection Reagent (Mirus Bio). At 48 and 72 h post-transfection, supernatants containing lentivirus were filtered through a 0.45 µm SFCA syringe filter (CellTreat) and transferred to target cells, together with fresh medium. Transduced cells were selected for 5 days with puromycin (1:3000, 3 µg/ml, Gibco) or 10 days with hygromycin (1:500, 100 µg/ml, Gibco).

## Cell lysis and immunoprecipitation
For anti-V5-IP, cells were transduced with lentivirus produced in 293 cells, using pLX-TRC313-MAEA, pLX-TRC313-MKLN1, pLX-TRC313-ZMYND19, or pLX-TRC313-GFP, as described in the lentivirus production section. Ten million cells were rinsed with PBS and lysed by ice-cold IP lysis buffer (ThermoFisher, #87788) with EDTA-free protease inhibitor (Roche), and cleared by $16,000 \times g$ centrifugation for 10 min at 4 °C in a benchtop microcentrifuge. Lysates were incubated with 30 µL of the pre-cleared beads per 1 mL of lysate for 3 h at 4 °C. Supernatants were then subjected to immunoprecipitation. For V5 pulldowns, cell lysates were incubated with anti-V5-tag magnetic beads (MBL) (30 µL beads/1 ml samples) for 3 h at 4 °C. For anti-Flag-IP, anti-Flag M2 antibody (Sigma, F1804) was incubated with 25 µl Pierce protein A/G magnetic beads per 1 ml of lysate (ThermoFisher, 88803) for 1 h at room temperature, washed with 1 × TBST once and 1 × TBST with 0.5 mM NaCl 3 times. For anti-HA IP, Pierce magnetic anti-HA

beads (ThermoFisher, 88837) were washed three times with lysis buffer, and 1 ml lysate was incubated with 30 µL of the beads for 3 h at 4 °C. Beads were then washed three times, once with IP lysis buffer (ThermoFisher), twice with lysis buffer 30 µl of beads in 1 ml lysate for each sample with 0.5 mM NaCl. To prevent the detection of immunoglobulins, HA peptide (Sigma) elution was performed in lysoIPs shown in Fig. 4f. For anti-GFP IP, GFP-Trap® agarose beads (PGTlabs) were washed three times with lysis buffer, and 1 ml lysate was incubated with 25 µL of beads for 6 h at 4 °C. Beads were washed as described for anti-HA IPs. Washed beads were boiled for 5 min in Laemmli Sample Buffer (Biorad) and subject to SDS/PAGE and immunoblot analysis.

## Growth curve analysis
For growth curve analysis, cells were counted and then normalized to the same starting concentration. Live cell numbers were quantitated at each timepoint by cell counting with the TC20 automatic cell counter (Bio-Rad). Fold change of live cell number at each timepoint was calculated as a ratio of the value divided by the input value.

## CRISPR-Cas9 screen
The Broad Institute Brunello sgRNA lentivirus library was used to generate YCCEL1 libraries. Briefly, 130 million YCCEL1 stably expressing Cas9 were transduced with the Brunello library at a multiplicity of infection of 0.3 by spinoculation in 12-well plates at $300 \times g$ for 2 h in the presence of 4 µg/µl of polybrene. The library contains four distinct sgRNAs of each human gene and multiple control sgRNAs. Cells were incubated at 37 °C with 5% $CO_2$ for 6 h, at which point the EMEM media was exchanged to remove polybrene. 48 h later, puromycin was added to select transduced cells at 3 mg/mL. After 5 days, cells were then cultured in the presence of 0.5 µM alpelisib versus DMSO vehicle control. Cells were passaged every 3 days in fresh alpelisib or DMSO for 14 days. Genomic DNA was then harvested with the Blood and Cell Culture DNA Maxi Kit (Qiagen) from 40 million cells per screen replicate, according to the manufacturer's protocol. PCR-amplified sgRNA abundances were quantified by an Illumina HiSeq sequencer[34] at the Broad Institute. The STARS algorithm was applied to calculate hit statistical significance, using a stringent cutoff of $q < 0.05$ (*p*-value adjusted for the False Discovery Rate)[34].

## Screen validation
sgRNA oligos were obtained from Integrated DNA Technologies and cloned into the pLentiGuide-Puro vector (Addgene plasmid #52963, a gift from Feng Zhang). Lentiviruses were produced in 293T cells by co-transfection of pLentiGuide-puro with psPAX2 and VSV-G. Transduction and puromycin selection were done as described as above. Live cell numbers were quantitated using propidium-iodide stating and measurement by a TC20 automatic cell counter (Bio-Rad). sgRNA sequences used are listed in Supplementary Information.

## Flow cytometry

FACS analysis was performed on a BD FACSCalibur instrument and analyzed by FlowJo V10. For cell cycle analysis, cells were fixed in 70% ethanol overnight at 4 °C, washed twice with PBS, treated with staining buffer (propidium iodide 5 µg/ml, RNase A 40 µg/ml, and 0.1% Triton X-100 in PBS) for 30 min at room temperature. For glucose uptake analysis, live trypsinized cells were incubated with 10 mg/mL 2-NBDG in complete media at 37 °C and tested every 15 min for 2 h.

## Immunoblot analysis

Immunoblot was performed as previously described[98]. In brief, WCLs were prepared with ice-cold RIPA lysis buffer (NaCl, 150 mM; NP40, 1%; DOC, 0.5%; SDS, 0.1%; Tris (pH7.4), 50 mM) microcentrifuged on top speed for 5 min at 4 °C, and boiled with Laemmli Sample Buffer (Biorad). Samples were separated by SDS-PAGE electrophoresis, transferred onto the nitrocellulose membranes, and blocked with 5% milk or 5% BSA in TBST buffer. Membranes were washed three times with TBST and then probed with primary antibodies at 4 °C overnight. Membranes were then washed three times with TBST and incubated with secondary antibody for 1 h at room temperature. Membranes were washed three times with TBST and then developed by incubation with ECL chemiluminescence for 30 s (Millipore). Images were captured by a Licor FC platform. Band intensities were measured with Image Studio Lite Version 5.2. The following antibodies were used in this study. Antibodies against the following were purchased from Cell Signaling Technology: Phospho-Akt (Ser473) (D9E, Cat#4060S), Phospho-Akt (Thr308) (244F9, Cat#4056S), Akt (Cat#9272S), GPX4 (Cat#52455S), Phospho-p70 S6 Kinase (Thr389) (108D2, Cat#9234S), p70 S6 Kinase (Cat#9202S), Phospho-4E-BP1 (Ser65) (Cat#9451S), 4E-BP1 (53H11, Cat#9644S), V5-Tag (D3H8Q, Cat#13202S), Phospho-AMPKα (Thr172) (40H9, Cat#2535S), AMPKα (D5A2, Cat#2532S), Phospho-eIF2α (Ser51, Cat#9721S), eIF2α (D7D3, Cat#5324S), LC3B (D11, Cat#3868S), Raptor (24C12, Cat#2280S), HA-Tag (C29F4, Cat#3724S), GAPDH (D16H11, Cat#5174S), Golgin-97 (D8P2K, Cat#13192S), Calreticulin (D3E6, Cat#12238S), VDAC1 (D73D12, Cat#4661S), Myc-Tag (9B11, Cat#2276S), Myc-Tag (71D10, Cat#2278S), DYKDDDDK Tag (Cat#2368S), NPRL2 (D8K3X, Cat#37344S), FLCN (D14G9, Cat#3697S), mTOR (7C10, Cat#2983S), Phospho-TFEB (Ser211) (E9S8N, Cat37681S), TFEB (D2O7D, Cat#91767S), Phospho-TFE3 (Ser321) (23816S), TFE3 (F3X8T, Cat#19950T), anti-Rabbit IgG HRP-coupled secondary antibody (Cat#7074S), anti-Mouse IgG HRP-coupled secondary antibody (Cat#7076S), anti-Rat IgG HRP-coupled secondary antibody (Cat#7077S). Antibodies against the following were purchased from Proteintech: KRAS (Cat# 12063-1-AP) and USP7 (Cat# 66514-1-Ig). Antibodies against the following were purchased from Biolegend: Beta-actin (Cat#664802) and ATF4 (Cat#693901). Anti-MAEA (Cat#AF7288) antibody was purchased from R&D Systems. Anti-Sheep IgG HRP-coupled secondary antibody (Cat#A3415), anti-puromycin antibody (12D10, Cat#MABE343), Anti-ZMYND19 antibody produced in rabbit (Cat#HPA020642), and anti-GFP (GF28R, Cat# MA5-15256) antibody were purchased from Millipore Sigma. Anti-MKLN1 (C-12, Cat#sc-398956), anti-ZMYND19 (E-4, Cat#sc-398514) and anti-LAMP1 (H4A3, Cat#sc-20011), anti-BZLF1 (BZ1, Cat#sc-53904), anti-BMRF1 (0261, Cat#sc-58121), anti-EBNA1 (1EB12, Cat#sc-81581) antibodies were purchased from Santa Cruz Biotechnology. WDR26 antibody (Cat#A302-245A) was purchased from Fortis Life Science. Anti-LAMP-2 antibody (Cat#H4B4) was purchased from DSHB. Alexa Fluor® 594 AffiniPure Donkey Anti-Mouse IgG (H + L) (Cat#715-585-150) and Alexa Fluor® 488 AffiniPure Goat Anti-Rabbit IgG (H + L) (Cat#111-545-144) were purchased from Jackson ImmunoResearch.

## RNAseq analysis

Total RNA was isolated by the RNeasy Mini kit (Qiagen), following the manufacturer's protocol. Removal of the residual genomic DNA contamination was included in the RNA preparation steps. To construct indexed libraries, 1 mg of total RNA was used to select polyA mRNA using the NEBNext Poly(A) mRNA Magnetic Isolation Module (New England Biolabs), followed by library construction via the NEBNext Ultra RNA Library Prep Kit (New England Biolabs). Three replicates were used for each condition. Libraries were multi-indexed, pooled, and sequenced on an Illumina NextSeq 500 sequencer, using single-end 75 bp reads (Illumina) at the Dana-Farber Molecular Biology core. Adaptor-trimmed Illumina reads for each individual library were mapped to the human GRCh37.83 transcriptome using STAR2.5.2b[99]. Feature Count was used to estimate the number of reads mapped to each contig[100]. Only transcripts with at least 5 cumulative mapping counts were used in this analysis. DESeq2 was used to evaluate differential expression (DE)[101]. Each DE analysis used pairwise comparison between the experimental and control groups. Differentially expressed genes were identified and a p-value < 0.05 cutoff was used. Differentially expressed genes were subjected to Enrichr analysis[102]. Top Enrichr terms and volcano plots were visualized using Graphpad Prism 7.

## MAEA cDNA rescue

MAEA sgRNA PAM site mutations were introduced into the entry vector pENTR223-MAEA (DNASU#HsCD00510985), containing the DNA sequence encoding N-terminally V5-tagged full-length human MAEA. Point mutagenesis was performed using the Q5 site-directed mutagenesis kit (New England Biolabs), designated as pENTR223-MAEA-PAM. Genomic sequences with PAM mutations are underlined, CCAGGAGTACCCGACCCTCAAAG and GCGTTGCGGCTACTACAACACAG. MAEA cDNAs were transferred to the lentiviral destination vector pLX-TRC313 (Broad Institute) using Gateway LR Clonase II Enzyme Mix (Thermo Fisher Scientific). This appended the sequence for a V5-tag onto the MAEA C-terminus. The lentivirus vector pLXTRC313-GFP, which stably expresses a GFP cDNA, was used as a control. YCCEL1 Cas9 cells were transduced with either pLX-TRC313 GFP or MAEA encoding lentiviruses. Cells were selected with hygromycin (100 µg/ml, Gibco) for 2 weeks. Heterogeneous protein expression was confirmed by immunoblot using anti-V5 tag antibody. CRISPR targeting of endogenous MAEA was then performed as described above. Subsequently, cells were selected with puromycin (1:3000, 3 µg/ml, Gibco). sgRNA sequences are listed in the Supplementary Information.

## Targeted metabolite profiling

$3 \times 10^6$ of YCCEL1 expressing control or MAEA sgRNA were seeded into a T75 flask with 15 mL of EMEM with 10% FBS. Two days after seeding, cells were treated with 0.5 µM alpelisib or DMSO for 30 h. $n = 6$ individual samples per group. Cells were collected with scrapers (Corning#353089) and washed with 5 mL of room temperature PBS. Then, pellets were resuspended in 1 mL of dry ice-cold 80% methanol, incubated at −80 °C for 30 min and centrifuged at $21,000 \times g$ for 5 min at 4 °C. Supernatants were collected in pre-chilled tubes and stored at −80 °C. Six replicates were included for each treatment. On the day of analysis, supernatants were incubated on ice for 20 min, clarified by centrifugation at $21,000 \times g$ at 4 °C, and dried down with a speed vacuum concentrator (Savant SPD 1010, Thermofisher Scientific). Samples were re-suspended in 100 µL of 60/40 acetonitrile/water, vortexed, sonicated in ice-cold water, and incubated on ice for 20 min. Following centrifugation at $21,000 \times g$ for 20 min at 4 °C, supernatants were collected for pooled QC. Metabolite profiling was performed at the Beth Israel Deaconess Mass Spectrometry Core. Specifically, samples were re-suspended using 20 µL HPLC grade water for mass spectrometry. 5–7 µL were injected and analyzed using a hybrid 6500 QTRAP triple quadrupole mass spectrometer (AB/SCIEX) coupled to a Prominence UFLC HPLC system (Shimadzu) via selected reaction monitoring (SRM) of a total of 300 endogenous water-soluble metabolites for steady-state analyses of samples. Some metabolites were targeted in both positive and negative ion mode for a total of 311 SRM

transitions using positive/negative ion polarity switching. ESI voltage was +4950 V in positive ion mode and −4500 V in negative ion mode. The dwell time was 3 ms per SRM transition and the total cycle time was 1.55 s. Approximately 9–12 data points were acquired per detected metabolite. For targeted 13 C flux analyses, isotopomers from ~140 polar molecules were targeted with a total of 460 SRM transitions. Samples were delivered to the mass spectrometer via hydrophilic interaction chromatography (HILIC) using a 4.6 mm i.d × 10 cm Amide XBridge column (Waters) at 400 μL/min. Gradients were run starting from 85% buffer B (HPLC grade acetonitrile) to 42% B from 0 to 5 min; 42% B to 0% B from 5 to 16 min; 0% B was held from 16 to 24 min; 0% B to 85% B from 24 to 25 min; 85% B was held for 7 min to re-equilibrate the column. Buffer A was comprised of 20 mM ammonium hydroxide/20 mM ammonium acetate (pH = 9.0) in 95:5 water:acetonitrile. Peak areas from the total ion current for each metabolite SRM transition were integrated using MultiQuant v3.2 software (AB/SCIEX). Metabolites with CV < 30% in pooled QC were used for the statistical analysis. The quality of integration for each metabolite peak was reviewed. Metabolites with $p$-values < 0.05, log2(fold change) > 1 or < −1 were used for pathway analysis using MetaboAnalyst 5.0 (https://www.metaboanalyst.ca/MetaboAnalyst/ModuleView.xhtml). Heatmaps of metabolites in the pathways were generated by feeding $Z$-score values into Morpheus software (https://software.broadinstitute.org/morpheus/).

### Seahorse analysis

YCCEL1 were seeded in 96-well plates at 50,000 per well. After 24 h, each group was treated with DMSO or 0.5 μM alpelisib for 8 h. The sensor cartridge plate was hydrated overnight with XF calibrant. 1.5 μM oligomycin, 0.5 μM FCCP, and 0.5 μM antimycin were added at the indicated timepoints. Oxygen consumption rates and Extracellular acidification rate were measured at each time point by Agilent Seahorse XF96 analyzer using a Seahorse XF96 sensor cartridge. The Seahorse XF Cell Mito Stress Test Kit was purchased from Agilent (Cat#103015-100). Oligomycin, CCCP, and antimycin were used at 1.5, 0.5, and 0.5 μM, respectively for Seahorse analysis.

### Proteomic analysis

$1 \times 10^6$ of YCCEL1 expressing control or MAEA sgRNA were seeded into a T75 flask with 15 mL of EMEM with 10% FBS. Two days after seeding, cells were treated with 0.5 μM alpelisib or DMSO for 30 h. $n$ = 3 individual samples per group. Collected cell pellets were lysed in 400 μl of lysis buffer containing 50 mM Tris pH 7.5, 1% (w/v) SDS, 150 mM NaCl, 1 mM EDTA, supplemented with Roche protease inhibitor cocktail. The lysate was pulse sonicated briefly and centrifuged at $15,000 \times g$ for 15 min at 4 °C. Samples were quantified using the Pierce BCA protein assay kit (Thermo Fisher Scientific, Cat#23225) and 30 μg of each sample was reduced with tris(2-carboxyethyl)phosphine, alkylated with iodoacetamide, and digested with trypsin. The digests were desalted using BioPureSPN C18 spin column (The Nest Group) and labeled with TMT 10 plex isobaric label reagents (Thermo Fisher Scientific, Cat#90110) following the manufacturer's instructions. After labeling, an equal amount of peptides from all samples was pooled together and fractionated into 13 fractions using Pierce high pH reverse-phase peptide fractionation kit (Thermo Fisher Scientific, Cat # 84868). All the fractions were dried in a SpeedVac concentrator and resuspended in 1% formic acid for LC-MS/MS analysis. The fractions were analyzed on a Q Exactive HF mass spectrometer (Thermo Scientific) coupled to an UltiMate 3000 RSLCnano HPLC system (Thermo Scientific). Samples were injected onto a PepMap100 trap column (0.3 × 5 mm packed with 5 μm C18 resin; Thermo Scientific), and peptides were separated by reversed phase HPLC on a BEH C18 nanocapillary analytical column (75 μm i.d. × 25 cm, 1.7 μm particle size; Waters) using a 2.5-h gradient formed by solvent A (0.1% formic acid in water) and solvent B (0.1% formic acid in acetonitrile). Eluted peptides were analyzed by the mass spectrometer in data-dependent mode with

survey scan from 350 to 1800 m/z followed by MS/MS scans at 45,000 resolution on the 15 most abundant ions. The automatic gain control (AGC) targets for MS1 and MS2 were set at 3E6 and 2E5 ions, respectively. The maximum ion injection times for MS1 and MS2 were set at 50 and 120 ms, respectively. The isolation window was 0.7 m/z, normalized collision energy was 30%, first mass was fixed at 110 m/z, and charge-state screening was used to reject unassigned, single, and >7 charged ions. Proteins and peptides were identified and quantified using Proteome Discoverer v2.4 (Thermo Scientific). MS/MS spectra were searched against the SwissProt human and EBV protein databases (June 2021) and consensus identification lists were generated with false discovery rates set at 1% for protein and peptide identifications.

### Protein half-life analysis

Cycloheximide chase analysis was performed as follows. YCCEL1 were seeded in 6-well plates at a density of 300,000 cells/ml. Forty-eight hours later, cells were treated with 50 μg/mL cycloheximide vs DMSO control. Cells were then harvested at the appropriate timepoints, lysed, and subject to immunoblot analysis.

### Puromycin chase analysis

YCCEL1 were seeded into 6-well plates at a density of 300,000 cells/ml. Forty-eight hours later, cells were treated with 10 μg/mL puromycin for 20 min at 37 °C. WCLs were prepared and analyzed by immunoblot, using an anti-puromycin monoclonal antibody to visualize polypeptides newly synthesized during the puromycin pulse.

### In vitro transcription

The IVT DNA fragments were synthesized from IDT, with a T7 promoter (TAATACGACTC ACTATAGG) at 5′ end followed by the coding sequence of interest, followed by the sequence encoding the V5 tag (GGTAAGCCTATCCCTAACCCTCTCCTCGGTCTCGAT TCTACG), followed by a 3′-UTR (GCTCGCTTTCTTGCTGTCCAATTTCTATTAAAGGT T CCTTTGTTCCCTAAGTCCAACTACTAAACTGGGGGATATTATGAAG GGCCTTGAGCATCTGGATTCTGCCTAATAAAAAACATTTATTTTCATT GC) and polyA sequence (AAAAA). To add multiple polyA tails, fragments were amplified by the IVT-R reverse primer and forward primers to the T7 promoter and 5′ sequence of the gene of interest (the sequences for the IVT-MKLN1-F, IVT-ZMYND19-F and IVT-GFP-F primers are listed in the supplement). The template was subject to in vitro transcription, using the mMACHINE™ T7 Transcription kit (Thermofisher, AM1344), as described previously[103]. Briefly, DNA fragments were amplified and the correct length was confirmed by gel electrophoresis. Following clarification by Microspin G-25 columns (Cytiva, #27532501), Products were further cleaned by phenol: chloroform: isoamyl alcohol (25:24:1) extraction. The purified DNA was then precipitated with 0.3 M sodium acetate (pH5.2, Sigma) and 5 μg/ml glycogen (Invitrogen) in 75% ethanol, washed with 70% ethanol, air dried, and dissolved with nuclease-free water. DNA was then used for in vitro transcription reactions, according to manuscript's protocol using 10 μl 2 × NTP/CAP, 0.75 μl GTP, 2 μl 10 × reaction buffer, 1 μg DNA template and 2 μl enzyme mix at 37 °C for 3 h. Reactions were stopped by addition of 2 μl DNase and incubation at 37 °C for 15 min. Finally, mRNAs were purified by phenol:chloroform extraction and isopropanol precipitation.

### mRNA electroporation

The Neon Transfection System (Thermo Fisher) was used to deliver in vitro transcribed mRNA. Based on gene length, 500 ng GFP, 500 ng ZMYND19, or 1500 ng MKLN1 encoding mRNA was mixed with 0.6 million cells in 10 μl Buffer T (Thermo Fisher) per reaction. The electroporation parameters were 1350 v pulse voltage, 20 ms of pulse width and 2 pulses for YCCEL1; 1500 v, 30 ms, and 1 pulse for HEK-293T. Cells were then rapidly returned to the incubator and cultured as described above.

For delivery of RNAs for CRISPR editing, pre-designed crRNA and tracrRNA were ordered from IDT. 0.6 µl of crRNA-tracrRNA (0.12 nmol of crRNA and 0.12 nmol tracrRNA for each reaction) was mixed with 0.6 million cells in 10 µL Buffer T. Cells were subjected to electroporation using the following Neon program (1350 v pulse voltage, 20 ms pulse width, 2 pulses for YCCEL1, and 1500 v, 30 ms, and 1 pulse for HEK-293T) and a 10 µL tip. crRNA sequences are listed in Supplementary Information.

## Live cell imaging

YCCEL1 were plated at a density of 300,000 cells/ml in glass-bottom 35 mm dishes (MATTEK#P35GC). After 48 h, media was exchanged and cells were transfected with cDNA encoding an MAEA-GFP chimera (Sino Biological) using TransIT (Mirusbio#MIR2306), according to manufacturer's protocol. After 24 h, cells were stained with Lysotracker 0.5 µM (Invitrogen#L7528) at 37 °C for 30 min. Then, image acquisition was performed, using a Zeiss LSM 800 microscope with parameters LSM scan speed 8 and frame time 1.86 s. Zeiss Zen Lite (Blue) software was used for image analysis.

## Immunofluorescence

Seeded cells were permeabilized with 0.5% Triton X-100/PBS for 5 min, blocked with 1% BSA/PBS for 1 h at room temperature and incubated with anti-mTOR (1:250) and anti-LAMP2 (1:150) primary antibodies for 1 h at 37 °C. Then, cells were washed three times with TBS and incubated with anti-mouse (1:250) and anti-rabbit (1:250) secondary antibodies for 1 h at 37 °C in the dark. Cells were washed three times with TBS and stained with DAPI (1:5000) for 10 min. Image acquisition was performed by a Zeiss LSM 800 instrument. Image analysis was performed with Zeiss Zen Lite (Blue) software. ImageJ Coloc2 was used to score the colocalization of mTOR and lysosomes. For Supplementary Fig. 12e, cells were stained with an anti-TFEB primary antibody (1:50) followed by an anti-mouse secondary antibody (1:250). Washing and incubation procedures were performed as described above. Three-dimensional Z-stack images were obtained using a Zeiss LSM 800 confocal microscope. Image analysis was performed with Zeiss Zen Lite (Blue) software.

## Lysosomal immunopurification

LysoIP was performed as described previously[72]. Briefly, 35 million YCCEL1 stably expressing HA-tagged TMEM192 were harvested by scrapers on ice, washed twice with ice-cold TBS, and resuspended in 1 ml of ice-cold KPBS (136 mM KCl, 10 mM KH2PO4, pH 7.25). Cell suspensions were centrifuged at 1000 × g for 2 min at 4 °C. Pelleted cells were resuspended in 950 µL of ice-cold KPBS, and 25 µL was reserved for a whole cell fraction. The remaining cells were homogenized on ice with a 2 ml homogenizer. The homogenate was centrifuged at 1000 × g for 2 min at 4 °C. Supernatants were incubated with 100 µl of anti-HA magnetic beads (Pierce/Thermo) for 10 min at 4 °C with rotation. Beads were washed with ice-cold KPBS three times and then eluted using 1× SDS loading buffer for immunoblot analysis.

## Proteinase K protection assay

Proteinase K protection assays were performed as previously described[104,105]. Briefly, YCCEL1 with stable HA-ZMYND19 expression were pelleted and resuspended in KPBS buffer (136 mM KCl, 10 mM KH2PO4, pH 7.25). Cells were Dounce homogenized by 15 strokes and debris was pelleted by microcentrifuge spin at 1000 × g for 2 min at 4 °C. Supernatants were subjected to digestion by proteinase K (10 µg/ml, New England Biolabs, #P8107S) treatment, with or without 1% Triton ×-100 (Sigma-Aldrich, #T8787) to disrupt lysosomal membranes on ice for 15 min. Reactions were stopped by the addition of protease inhibitor cocktail (Sigma-Aldrich, #11873580001) and 2× SDS loading buffer. Samples were then heated at 95 °C for 10 min.

## Structural prediction

Protein structural prediction for ZMYND19, MKLN1, or Raptor was performed with AlphaFold Collab-Multimer[106]. Structure visualization was performed with Polo version 2.5.

## Amino acids starvation and stimulation

YCCEL1 cells were washed with PBS and then starved by incubation in amino acid-free EMEM with 10% dialyzed FBS (Gibco) at 37 °C for 50 min. Amino acid-free EMEM was prepared based on formulation of ATCC 30-2003. Amino acid stimulation was then performed by adding 100× MEM amino acid solution (1:100, ThermoFisher) at 37 °C for 10 min.

## Transmission electron microscopy

A pellet of cells was fixed for at least 2 h at room temperature in fixative (2.5% glutaraldehyde 1.25% paraformaldehyde and 0.03% picric acid in 0.1 M sodium cacodylate buffer (pH 7.4)), washed in 0.1 M cacodylate buffer and post-fixed with 1% osmiumtetroxide (OsO4)/1.5% potassiumferrocyanide (KFeCN6) for 1 h, washed twice in water, once in maleate buffer (MB) and then incubated in 1% uranyl acetate in MB for 1 h, followed by 2 washes in water and subsequent dehydration in grades of alcohol (10 min each; 50%, 70%, 90%, 2 × 10 min 100%). Samples were then put in propyleneoxide for 1 h and infiltrated overnight in a 1:1 mixture of propyleneoxide and TAAB (TAAB Laboratories Equipment Ltd, https://taab.co.uk). The following day, samples were embedded in TAAB Epon and polymerized at 60 °C for 48 h. Ultrathin sections (about 60 nm) were cut on a Reichert Ultracut-S microtome, picked up on to copper grids stained with lead citrate and examined in a JEOL 1200EX Transmission electron microscope or a TecnaiG[2] Spirit BioTWIN and images were recorded with an AMT 2k CCD camera. Quantification of autophagosome-like double membrane structures was done as previously described[62,107]. A total of 25 cells per group were analyzed with ImageJ Region of Interest analysis to quantitate autophagosome-like structures and lipid droplets in control versus MAEA-depleted YCCEL1. The quantities of lipid droplet and autophagosome-like structures were assessed by the ratio of their number to cytoplasm area[107].

## Data visualization

Figures were drawn with GraphPad, Biorender, Powerpoint, Adobe illustrator, and R ggplot2[108]. ImageJ was used to analyze transmission electron microscopy images.

## Reporting summary

Further information on research design is available in the Nature Portfolio Reporting Summary linked to this article.

## Data availability

The RNAseq data generated in this study have been deposited in the GEO database under accession code GSE271761 and GSE271762. The mass spectrometry proteomics data have been deposited in the ProteomeXchange Consortium through the MassIVE repository under the dataset identifier PXD069168. The LC/MS metabolomics data have been deposited in the MassIVE dataset identifier MSV000099428. All plasmids generated for this study will be made available on request, where permissible. CRISPR screen data will be shared upon request. The processed CRISPR screen data, RANseq data, metabolite profiling, proteomic analysis, flow cytometry data are provided in the Supplementary Data files. CRISPR screening data is publicly available through the Sequence Read Archive (SRA) under project ID PRJNA1345411. The live cell data generated in this study are provided in the Supplementary Movie. All oligonucleotide information is provided in Supplementary Data 8. Source data are provided with this paper.

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

## Acknowledgements

This work was supported by P01CA269043, by R01DE033907, and by a Burroughs Wellcome Career Award in Life Sciences to B.E.G. Y.W., Y.S., B.M., and S.W. were supported by American Cancer Society Post-doctoral Fellowships PF-24-1194768-01-TBE, PF-23-1144614-01-IBCD, PF-24-1308318-01-TBE, and PF-24-1250090-01-IBCD, respectively. Y.L. was supported by a Lymphoma Research Foundation Fellowship. R.G was supported by R00 DE031016. We thank Kuang Shen, Naama Kanarek, and Brendan Manning for helpful discussions. We thank Brendan Manning for sharing the GFP-Raptor expression vector. We thank Vamsi Mootha for use of the SeaHorse Flux analyzer. We thank Maria Ericsson and the Harvard Medical School electron microscopy core for technical assistance. RNA-seq datasets have been deposited into the NIH GEO omnibus and will be released upon publication. The mass spectrometry proteomics data have been deposited in the ProteomeXchange Consortium.

## Author contributions

Y.W., I.T., P.M.L., and B.E.G. designed experiments. Y.W., Y.L., Y.S., B.M., B.I.P., S.W., H.Y.T., and J.M.A. performed experiments. Y.W., R.G., and B.E.G. analyzed data. Y.W. and B.E.G. wrote the manuscript.

## Competing interests

The authors declare no competing interests.
