## [Peer Review file · Nature Communications]

The CTLH Ubiquitin Ligase Substrates ZMYND19 and MKLN1 Negatively Regulate mTORC1 at the Lysosomal Membrane

Corresponding Author: Professor Benjamin Gewurz

Version 0:

Reviewer comments:

Reviewer #1

(Remarks to the Author)

EBV-associated gastric cancers (EBVaGCs) often possess an elevated level of PI3K. To identify a novel therapeutic target for such EBVaGCs, the authors performed a human genome-wide CRISPR/Cas9 screen while including the PI3K inhibitor drug alpelisib. Among the candidates they identified were subunits of the E3 ligase CTLH. Knockout of multiple CTLH E3 ligase subunits additively blocked EBVaGC proliferation together with alpelisib. They proceeded to try to identify the precise mechanism for this finding by performing a very large series of very well controlled experiments employing a wide variety of methods. They succeeded in showing that the CTLH substrates ZMYND19 and MKLN1 form a complex that associates with Raptor and RagA/C. In the absence of CTLH activity, ZMYND19 and MKLN1 are stabilized and inhibit mTORC1 at a step distal to its lysosomal membrane recruitment. Thus, they successfully identified a novel EBVaGC therapeutic target whose inhibition may also be synergistic with PI3K blockade in tumors possessing aberrantly elevated PI3K activity. They also identified a previously unknown pathway by which CTLH-dependent activity affects mTORC1 activity.

This study is both quite novel and significant in that the authors have identified a new potential target for therapeutic treatment of patients with cancers containing elevated levels of PI3K. The conclusions are all very well supported by lots of data obtained employing a wide variety of methodologies. The experiments are all very well controlled with sufficient details presented in the Methods sections for the work to be reproduced by others. The authors also state that their datasets will be made available upon publication of this article. In summary, the study reported here is of exceptionally high quality and quantity for the field. This reviewer has a couple of suggestions that could potentially make this work of even greater significance and relevance specifically to the EBV field.

Comments:

1. Given a primary rationale for the research described here was to identify new candidate targets for the development of therapeutics for treating patients with EBV-associated gastric cancer, it would be nice for the authors to show data performed with SNU-719 cells (or another EBV-positive cell line with elevated PI3K) for a few of their key experiments (e.g., Fig. 2C, Fig. 3C), i.e., does sgMAEA + alpelisib also dramatically kill SNU-719 or the EBV-NPC cell line NPC43? In other words, can their key finding be generalized?
2. Likewise, it would be nice to present a little bit of data showing the effect of sgMAEA + alpelisib on the expression of some EBV-encoded genes, e.g., does sgMAEA + alpelisib enhance EBV reactivation leading to cell death?

Minor comments:

1. Please clarify what is being referred to as the "extended data tables".
2. Page 18, line 479 – Should "as yet identified" be "as-yet-unidentified"?

Reviewer #2

(Remarks to the Author)

The aim of study was to identify synthetically lethal mutations by CRISPR screens in Epstein-Barr virus-associated gastric carcinoma (EBVaGC) treated with PI3K inhibitor alpelisib. The authors looked for target whose inhibition was synthetic lethal with alpelisib. Among other candidates, they identified members of the CTLH complex, an E3 ubiquitin ligase complex as depleted in the surviving cells.

They used YCCEL1 (EBV GC) cell line for CRISPR screen because they previously characterized PI3KCA kinase domain gain of function mutation that increases PI3K signaling. Their screen for synthetic lethality identified 3 components of the CTLH complex, MAEA, WDR26 and YPEL5. The authors went on to confirm that MAEA downregulation promoted cell death in conjunction with alpelisib. They then identify proteins upregulated upon MAEA knockout and identified several candidate proteins. They focused on 2 proteins that are known targets of the CTLH complex, its own subunit muskelin (MKLN1), and ZMYND19, a protein not well characterized but previously identified as target of the CTLH complex. They showed that their combined increased levels inhibit mTORC1, therefore generating the synthetic lethality in combination with alpelisib. They characterize the domains important for their inhibitory action on mTORC1 and identified an interaction of ZMYND19 with Raptor at the lysosomal membrane. They conclude that MKLN1 and ZMYND19 function to block mTORC1 activation when their levels are increased through MAEA knockout. Their effect on mTORC1 inhibition is not dependent on TSC2 and appears to be at a late stage of activation after recruitment to the lysosomal membrane. However the precise mechanism of action remains unclear.

Overall, this is an interesting study proving novel data that uncovers a new pathway of mTORC1 inhibition. The experiments are overall well-conducted and well controlled and the conclusions drawn from the data are generally sound. Some exceptions, or shortcomings are noted below.

One main weakness is that they do not identify the mechanism of mTORC1 inhibition by the MKLN1 and ZMYND19 proteins. There are also several elements that are missing in the interpretation of the data, and much has to do with the complexity of the CTLH complex, which functions and mechanism of action are still poorly understood. The authors go back and forwards in their experiments and in the narrative between the effect of MAEA knockout and the upregulation of ZMYND19 and MKLN1 which are interdependent, however the overall implication of the CTLH complex in the negative mTOR regulation is unclear. They conclude that loss of CTLH activity results in increased expression of ZMYND19 and MKLN1 and that these 2 products function to inhibit mTORC1.

Specific points to address:

There are a number of issues with the references. Some references are either missing or incorrect. For instance:

Line 83: "While CTLH negatively regulates gluconeogenesis in yeast²²⁻²⁴, this metabolism role has not been conserved in higher organisms..." that statement needs to be referenced. Similarly, a second statement in line 140-141 needs to be referenced (This metabolic role does not appear to have been conserved in higher organism CTLH complexes)

Other issues with referencing are included in the comments below.

Experimentally

1) Fig. 1 : WDR26 identified with MAEA as high confidence candidate whose depletion is synthetic lethal with alpelisib. Did the author test whether WDR26 KO also caused synthetic lethality in follow-up experiments? It is strange that WDR26 was found in the screen, because WDR26 KO was reported in a previous study to reduce the overall levels of MKLN1, through a decrease of nuclear MKLN1. The cytoplasmic levels appear to be unchanged upon WDR26 knockout, so how could the loss of WDR26 promote the effect observed through the reported MAEA knockout? The authors should investigate whether WDR26 loss also results in synthetic lethality.

2) Fig 2: The authors perform evaluation of changes in cellular metabolism upon MAEA knockout. They fail to mention that a previous study evaluated the effect of downregulating/knocking out CTLH complex activity on metabolism (Maitland et al., 2021 FASEB – cited in the manuscript as ref 53). The authors should make reference to this study and comment on whether their results are similar, or different.

3) Fig. 3 identification of MAEA/CTLH substrates involved in mTOR regulation.

They identify targets of the CTLH complex affected by MAEA knockout. They fail to indicate that MKLN1 and ZMYND19 have previously been identified as targets of the CTLH complex (Maitland et al., 2019, and Mohamed et al., 2021). This reviewer understands that the authors wanted to employ an unbiased approach to identify targets that would be responsible for the synthetic lethality, however, these 2 proteins are known targets of the CTLH complex, so it should be acknowledged that they are not novel targets and have been identified in previous work.

The authors determine that ZMYND19 and MKLN1 are necessary for cells death triggered by MAEA KO plus alpelisib, as KO of both ZMYND19 and MKLN1 rescued cell survival. But what they look at in Fig 3j is unclear. They electroporate ZMYND19 and MKLN1 mRNAs and conclude that both inhibit mTOR (through measuring phosphoS6).

There are several questions/issues regarding panel 3j:

a. P10: line 256-7: the sentence needs to be corrected : Furthermore, electroporation of in vitro transcribed (IVT) ZMYND19 mRNA (used to overcome its proteasomal turnover) decreased mTOR target S6 T389 phosphorylation levels,-this should be changed to "had no effect? - ...whereas combined ZMYND19 and MKLN1 mRNA electroporation decreased S6 T389 phosphorylation (Fig. 3j, Extended data Fig. S3n).

b. What cells are they using? WT cells or MAEA knockout? That is very important to state, since they concluded that ZMYND19 and MKLN1 are necessary for cells death triggered by MAEA KO in panel 3i.

c. Fig 3j: effects on S6 phosphorylation are not quantified – despite statement saying n=3 for all blots. This should be done.

d. Also related to this figure: in Fig S3j – text on line 243, the authors mention that "...MKLN1 depletion also increased steady-state ZMYND19 levels, suggesting that it supports ZMYND19 turnover". This is quite confusing, and the authors don't elaborate on this, but does this suggest that ZMYND19 is a target of muskelin when the CTLH complex is active (in WT cells). Their model in Fig. 5j is not consistent with that, this needs to be clarified.

e. Overall, these data indicate that the effects of MKLN1 and ZMYND19 are independent of the CTLH complex activity. Is that what the authors conclude? Because the model in Fig 5j does not take that into account. Could the authors overexpress MKLN1 and ZMYND19 in WT cells and determine whether this synergizes with alpelisib to determine whether it compromises cell proliferation as in Fig 3i?

4) Fig 4 localization of muskelin and ZMYND19 at lysosomes

- a. Localization of MKLN1 to the lysosome was also previously reported. Heisler et al., Neuron 2011, PMID: 21482357 ; Heisler et al., Neuron 2018 PMID: 30174115. These papers should be cited.
- b. Line 285: "Taken together, these data suggest that ZMYND19 recruits MKLN1 to lysosomal outer membrane regions." Fig 4f: the diagram indicates CTLH complex inhibits ZMYND19 and MKLN1 localization at the lysosomal membrane: it is unclear where does this conclusion come from? Could the authors clarify how they reached that conclusion?
- c. They also make the claim that ZMYND19 recruits CTLH to lysosomes but there is no data to back up the claim up. Why not do the LysoIP in ZMYND19 knockdown cells and see if it changes?

Fig 5: ZMYND19 and MKLN1 domains important for mTOR inhibition

a. Fig 5 a and b could be put in the supplemental materials as this is data that is already known from previous publications (Mohamed et al, EMBO J 2021: Sherpa et al., 2022.)

b. Is the effect of MKLN1-ZMYND19 dependent on the MAEA knockout? Meaning, can transfections, or overexpression of the 2 proteins on their own affect mTORC1 activation, or is that

c. The blots shown in Fig 5 e, g and i are not quantified and they should be quantified to accurately assess the effects of MKLN1 and ZMYND19. In Fig 5e, it is impossible to tell what is happening with the levels of P-S6 since total S6 levels are different from one lane to the other. It is understood that it is difficult to quantify since levels of expression of the mutants are varying, but it seems that MKLN1 mutant K1-6 and K6 are actually activating S6 phosphorylation rather than failing to block mTOR.

6) Fig 6: ZMYND19 associates with RAPTOR and RagA/C

The authors conclude that the ZMYND19+muskelin complex interacts with the inactive RagA/C complex .Conclusion is reached by coIPing FLAG out of cells cotransfected with FLAG-muskelin and FLAG-ZMYND19, along with MYC-Raptor and HA-Rag proteins.

a. line 351-2: the authors claim that "the ZMYND19 zinc finger supports association with MKLN1, Raptor and CTLH." There is no indication in these data that the other CTLH complex subunits also are part of that complex. The authors should test either MAEA or RanBP9, or WDR26 (or others), to prove that the complex is also implicated.

b. They should perform this CoIP in cells depleted of muskelin to determine whether muskelin is important for this interaction to occur

Globally, the fundamental observation that loss of MAEA synergistically blocks cell proliferation with alpelisib is compelling and suggests yet unappreciated role for the CTLH complex in mTOR regulation. However the data does not lead to a comprehensive model of how mTOR is inhibited and data interpretation is made difficult by the lack of quantification of key data, and by the apparent confusion regarding the mechanism of action of the CTLH complex. The authors sometimes mention MKLN1 as a target and sometimes as a member of the CTLH complex. MKLN1 is both a complex member and a substrate as described in previous publications (Maitland et al., 2019, Lampert et al, 2018, Sherpa et al., 2021), so their model should take that into account. Also, their data suggest that ZMYND19 may actually be a target of the MKLN1-CTLH complex since its levels are upregulated in MKLN1 KO cells, and while this is acknowledged, it is not taken into consideration in their model. Therefore, models presented in 4f and 5j are inaccurate. The authors actually propose that ZMYND19 is a lysosomal protein that is targeted by the CTLH complex through a MKLN1 interaction in their discussion, but their experimental strategy does not reflect that. If this is correct, then they should (as suggested above) repeat some of their experiments in Fig. 6 in MKLN1 KO cells to determine the contribution of MKLN1 to the mTOR regulation by ZMYND19.

Reviewer #3

(Remarks to the Author)

General Comments

Epstein-Barr virus-associated gastric carcinoma (EBVaGC) is primarily associated with increased activation of the PI3K signaling pathway, mostly due to activating mutations occurring in its catalytic subunit p110 α . Consequently, treatment with the PI3K p110 α inhibitor alpelisib significantly reduces the growth of these cancer cells.

In this manuscript, Wang et al. performed a genome-wide CRISPR/Cas9 screen to identify genes that synergistically inhibit the proliferation of EBVaGC cells when combined with alpelisib. They identified several subunits of the C-terminal to LisH (CTLH) E3 ligase, including the catalytic MAEA subunit. The authors propose that depletion of CTLH subunits, particularly MAEA, results in increased levels of its substrates ZMYND19 and MKLN1, the latter also being a component of the CTLH complex. These two proteins, in turn, are proposed to inhibit mTORC1 signaling somehow by associating with Raptor and Rag GTPases at the lysosome.

While this study presents potentially interesting findings, it lacks an explanation of the mechanism underlying mTORC1 downregulation and the physiological relevance of this observation. The data shown in the manuscript suggest that increased levels of ZMYND19 and MKLN1 do not affect mTOR lysosomal recruitment or its activation by Rheb. In the discussion section, the authors hypothesize that these two proteins could impair the ability of Raptor to associate with TOS-containing mTORC1 substrates, such as S6K and 4E-BP1, or preclude the association between mTORC1 and Rheb. Both hypotheses need to be tested to dissect the mechanism underlying ZMYND19-MKLN1-dependent inhibition of mTORC1 signaling. Since transcription factors EB and E3 are mTORC1 substrates that do not contain a TOS motif (PMID: 32612235), analyzing their phosphorylation state as well as their nuclear/cytosolic localization can be useful to test the first hypothesis. However, both hypotheses should be exhaustively tested through biochemical and imaging analyses.

Moreover, the relevance of these findings needs to be demonstrated in the context of physiological regulation of mTORC1 activity, particularly in response to nutrient availability. Notably, a very recent paper (PMID: 38788716) demonstrates that mTORC1 inhibition promotes the degradation of the HMGCS1 enzyme via the CTLH complex, thus impacting cell proliferation by affecting the mevalonate pathway. Importantly, they show that deletion of CTLH E3 activity, through MAEA depletion, did not alter mTOR signaling or autophagy in HEK293T cells, which contrasts with the main findings presented in this manuscript. Therefore, the evidence that CTLH genetic inactivation is responsible for the downregulation of mTORC1 signaling needs further corroboration.

Throughout the manuscript, mTORC1 signaling is monitored by analyzing the phosphorylation levels of P-S6K only (or P-S6, which is not clear—see specific comments). The authors should monitor and quantify the phosphorylation levels of both P-S6K (at Thr389) and P-4E-BP1 (at Ser-65) relative to the corresponding total proteins, in both fed and starved conditions, with torin-1 treatment added as an internal control. However, the relevance of mTORC1 inhibition in the context of EBVaGC is not demonstrated, as treatment with rapamycin seems to have a subtle effect on the growth of these cells (Fig. S3F), suggesting that other functions downstream of the CTLH complex could impair the growth of these cancer cells rather than the regulation of the mTORC1 pathway.

Therefore, as it stands, the study does not support the claims and conclusions made by the authors.

Specific Comments

1. Fig. 2C: Provide more detailed information in the figure legend, including sample comparisons and statistical methods applied.
2. Fig. 3B: Clarify whether the authors are monitoring P-S6K phosphorylation or P-S6 phosphorylation. The panel indicates P-S6 (T389), but S6 phosphorylation is typically monitored at Ser235/236 or Ser240/244, whereas S6K phosphorylation is monitored at Thr389. If the Western blot refers to P-S6K and pan-S6K, this needs to be corrected. The same error recurs in all figures monitoring mTORC1 signaling (Figs. 3C, 3J, 7E, 7G, S3E, S3N, S6C, S6D, S7A). It should be correctly indicated whether P-S6K phosphorylation or P-S6 phosphorylation is being monitored. Additionally, S6 phosphorylation is not a good readout of mTORC1 activity since it is an indirect substrate of mTOR (phosphorylated by S6K). For all these panels, the authors should provide analysis and quantification of phosphorylation levels of both S6K and 4E-BP1 relative to corresponding total proteins. Both fed and starved conditions should be analyzed, with torin-1 treatment as an internal control for mTORC1 signaling inhibition.
3. Fig. S3F: The four conditions shown (sgControl and sgMAEA in DMSO or rapamycin) should be compared together and analyzed through ANOVA (Student's t-test cannot be used when comparing more than two samples or conditions). However, it seems that rapamycin treatment does not significantly impact cell survival, suggesting that mTORC1 signaling does not significantly support the growth of these gastric tumor cells. These results do not align with the manuscript's main claim.
4. Fig. 3D: Quantification of lipid droplets should be provided to support the claim that their number significantly increases upon MAEA depletion. Since autophagy activation would promote lipid droplet clearance (PMID: 19339967) rather than accumulation, this observation contradicts the authors' claim that MAEA depletion induces mTORC1 downregulation and hence autophagy activation. To assess autophagy activation, the authors should check levels of lipidated LC3 (by WB) and LC3 puncta (by IF) upon MAEA depletion.
5. Fig. 4D, 4E: To prove that the proteins of interest are indeed specifically associated with lysosomes, the authors should also monitor markers for other cellular compartments in the same WB (ER, Golgi, mitochondria, similar to what is shown in PMID: 29074583) to demonstrate the purity of their lysosomal fractions.
6. Fig. 4F: To prove that ZMYND19 and MKLN1 associate with the outer (cytoplasm-facing) leaflet of the lysosome, the authors should treat affinity-captured lysosomes with increasing concentrations of proteinase K and show that these two proteins are sensitive to proteinase K digestion, contrary to luminal proteins (e.g., LC3, hydrolases) that will be protected. Additionally, the authors should analyze lysosomal localization of endogenous MKLN1 and ZMYND19 by imaging analysis in response to MAEA depletion and mTORC1 activity (nutrient/starvation conditions). Total levels of these proteins should also be checked by WB in response to nutrient availability.
7. Fig. 6C and Fig. S6B: The authors report that both figures indicate the zinc finger of ZMYND19 is important for binding with Raptor, but Fig. S6B demonstrates the opposite! The two figures are not consistent: the zinc finger of ZMYND19 seems not to be important for binding with Raptor in 293T cells but is important in YCCEL1 cells. Quantification should be reported for these data, and the authors should reconcile this discrepancy.
8. Fig. 6E: The data are insufficient to support the claim that ZMYND19/MKLN1 preferentially associate with inactive Rags. The slight difference observed in the WB seems to align with levels of expression of RagC constructs. This result needs to be validated through quantification of IP and imaging analysis. If the authors propose that this association occurs during starvation, this needs to be demonstrated in fed vs. starved cells and following endogenous Rags.
9. Fig. 7G: Authors should test also starved versus fed condition and test whether Rheb overexpression shows the same phenotype.

Comments on the Text

1. Row 63: RagA and RagB are functionally interchangeable, as are RagC and RagD, hence four heterodimer configurations are possible: RagA/RagC, RagA/RagD, RagB/RagC, and RagB/RagD. Their activation state depends only on amino acid loading. Please correct the sentence.
2. Row 149: SESN2 is not a component of the GATOR1 complex but a regulator of its activity (it binds and inhibits GATOR2, which inhibits GATOR1). Please correct this.
3. Row 152: The complex name is Regulator, not regulator. Please correct this.

Version 1:

Reviewer comments:

Reviewer #1

(Remarks to the Author)

The authors have now performed a large number of additional experiments in response to each of the comments, suggestions, and concerns expressed by the three reviewers. They have also now thoroughly quantified their data and made numerous improvements to the text, methods, and reference sections in response to reviewers' comments. This reviewer believes the authors have now adequately addressed essentially all of the concerns of the reviewers.

Reviewer #2

(Remarks to the Author)

The authors have satisfactorily addressed my concerns and I support publication of the manuscript.

Reviewer #3

(Remarks to the Author)

In this revised version of the manuscript entitled "The CTLH1 Ubiquitin Ligase Substrates ZMYND19 and MKLN1 Negatively Regulate mTORC1 at the Lysosomal Membrane," Wang et al. present additional experiments supporting the claim that ZMYND19 and MKLN1 act in concert to inhibit mTORC1 signaling, thereby activating the autophagy pathway. However, the manuscript does not provide further insight into the mechanism underlying the ZMYND19-MKLN1-dependent inhibition of mTORC1 signaling. The data suggest that these CTLH substrates do not regulate RagGTPases activity, as MAEA depletion still leads to mTORC1 inhibition in cells overexpressing a lysosomal-anchored Raptor (Raptor-Rheb) construct or in cells KO for GATOR1. Moreover, the TSC axis does not appear to be involved, as neither TSC depletion nor Rheb overexpression affects the observed phenotype.

In the discussion section, the authors propose the interesting hypotheses that ZMYND19/MKLN1 may prevent Raptor binding to the TOS motif found in 4EBP1, S6K1, and eventually additional mTORC1 regulators, or alternatively, ZMYND19/MKLN1 may prevent the association between Regulator-bound mTORC1 and Rheb. These hypotheses require experimental validation to fully elucidate the mechanism through which CTLH substrates regulate mTORC1 signaling. I also suggested to test the activation state of TFEB which behave as a peculiar mTORC1 substrate dependent on FLCN axis but not data were provided on this. Furthermore, as noted in my previous review, the manuscript lacks information regarding the physiological relevance of this mechanism. The authors should address whether they intend to propose that CTLH activity is reduced during nutrient (particularly amino acid) starvation, and whether increased levels of ZMYND19/MKLN1 serve to inhibit mTORC1 signaling under such conditions.

As it stands, this manuscript is missing key experimental evidence and mechanistic details, and therefore lacks the substantiation needed for acceptance for publication.

Point-by-point response to the reviewers' comments.

We thank each of the reviewers for the time they spent reviewing the manuscript and for their constructive and helpful comments, each of which is addressed below.

Reviewer #1 (Remarks to the Author):

EBV-associated gastric cancers (EBVaGCs) often possess an elevated level of PI3K. To identify a novel therapeutic target for such EBVaGCs, the authors performed a human genome-wide CRISPR/Cas9 screen while including the PI3K inhibitor drug alpelisib. Among the candidates they identified were subunits of the E3 ligase CTLH. Knockout of multiple CTLH E3 ligase subunits additively blocked EBVaGC proliferation together with alpelisib. They proceeded to try to identify the precise mechanism for this finding by performing a very large series of very well controlled experiments employing a wide variety of methods. They succeeded in showing that the CTLH substrates ZMYND19 and MKLN1 form a complex that associates with Raptor and RagA/C. In the absence of CTLH activity, ZMYND19 and MKLN1 are stabilized and inhibit mTORC1 at a step distal to its lysosomal membrane recruitment. Thus, they successfully identified a novel EBVaGC therapeutic target whose inhibition may also be synergistic with PI3K blockade in tumors possessing aberrantly elevated PI3K activity. They also identified a previously unknown pathway by which CTLH-dependent activity affects mTORC1 activity.

This study is both quite novel and significant in that the authors have identified a new potential target for therapeutic treatment of patients with cancers containing elevated levels of PI3K. The conclusions are all very well supported by lots of data obtained employing a wide variety of methodologies. The experiments are all very well controlled with sufficient details presented in the Methods sections for the work to be reproduced by others. The authors also state that their datasets will be made available upon publication of this article. In summary, the study reported here is of exceptionally high quality and quantity for the field. This reviewer has a couple of suggestions that could potentially make this work of even greater significance and relevance specifically to the EBV field.

>>thank you for this excellent summary and for the positive comments.

Comments:

1. Given a primary rationale for the research described here was to identify new candidate targets for the development of therapeutics for treating patients with EBV-associated gastric cancer, it would be nice for the authors to show data performed with SNU-719 cells (or another EBV-positive cell line with elevated PI3K) for a few of their key experiments (e.g., Fig. 2C, Fig. 3C), i.e., does sgMAEA + alpelisib also dramatically kill SNU-719 or the EBV-NPC cell line NPC43? In other words, can their key finding be generalized?

>>Thanks for these suggestions. We now provide data in Supplementary Fig. 2A-B that MAEA depletion and alpelisib additively block the survival of SNU-719 EBV+ gastric carcinoma cells.

We show in Fig 3B that MAEA KO blocks S6K and 4E-BP1 phosphorylation in SNU-719. YCCEL1 and SNU-719 are the only gastric carcinoma cell lines from EBV+ tumors currently available.

We also provide data in Supplementary 4D that MAEA depletion impairs mTOR activation in SNU-719, as judged by reduction in S6K, 4EBP1 phosphorylation and AKT phosphorylation, and that MAEA depletion together with alpelisib more strongly blocks each of their phosphorylation

2. Likewise, it would be nice to present a little bit of data showing the effect of sgMAEA + alpelisib on the expression of some EBV-encoded genes, e.g., does sgMAEA + alpelisib enhance EBV reactivation leading to cell death?

>>Thanks for this suggestion. We now provide new data to address this point in Supplemental Fig. S2g-i and S4I, shown below. We did not observe induction of the immediate early BZLF1 or early BMRF1 proteins in either YCCEL1 or SNU719 upon MAEA editing, with or without alpelisib.

We have also performed RNAseq analysis of YCCEL1 control versus MAEA KO or MAEA KO + alpelisib, where we found nearly no significantly changed EBV gene expression at a $p < 0.05$ cutoff.

Minor comments:

1. Please clarify what is being referred to as the “Supplementary tables”.

>>thanks for pointing this out. We now refer to each of the tables by *Nature Communications* format. We have five tables which are presented in the supplementary material and labeled what each of the Supplementary tables are. They are now referred to as Supplementary Data 1-5. We also have 1 supplementary movie, referred to as Supplementary Movie 1.

2. Page 18, line 479 – Should “as yet identified” be “as-yet-unidentified”?

>>thanks very much for catching this, we made the suggested change.

Reviewer #2 (Remarks to the Author):

The aim of study was to identify synthetically lethal mutations by CRISPR screens in Epstein-Barr virus-associated gastric carcinoma (EBVaGC) treated with PI3K inhibitor alpelisib. The authors looked for target whose inhibition was synthetic lethal with alpelisib. Among other candidates, they identified members of the CTLH complex, an E3 ubiquitin ligase complex as depleted in the surviving cells.

They used YCCEL1 (EBV GC) cell line for CRISPR screen because they previously characterized PI3KCA kinase domain gain of function mutation that increases PI3K signaling. Their screen for synthetic lethality identified 3 components of the CTLH complex, MAEA, WDR26 and YPEL5. The authors went on to confirm that MAEA downregulation promoted cell death in conjunction with alpelisib. They then identify proteins upregulated upon MAEA knockout and identified several candidate proteins. They focused on 2 proteins that are known targets of the CTLH complex, its own subunit muskelin (MKLN1), and ZMYND19, a protein not well characterized but previously identified as target of the CTLH complex. They showed that their combined increased levels inhibit mTORC1, therefore generating the synthetic lethality in combination with alpelisib. They characterize the domains important for their inhibitory action on mTORC1 and identified an interaction of ZMYND19 with Raptor at the lysosomal membrane. They conclude that MKLN1 and ZMYND19 function to block mTORC1 activation when their levels are increased through MAEA knockout. Their effect on mTORC1 inhibition is not dependent on TSC2 and appears to be at a late stage of activation after recruitment to the lysosomal membrane. However the precise mechanism of action remains unclear.

Overall, this is an interesting study proving novel data that uncovers a new pathway of mTORC1 inhibition. The experiments are overall well-conducted and well controlled and the conclusions drawn from the data are generally sound. Some exceptions, or shortcomings are noted below.

One main weakness is that they do not identify the mechanism of mTORC1 inhibition by the MKLN1 and ZMYND19 proteins. There are also several elements that are missing in the interpretation of the data, and much has to do with the complexity of the CTLH complex, which functions and mechanism of action are still poorly understood. The authors go back and forwards in their experiments and in the narrative between the effect of MAEA knockout and the upregulation of ZMYND19 and MKLN1 which are interdependent, however the overall implication of the CTLH complex in the negative mTOR regulation is unclear. They conclude that loss of CTLH activity results in increased expression of ZMYND19 and MKLN1 and that these 2 products function to inhibit mTORC1.

>> Thank you for this excellent summary and constructive comments. Specific points are addressed below.

Specific points to address:

There are a number of issues with the references. Some references are either missing or incorrect. For instance:

Line 83: "While CTLH negatively regulates gluconeogenesis in yeast²²⁻²⁴, this metabolism role has not been conserved in higher organisms..." that statement needs to be referenced. Similarly, a second statement in line 140-141 needs to be referenced (This metabolic role does not appear to have been conserved in higher organism CTLH complexes). Other issues with referencing are included in the comments below.

>>We appreciate you for pointing this out. We added references, including here and in the other places discussed in the comments below. Many of these references were cited in our originally submitted manuscript, but we cite them and others again in these specific places.

Experimentally

1) Fig. 1 : WDR26 identified with MAEA as high confidence candidate whose depletion is synthetic lethal with alpelisib. Did the author test whether WDR26 KO also caused synthetic lethality in follow-up experiments? It is strange that WDR26 was found in the screen, because WDR26 KO was reported in a previous study to reduce the overall levels of MKLN1, through a decrease of nuclear MKLN1. The cytoplasmic levels appear to be unchanged upon WDR26 knockout, so how could the loss of WDR26 promote the effect observed through the reported MAEA knockout? The authors should investigate whether WDR26 loss also results in synthetic lethality.

>>Thanks for these suggestions and opportunity to clarify these points. We now provide data in Supplementary Fig. 7 that WDR26 depletion is indeed synthetic lethal with alpelisib, in agreement with the CRISPR screen result. We note that WDR26 did score somewhat less strongly than MAEA (Fig. 1e), and consistent with this result, the effects of WDR26 KO + alpelisib were not quite as strong as with MAEA KO + alpelisib, but they were nonetheless significant. To investigate this result further, we now also provide data in Supplementary Fig. 7b to show that WDR26 KO decreased MKLN1 steady state levels somewhat, consistent with the prior publication that WDR26 is important for the assembly of a nuclear CTLH complex that targets nuclear (but not cytoplasmic) MKLN1 (PMID 35833506). We now cite

this prior publication. Consistent with this result, WDR26 KO did not increase ZMYND19, which we suggest to be localized in a peri-lysosomal subcellular region. Consistent with this result, we now present in Supplementary Fig. 7c that WDR26 KO did not substantially impair mTOR activity. Taken together, these data suggest that WDR26 does synthetically block YCCEL1 survival together with PI3K inhibition, likely as part of a distinct nuclear CTLH complex, but given space constraints feel it is beyond the scope of this manuscript to work out this additional mechanism. We now added discussion about this to the manuscript discussion section and cited additional literature.

2) Fig 2: The authors perform evaluation of changes in cellular metabolism upon MAEA knockout. They fail to mention that a previous study evaluated the effect of downregulating/knocking out CTLH complex activity on metabolism (Maitland et al., 2021 FASEB – cited in the manuscript as ref 53). The authors should make reference to this study and comment on whether their results are similar, or different.

>>Thank you for this point. We now reference and discuss the Maitland *et al.* study characterization of CTLH KO on metabolism, in the results section and particularly in the discussion section.

3) Fig. 3 identification of MAEA/CTLH substrates involved in mTOR regulation.

They identify targets of the CTLH complex affected by MAEA knockout. They fail to indicate that MKLN1 and ZMYND19 have previously been identified as targets of the CTLH complex (Maitland et al., 2019, and Mohamed et al., 2021). This reviewer understands that the authors wanted to employ an unbiased approach to identify targets that would be responsible for the synthetic lethality, however, these 2 proteins are known targets of the CTLH complex, so it should be acknowledged that they are not novel targets and have been identified in previous work.

>>Indeed, we did not wish to claim these to be newly identify these as CTLH substrates and were well aware of prior literature that implicated them as CTLH substrates, which we therefore cited when they first came up in the original submission (“Consistent with prior studies^{53, 54}, the abundances of MKLN1 and ZMYND19 were amongst the most significantly increased by CTLH perturbation”. Reference 53 was a

study by Maitland et al published in 2021 and reference 54 is the Mohamed et al study. We now also cite the Maitland et al. 2019 study as well. We are very careful in this resubmitted version to point out prior findings in this area.

The authors determine that ZMYND19 and MKLN1 are necessary for cells death triggered by MAEA KO plus alpelisib, as KO of both ZMYND19 and MKLN1 rescued cell survival. But what they look at in Fig 3j is unclear. They electroporate ZMYND19 and MKLN1 mRNAs and conclude that both inhibit mTOR (through measuring phosphoS6).

>>Thank you for the opportunity to clarify this point. In panel 3j (now 3h), we tested whether ZMYND19 and/or MKLN1 over-expression are sufficient to block mTOR activity, when over-expressed alone or together. To do so, we established single cell MKLN1 and ZYMDN19 KO 293T cells. We then used electroporation of *in vitro* transcribed mRNA to provide a burst of MKLN1 and/or ZMYND19 expression. By using a 293T single cell knockout clone that lacks MKLN1 and ZMYND19, we could cleanly test the effect of overexpression of either or both. This also enabled us to ask whether either could block mTOR activity without the other. We used electroporation of *in vitro* transcribed mRNA as this approach provides a burst of expression and overcomes the short protein half-lives. Fig. 3h provides a key line of evidence that MKLN1/ZMYND19 co-expression is sufficient to decrease mTOR activity, as read out by phospho-S6K and phospho-4E-BP1 levels. By contrast, overexpression of either ZMYND19 or MKLN1 alone could not decrease mTOR activity by these measures.

There are several questions/issues regarding panel 3j:

a. P10: line 256-7: the sentence needs to be corrected : Furthermore, electroporation of *in vitro* transcribed (IVT) ZMYND19 mRNA (used to overcome its proteasomal turnover) decreased mTOR target S6 T389 phosphorylation levels,-this should be changed to “had no effect? -whereas combined ZMYND19 and MKLN1 mRNA electroporation decreased S6 T389 phosphorylation (Fig. 3j, Supplementary Fig. 3n).

>>We amended the text as suggested.

b. What cells are they using? WT cells or MAEA knockout? That is very important to state, since they concluded that ZMYND19 and MKLN1 are necessary for cells death triggered by MAEA KO in panel 3i.

>> In original panel 3i, now 3g, we used Cas9+ YCCEL1 which expressed a series of sgRNAs as labeled below the graph. The extent of depletion is shown in the immunoblot. We now labeled that clearly on the figure, and in both the original and resubmission it is in the figure legend.

c. Fig 3j: effects on S6 phosphorylation are not quantified – despite statement saying n=3 for all blots. This should be done.

>>We now provide quantitation of S6K and 4E-BP1 versus total cell levels of each in original panel 3j (now 3h) and throughout.

d. Also related to this figure: in Fig S3j – text on line 243, the authors mention that “...MKLN1 depletion also increased steady-state ZMYND19 levels, suggesting that it supports ZMYND19 turnover”. This is quite confusing, and the authors don’t elaborate on this, but does this suggest that ZMYND19 is a target of muskelin when the CTLH complex is active (in WT cells). Their model in Fig. 5j is not consistent with that, this needs to be clarified.

>>We apologize for the confusion. We found that MKLN1 KO increased steady state ZMYND19 levels. For instance, we now show in Supplementary Fig. 5e the following, with quantitation, where mkln1 KO increased zmynd19 steady state levels by 2 fold.

Supplementary Fig. 6e

We also found that ZMYND19 steady state levels were higher when we reconstituted ZMYND19 expression alone in ZMYND19/MKLN1 double KO 293T cells, versus when we reconstituted both ZMYND19 and MKLN1 (please compare lanes 2 and 4 in Fig 3h below). We also added references to support the prior observation that MKLN1 is both a component of and substrate of CTLH.

Fig. 3h

As suggested, we amended the Figure 5 model (now Fig. 5h) to better depict this.

e. Overall, these data indicate that the effects of MKLN1 and ZMYND19 are independent of the CTLH complex activity. Is that what the authors conclude? Because the model in Fig 5j does not take that into account. Could the authors overexpress MKLN1 and ZMYND19 in WT cells and determine whether this synergizes with alpelisib to determine whether it compromises cell proliferation as in Fig 3i?

>>yes, our model is that CTLH targets MKLN1 and ZMYND19 for degradation, but that when CTLH activity is interrupted, MKLN1 and ZMYND19 accumulate, associate and block mTOR independently of CTLH. As shown above, MKLN1/ZMYND19 over-expression impairs S6K and 4E-BP1 phosphorylation. We show in Fig. 3 that over-expression of MKLN1/ZMYND19 is sufficient to block mTOR activity in mRNA transfected 293 cells. We also found as suggested that MKLN1/ZMYND19 over-expression synergizes with alpelisib, now presented in Fig. S6J-K:

4) Fig 4 localization of muskelin and ZMYND19 at lysosomes

a. Localization of MKLN1 to the lysosome was also previously reported. Heisler et al., Neuron 2011, PMID: 21482357 ; Heisler et al., Neuron 2018 PMID: 30174115. These papers should be cited. >>thanks, we now cite each of these papers. However, by our reading, it is a bit different from what we found. PMID 21482357 suggests MKLN1 is recruited to sorting endosome, MVB and then are turned over within lysosomes as shown below in their model:

PMID 30174115 suggests that Muskelin directs lysosomal trafficking of the prior protein PrP^C to facilitate its degradation within lysosomes. Nonetheless, we cite both papers.

b. Line 285: "Taken together, these data suggest that ZMYND19 recruits MKLN1 to lysosomal outer membrane regions." Fig 4f: the diagram indicates CTLH complex inhibits ZMYND19 and MKLN1 localization at the lysosomal membrane: it is unclear where does this conclusion come from? Could the authors clarify how they reached that conclusion?

>>We present data that ZMYND19 associates with the lysosome outer membrane region, likely through protein-protein interactions. Lyso-IP studies found that both ZMYND19 and MKLN1 co-purify with lysosomes. We now present proteinase K degradation data to indicate that ZMYND19 and MKLN1 are degraded by proteinase K addition to extracts prepared by gentle Dounce homogenization, whereas lysosome resident Cathepsin L is protected until Triton-X100 was also added (Triton disrupts lysosomal membranes), presented now in Fig. 4h.

We also found that CRISPR ZMYND19 depletion reduced the amount of MKLN1 that co-purified with lysosomes by the Lyso-IP approach, presented now in Fig. 4g.

We therefore thought it was logical that CTLH targets ZMYND19, potentially in a MKLN1-dependent manner as discussed above, at the lysosome outer membrane. Furthermore, the lysosomal outer membrane is logical place for MKLN1/ZMYND19 to inhibit mTORC1, because it is the location where mTORC1 activity is highly regulated.

c. They also make the claim that ZMYND19 recruits CTLH to lysosomes but there is no data to back up the claim up. Why not do the LysoIP in ZMYND19 knockdown cells and see if it changes?

>> We suspect that a subpopulation of CTLH is transiently recruited to lysosomes in order to target ZMYND19 for degradation. We present evidence that a subpopulation of ZMYND19 co-purifies with lysosomes and is likely associated with the lysosomal outer membrane. While we did observe overlap in immunofluorescence signal between a subpopulation of MAEA and lysosomes (**Fig. 4a**), we observed very mild CTLH copurification with lysosome pulldowns (please see **Fig. 4d** and **Supplementary Fig. 8c**).

We therefore now clarify this in the text our model that CTLH transiently associates with and targets ZMYND19 for degradation at the lysosomal outer membrane. It is not uncommon for ubiquitin ligases to transiently associate with their substrates. As discussed above, we now also present data that ZMYND19 KO impairs MKLN1 recruitment to lysosomes by the lyso-IP approach, presented in Fig. 4g.

Fig 5: ZMYND19 and MKLN1 domains important for mTOR inhibition

a. Fig 5 a and b could be put in the supplemental materials as this is data that is already known from previous publications (Mohamed et al, EMBO J 2021; Sherpa et al., 2022.)

>>We moved the data as suggested to **Supplementary Fig. S9a and b.**

b. Is the effect of MKLN1-ZMYND19 dependent on the MAEA knockout? Meaning, can transfections, or overexpression of the 2 proteins on their own affect mTORC1 activation, or is that

>>Please see the discussion above where this question was asked and answered above. No, they are not dependent on MAEA KO, as shown in **Fig. 3h** and **Supplementary Fig. 6i.**

c. The blots shown in Fig 5 e, g and i are not quantified and they should be quantified to accurately assess the effects of MKLN1 and ZMYND19. In Fig 5e, it is impossible to tell what is happening with the levels of P-S6 since total S6 levels are different from one lane to the other. It is understood that it is difficult to quantify since levels of expression of the mutants are varying, but it seems that MKLN1 mutant K1-6 and K6 are actually activating S6 phosphorylation rather than failing to block mTOR.

>>Thanks for this point. We now provide quantitation for the phosphoblots, now presented in **Fig. 5 c, f, g.** We agree that these help with interpretation of the experiments.

6) Fig 6: ZMYND19 associates with RAPTOR and RagA/C

The authors conclude that the ZMYND19+muskelin complex interacts with the inactive RagA/C complex. Conclusion is reached by coIPing FLAG out of cells cotransfected with FLAG-muskelin and FLAG-ZMYND19, along with MYC-Raptor and HA-Rag proteins.

a. line 351-2: the authors claim that “the ZMYND19 zinc finger supports association with MKLN1, Raptor and CTLH.” There is no indication in these data that the other CTLH complex subunits also are part of

that complex. The authors should test either MAEA or RanBP9, or WDR26 (or others), to prove that the complex is also implicated.

>>Thanks for this question. We now show in Fig. 5b that the ZMYND19 zinc finger contributes to the association between MAEA and MKLN1 in YCCEL1. In addition, Supplementary Fig. 8c shows that the ZMYND19 zinc finger contributes to the association between MAEA and MKLN1 in 293T. These build on data in Supplementary Fig. 8a that MAEA and MKLN1 associate and in Supplementary Fig. 8b that ZMYND19 and MAEA associate in YCCEL1.

Fig. 5b

Supplementary Fig. 9:

b. They should perform this CoIP in cells depleted of muskelin to determine whether muskelin is important for this interaction to occur

>> Thanks for this point. We performed the suggested experiment, which is now shown in Supplementary Fig. 10c. We found that Raptor co-immunoprecipitated with ZMYND19 even in cells with MKLN1 depletion.

Globally, the fundamental observation that loss of MAEA synergistically blocks cell proliferation with alpelisib is compelling and suggests yet unappreciated role for the CTLH complex in mTOR regulation. However the data does not lead to a comprehensive model of how mTOR is inhibited and data interpretation is made difficult by the lack of quantification of key data, and by the apparent confusion regarding the mechanism of action of the CTLH complex. The authors sometimes mention MKLN1 as a target and sometimes as a member of the CTLH complex. MKLN1 is both a complex member and a substrate as described in previous publications (Maitland et al., 2019, Lampert et al, 2018, Sherpa et al., 2021), so their model should take that into account. Also, their data suggest that ZMYND19 may actually be a target of the MKLN1-CTLH complex since its levels are upregulated in MKLN1 KO cells, and while this is acknowledged, it is not taken into consideration in their model. Therefore, models presented in 4f and 5j are inaccurate.

>>we appreciate these points but discussed them all above. We discuss that MKLN1 is both a complex member and a substrate and cite the Maitland, Lampert and Sherpa studies. We updated the models in 4f and 5j (now Fig. 4i and 5h).

The authors actually propose that ZMYND19 is a lysosomal protein that is targeted by the CTLH complex through a MKLN1 interaction in their discussion, but their experimental strategy does not reflect that. If this is correct, then they should (as suggested above) repeat some of their experiments in Fig. 6 in MKLN1 KO cells to determine the contribution of MKLN1 to the mTOR regulation by ZMYND19.

>> We do not propose that ZMYND19 is a lysosome resident protein, but rather much like mTOR itself, but rather that it associates with the cytoplasmic-facing lysosomal outer membrane leaflet. As discussed above, we now present additional data including confocal imaging, LysoIP and Proteinase K digestion to support this claim. We propose that ZMYND19 does this since it is the subcellular location where mTORC1 activity is highly regulated, and where it would logically home to perturb a late stage in mTORC1 activation.

Reviewer #3 (Remarks to the Author):

General Comments

Epstein-Barr virus-associated gastric carcinoma (EBVaGC) is primarily associated with increased activation of the PI3K signaling pathway, mostly due to activating mutations occurring in its catalytic subunit p110 α . Consequently, treatment with the PI3K p110 α inhibitor alpelisib significantly reduces the growth of these cancer cells.

In this manuscript, Wang et al. performed a genome-wide CRISPR/Cas9 screen to identify genes that synergistically inhibit the proliferation of EBVaGC cells when combined with alpelisib. They identified several subunits of the C-terminal to LisH (CTLH) E3 ligase, including the catalytic MAEA subunit. The authors propose that depletion of CTLH subunits, particularly MAEA, results in increased levels of its substrates ZMYND19 and MKLN1, the latter also being a component of the CTLH complex. These two proteins, in turn, are proposed to inhibit mTORC1 signaling somehow by associating with Raptor and Rag GTPases at the lysosome.

While this study presents potentially interesting findings, it lacks an explanation of the mechanism underlying mTORC1 downregulation and the physiological relevance of this observation. The data shown in the manuscript suggest that increased levels of ZMYND19 and MKLN1 do not affect mTOR lysosomal recruitment or its activation by Rheb. In the discussion section, the authors hypothesize that these two proteins could impair the ability of Raptor to associate with TOS-containing mTORC1 substrates, such as S6K and 4E-BP1, or preclude the association between mTORC1 and Rheb. Both hypotheses need to be tested to dissect the mechanism underlying ZMYND19-MKLN1-dependent inhibition of mTORC1 signaling. Since transcription factors EB and E3 are mTORC1 substrates that do not contain a TOS motif (PMID: 32612235), analyzing their phosphorylation state as well as their nuclear/cytosolic localization can be useful to test the first hypothesis. However, both hypotheses should be exhaustively tested through biochemical and imaging analyses.

Moreover, the relevance of these findings needs to be demonstrated in the context of physiological regulation of mTORC1 activity, particularly in response to nutrient availability. Notably, a very recent paper (PMID: 38788716) demonstrates that mTORC1 inhibition promotes the degradation of the HMGCS1 enzyme via the CTLH complex, thus impacting cell proliferation by affecting the mevalonate pathway. Importantly, they show that deletion of CTLH E3 activity, through MAEA depletion, did not alter mTOR signaling or autophagy in HEK293T cells, which contrasts with the main findings presented in this manuscript. Therefore, the evidence that CTLH genetic inactivation is responsible for the downregulation of mTORC1 signaling needs further corroboration.

Throughout the manuscript, mTORC1 signaling is monitored by analyzing the phosphorylation levels of P-S6K only (or P-S6, which is not clear—see specific comments). The authors should monitor and quantify

the phosphorylation levels of both P-S6K (at Thr389) and P-4E-BP1 (at Ser-65) relative to the corresponding total proteins, in both fed and starved conditions, with torin-1 treatment added as an internal control. However, the relevance of mTORC1 inhibition in the context of EBVaGC is not demonstrated, as treatment with rapamycin seems to have a subtle effect on the growth of these cells (Fig. S3F), suggesting that other functions downstream of the CTLH complex could impair the growth of these cancer cells rather than the regulation of the mTORC1 pathway. Therefore, as it stands, the study does not support the claims and conclusions made by the authors.

>> Thanks for the excellent summary and points raised. We respond to the specific points below. However we wish to address here that we did find that MAEA knockout decreased S6K and 4E-BP1 phosphorylation in HEK-293T, as shown now in Supplementary Fig. 4e.

As discussed above in response to Reviewer 2's comments and presented in Fig. 3h, we also found that overexpression of MKLN1 and ZMYND19 by electroporation of in vitro transcribed mRNA into MKLN1/ZMYND19 double KO HEK-293T inhibited S6K and 4E-BP1 phosphorylation in 293T.

Our data therefore suggests that MAEA KO inhibits mTOR in 293T at an early timepoint post-CRISPR editing (day 4 after Cas9 RNP electroporation) and that a mechanism by which MKLN1/ZMYND19 overexpression blocks mTOR signaling is operative in HEK-293T. By contrast, PMID 38788716 performed experiments on 293T single cell clones that were sorted into 96-well plates using a limiting dilution method and then expanded. Such expansion would have taken weeks. We therefore speculate that

because PMID 38788716 performed experiments in 293T at a much later timepoint following CTLH editing, that there was a selection in their system against mTOR blockade downstream of MAEA KO. As a result, PMID 38788716 may not have observed effects of MAEA KO on mTOR signaling at the late timepoint in which the experiment was performed. In agreement with 38788716, our proteomic analysis did find upregulation of HMGCS1 in MAEA KO YCCEL1. We added discussion of this point to our discussion section.

We now also present data to show that despite MAEA KO effects on mTOR, MAEA KO + alpelisib did not significantly decreased 293T cell viability beyond treatment with alpelisib alone, as shown in new **Supplementary Fig. 2f**.

We are not certain why synthetic lethality is not observed in MAEA KO and alpelisib treated 293T, but suspect that this related to an aspect that has been lost with 293T. 293T have lost many pathways, including loss of the necroptosis and pyroptosis cell death pathways for example. It is also possible that the adenovirus E1A oncogene, expressed in 293T, could underlie this observation. We likewise suspect that PMID 38788716 did not find effects of MAEA KO on autophagy in 293T for similar reasons. We now provide additional evidence as described below that MAEA KO induced autophagy in gastric carcinoma cells, including through analysis of LC3 and by electron microscopy demonstration of increased numbers of multivesicular bodies suggestive of autophagy.

Specific Comments

1. Fig. 2C: Provide more detailed information in the figure legend, including sample comparisons and statistical methods applied.

>>We increased the figure legend detail as suggested and include statistical methods detail.

2. Fig. 3B: Clarify whether the authors are monitoring P-S6K phosphorylation or P-S6 phosphorylation. The panel indicates P-S6 (T389), but S6 phosphorylation is typically monitored at Ser235/236 or Ser240/244, whereas S6K phosphorylation is monitored at Thr389. If the Western blot refers to P-S6K and pan-S6K, this needs to be corrected. The same error recurs in all figures monitoring mTORC1 signaling (Figs. 3C, 3J, 7E, 7G, S3E, S3N, S6C, S6D, S7A). It should be correctly indicated whether P-S6K phosphorylation or P-S6 phosphorylation is being monitored.

>>thank you for this point. Yes, we monitored P-S6K phosphorylation. We corrected the error as suggested.

Additionally, S6 phosphorylation is not a good readout of mTORC1 activity since it is an indirect substrate of mTOR (phosphorylated by S6K). For all these panels, the authors should provide analysis and quantification of phosphorylation levels of both S6K and 4E-BP1 relative to corresponding total proteins. Both fed and starved conditions should be analyzed, with torin-1 treatment as an internal control for mTORC1 signaling inhibition.

>>We now provide quantitation throughout. As mentioned above, we monitored S6K and have corrected that. We now include fed/starved conditions and use Torin-1 treatment as an internal control for mTORC1 signaling inhibition in both YCCEL1 and SNU-719.

3. Fig. S3F: The four conditions shown (sgControl and sgMAEA in DMSO or rapamycin) should be compared together and analyzed through ANOVA (Student's t-test cannot be used when comparing more than two samples or conditions). However, it seems that rapamycin treatment does not significantly impact cell survival, suggesting that mTORC1 signaling does not significantly support the growth of these gastric tumor cells. These results do not align with the manuscript's main claim.

>>Thanks for this point. We now use Anova analysis and state that in the legend of this figure, which is now S4h. As shown below, we did see a significant effect of MAEA KO + rapamycin on cell survival. The combination of MAEA KO + rapamycin significantly reduced live cell numbers below those of either MAEA KO or rapamycin treatment alone.

We found similar results with YCCEL1 treated with rapamycin and alpelisib, as shown to the right below, consistent with the manuscript's main claim.

4. Fig. 3D: Quantification of lipid droplets should be provided to support the claim that their number significantly increases upon MAEA depletion. Since autophagy activation would promote lipid droplet clearance (PMID: 19339967) rather than accumulation, this observation contradicts the authors' claim that MAEA depletion induces mTORC1 downregulation and hence autophagy activation. To assess autophagy activation, the authors should check levels of lipidated LC3 (by WB) and LC3 puncta (by IF) upon MAEA depletion.

>>As suggested, we now provide in new **Supplemental Fig.5**. We quantitated lipid droplets per cytoplasm area from groups of 25 cells in collaboration with our electron microscopy core and using ImageJ software. We observed significantly more LD with MAEA depletion, as shown below. We also observed more double membrane, autophagosome-like structures which we now also present in new **Supplementary Fig. 5**.

While we agree that PMID 19339967 showed that autophagy reduced lipid droplets in liver cells, this phenomenon appears to be cell-type specific. PMID 25512609 (Madeira et al Mol Cell Biol 2015 35(4):737-46) found that mTORC1 inhibition including by rapamycin induces lipid droplet replenishment in yeast. Furthermore, PMID 25512609 (Nguyen et al, Dev Cell 2017 10;42(1):9-21.e5) found that mTORC1-regulated autophagy including by Torin 1 is necessary and sufficient for starvation-induced lipid droplet biogenesis. Thus, while 19339967 found that an autophagic lipophagy pathway degrades lipid droplets and that inhibition of autophagy results in lipid droplet accumulation in hepatocytes, the opposite was observed in murine embryonic fibroblasts, HeLa, Huh7, and U2OS in PMID 28697336. Therefore, our results suggest that YCCEL1 also exhibit behavior similar to the cells studied in PMID 28697336. Alternatively, it is possible that given emerging CTLH roles in lipid metabolism (i.e. mevalonate metabolism), that CTLH may also have roles in regulation of lipid droplet metabolism.

As suggested, we also measured LC3B-II (lipidated LC3) levels by immunoblot, and found it to be increased by amino acid starvation, by MAEA KO, and nearly additively by amino acid starvation + MAEA KO. Torin 1 was included as a + control. This is now presented in Supplementary Fig. 5a.

We also now provide immunofluorescence data in Supplementary Fig. 5b-c that cytoplasmic LC3 intensity is increased to higher levels by MAEA KO + amino acid starvation than by amino acid starvation alone.

5. Fig. 4D, 4E: To prove that the proteins of interest are indeed specifically associated with lysosomes, the authors should also monitor markers for other cellular compartments in the same WB (ER, Golgi, mitochondria, similar to what is shown in PMID: 29074583) to demonstrate the purity of their lysosomal fractions.

>>Thanks for this suggestion. We now add additional markers as suggested. We now include blots for Golgi marker Golgin97, ER marker calreticulin and mitochondrial marker VDAC.

6. Fig. 4F: To prove that ZMYND19 and MKLN1 associate with the outer (cytoplasm-facing) leaflet of the lysosome, the authors should treat affinity-captured lysosomes with increasing concentrations of

proteinase K and show that these two proteins are sensitive to proteinase K digestion, contrary to luminal proteins (e.g., LC3, hydrolases) that will be protected.

>>Thanks for this suggestion. We developed and performed a proteinase K assay, in which we used Dounce homogenization to gently disrupt plasma membranes (ie PMID 31262961). We used this approach instead of LysoIP, because despite multiple attempts, we found that Proteinase K degraded lysosome resident proteins such as Cathepsin K following the Lyso-IP approach. This new data is presented in Fig. 4h and shown below. We overexpressed HA-ZMYND19 in YCCEL1 (given low endogenous ZMYND19 levels). We then gently lysed cells by dounce homogenization. Supernatants were treated with proteinase K, which degraded endogenous MKLN1, HA-ZMYND19 and as a control GAPDH, but to a lesser extent lysosome resident Cathepsin L. As a further control, we also treated Dounce homogenized YCCEL1 with Triton X-100 to permeabilize lysosomes, together with Proteinase K. Combined Triton and proteinase K degraded Cathepsin L, as expected.

Additionally, the authors should analyze lysosomal localization of endogenous MKLN1 and ZMYND19 by imaging analysis in response to MAEA depletion and mTORC1 activity (nutrient/starvation conditions).

>>We now provide in Fig. 4f and Supplementary Fig. 8b immunofluorescence microscopy analysis of ZMYND19 and MKLN1 subcellular localization. We cross-compared their subcellular localization with lysosomal LAMP2 and observed a high degree of co-localization with LAMP2.

We also observed co-localization between a subpopulation of MKLN1 and LAMP2

Total levels of these proteins should also be checked by WB in response to nutrient availability.
 >> we did not find amino acid starvation to alter total cellular levels of MKLN1 or ZMYND19. We do not think that CTLH activity is regulated by amino acid starvation.

7. Fig. 6C and Fig. S6B: The authors report that both figures indicate the zinc finger of ZMYND19 is important for binding with Raptor, but Fig. S6B demonstrates the opposite! The two figures are not consistent: the zinc finger of ZMYND19 seems not to be important for binding with Raptor in 293T cells but is important in YCCEL1 cells. Quantification should be reported for these data, and the authors should reconcile this discrepancy.

>>We do not agree that there was a discrepancy. In both cases, we found that the zinc-finger deletion mutant co-immunoprecipitated a lower amount of Raptor. The magnitude of the result was stronger with YCCEL1 in Fig. 6C, but the result in original Fig. S6B (now S10B) also shows the same effect. We now provide quantitation as suggested, in which we normalized the amount of IP signal by input.

Figure 6C:

Figure S10B

8. Fig. 6E: The data are insufficient to support the claim that ZMYND19/MKLN1 preferentially associate with inactive Rags. The slight difference observed in the WB seems to align with levels of expression of RagC constructs. This result needs to be validated through quantification of IP and imaging analysis. If the authors propose that this association occurs during starvation, this needs to be demonstrated in fed vs. starved cells and following endogenous Rags.

>> We now provide quantification.

We did not wish to propose that this happens during starvation, as we observed in cells +AA and have therefore modified the text.

9. Fig. 7G: Authors should test also starved versus fed condition and test whether Rheb overexpression shows the same phenotype.

>>As suggested, we now also present in Supplementary Fig. 11d that MAEA KO impairs mTORC1 activation in cells with Rheb overexpression. We included fed vs starved conditions and quantitation.

Comments on the Text

1. Row 63: RagA and RagB are functionally interchangeable, as are RagC and RagD, hence four heterodimer configurations are possible: RagA/RagC, RagA/RagD, RagB/RagC, and RagB/RagD. Their activation state depends only on amino acid loading. Please correct the sentence.

>>Thanks for pointing this out, we had meant to convey this concept in the original draft. We now write this more clearly.

2. Row 149: SESN2 is not a component of the GATOR1 complex but a regulator of its activity (it binds and inhibits GATOR2, which inhibits GATOR1). Please correct this.

>>We corrected this error, thanks for pointing it out.

3. Row 152: The complex name is Ragulator, not regulator. Please correct this.

>>Thanks for spotting this typo.

Reviewers' comments:

Reviewer #1 (Remarks to the Author):

The authors have now performed a large number of additional experiments in response to each of the comments, suggestions, and concerns expressed by the three reviewers. They have also now thoroughly quantified their data and made numerous improvements to the text, methods, and reference sections in response to reviewers' comments. This reviewer believes the authors have now adequately addressed essentially all of the concerns of the reviewers.

>>We thank this Reviewer for recognizing the large amount of work that was done to address each of their specific points and feel that the manuscript was strengthened as a result.

Reviewer #2 (Remarks to the Author):

The authors have satisfactorily addressed my concerns and I support publication of the manuscript.

>>We thank this Reviewer for recognizing the large amount of work that was done to address each of their specific points and feel that the manuscript was strengthened as a result.

Reviewer #3 (Remarks to the Author):

In this revised version of the manuscript entitled "The CTLH1 Ubiquitin Ligase Substrates ZMYND19 and MKLN1 Negatively Regulate mTORC1 at the Lysosomal Membrane," Wang et al. present additional experiments supporting the claim that ZMYND19 and MKLN1 act in concert to inhibit mTORC1 signaling, thereby activating the autophagy pathway. However, the manuscript does not provide further insight into the mechanism underlying the ZMYND19-MKLN1-dependent inhibition of mTORC1 signaling. The data suggest that these CTLH substrates do not regulate RagGTPases activity, as MAEA depletion still leads to mTORC1 inhibition in cells overexpressing a lysosomal-anchored Raptor (Raptor-Rheb) construct or in cells KO for GATOR1. Moreover, the TSC axis does not appear to be involved, as neither TSC depletion nor Rheb overexpression affects the observed phenotype.

In the discussion section, the authors propose the interesting hypotheses that ZMYND19/MKLN1 may prevent Raptor binding to the TOS motif found in 4EBP1, S6K1, and eventually additional mTORC1 regulators, or alternatively, ZMYND19/MKLN1 may prevent the association between Ragulator-bound mTORC1 and Rheb. These hypotheses require experimental validation to fully elucidate the mechanism through which CTLH substrates regulate mTORC1 signaling.

>>We thank the reviewer for these important points and for pushing us to go further. However, we do wish to point out that the reviewer did not acknowledge that we addressed each of their nine Specific Comments in detail (we pasted the original point by point response at the bottom of this document). This was a major undertaking and involved for example the development of a protease protection assay to further validate the cytoplasmic subcellular localization of ZMYND19/MKLN1. We were surprised that this was not acknowledged by the reviewer, who instead took a negative tone. We were a bit thrown off by the fact that the reviewer listed points of the greatest importance in their assessment in their General Comments, but not in any of their nine Specific Comments.

We thank the reviewer for pushing us to go further to obtain a more detailed mechanism. In response to these reviewer concerns, we now present data (all changes are marked in the margin by simple tracked changes) that MAEA KO or ZMYND19/MKLN1 overexpression each strongly impede association between Raptor and Rheb. This is demonstrated by reciprocal co-immunoprecipitation analysis. We also present new evidence that ZMYND19/MKLN1 impede Raptor association with the TOS motif containing substrates S6K and 4EBP1. Taken together with the extensive evidence presented in our manuscript that ZMYND19/MKLN1 does not prevent mTORC1 recruitment by Ragulator-Rag or require the TSC axis, we now present evidence for how they block mTORC1 activity in new Figures 7D-E and Supplemental Figures 12 B-C and shown below. We now demonstrate that MAEA KO or ZMYND19/MKLN1 over-expression strongly impair Raptor and Rheb co-immunoprecipitation. This is shown in both directions (IP Raptor blot Rheb and IP Rheb, blot Raptor). We also present evidence that MAEA KO or ZMYND19/MKLN1 over-expression also impairs co-immunoprecipitation between Raptor and the mTORC1 TOS motif containing substrates S6K and 4EBP1. Based on available cryo-EM data, it is plausible that ZMYND19/MKLN1 therefore binds to the mTORC1 complex in a manner that occludes Raptor interaction with the five residue TOS motif and that also occludes mTOR interaction with Rheb, which could represent a contiguous surface (please see the published models below for reference) or which could be achieved through allostery.

From PMID 29236692

From PMID 31601708

Given that we began with a human genome-wide CRISPR-Cas9 screen, identified CTLH as a key hit, identified that CTLH is a key regulator of mTORC1, identified the key CTLH substrates involved in mTOR inhibition, and now also identified how they block mTORC1, we feel that the story is now complete enough for publication. We have spent over four years and extensive resources to reach this point and would be hard pressed to fit any more data into this manuscript.

Figure 7

I also suggested to test the activation state of TFEB which behave as a peculiar mTORC1 substrate dependent on FLCN axis but not data were provided on this.

We now present evidence in Supplemental Figures 12 D-E that CTLH disruption by MAEA KO impairs phosphorylation of both TFE3 and TFEB to similar levels as positive control Torin 1 treatment, and that both MAEA KO and Torin 1 each drive TFEB nuclear translocation. Taken together, these data further demonstrate that CTLH inhibition by MAEA KO broadly impairs mTORC1 activity, beyond inhibition of TOS containing substrates.

Furthermore, as noted in my previous review, the manuscript lacks information regarding the physiological relevance of this mechanism. The authors should address whether they intend to propose that CTLH activity is reduced during nutrient (particularly amino acid) starvation, and whether increased levels of ZMYND19/MKLN1 serve to inhibit mTORC1 signaling under such conditions. As it stands, this manuscript is missing key experimental evidence and mechanistic details, and therefore lacks the substantiation needed for acceptance for publication.

>>Thanks for this interesting point. We disagree with the reviewer that we do not provide relevance for the mechanism, because from our perspective, our study does provide a key context where CTLH activity is physiologically important: we show that it supports mTORC1 activation in EBV+ gastric carcinoma cells with hyperactive PI3K, the subject of our CRISPR screen. While we do agree with the reviewer that an important next step of this research program will be to define the nutritional input that controls physiological CTLH activity across cellular contexts, we feel strongly that this next logical stage of this program is beyond the scope of this manuscript. We have looked and have not found evidence that amino acid starvation regulates CTLH activity, since it did not increase MKLN1 or ZMYND19 abundances, in contrast to MAEA KO.

Therefore, it will be an open-ended screen to identify the nutritional input that is sensed to control CTLH activity, the mechanism by which it is sensed and the mechanism by which it regulates CTLH. Given that CTLH controls 3-hydroxy-3-methylglutaryl (HMG)-coenzyme A (CoA) synthase 1 (HMGCS1), the initial enzyme in the mevalonate pathway, we speculate that a lipid metabolite, possibly a substrate of the mevalonate or cholesterol biosynthesis pathway, may support CTLH activity. However, definition of this aspect of CTLH regulation will be a major undertaking. There would not be space to include the screen for the metabolite and all of the necessary validation and mechanistic analyses within this present manuscript, where we already have filled all of the main and supplementary figure space that we are

allowed to use. We began with a human genome-wide CRISPR-Cas9 screen, identified CTLH as a key synthetic target with PI3K inhibition, identified MKLN1 and ZMYND19 as the relevant CTLH substrates, and identified their mechanism of action.

--

Please find below our Point by Point response to Reviewer #3 from the first round of review

General Comments

Epstein-Barr virus-associated gastric carcinoma (EBVaGC) is primarily associated with increased activation of the PI3K signaling pathway, mostly due to activating mutations occurring in its catalytic subunit p110 α . Consequently, treatment with the PI3K p110 α inhibitor alpelisib significantly reduces the growth of these cancer cells.

In this manuscript, Wang et al. performed a genome-wide CRISPR/Cas9 screen to identify genes that synergistically inhibit the proliferation of EBVaGC cells when combined with alpelisib. They identified several subunits of the C-terminal to LisH (CTLH) E3 ligase, including the catalytic MAEA subunit. The authors propose that depletion of CTLH subunits, particularly MAEA, results in increased levels of its substrates ZMYND19 and MKLN1, the latter also being a component of the CTLH complex. These two proteins, in turn, are proposed to inhibit mTORC1 signaling somehow by associating with Raptor and Rag GTPases at the lysosome.

While this study presents potentially interesting findings, it lacks an explanation of the mechanism underlying mTORC1 downregulation and the physiological relevance of this observation. The data shown in the manuscript suggest that increased levels of ZMYND19 and MKLN1 do not affect mTOR lysosomal recruitment or its activation by Rheb. In the discussion section, the authors hypothesize that these two proteins could impair the ability of Raptor to associate with TOS-containing mTORC1 substrates, such as S6K and 4E-BP1, or preclude the association between mTORC1 and Rheb. Both hypotheses need to be tested to dissect the mechanism underlying ZMYND19-MKLN1-dependent inhibition of mTORC1 signaling. Since transcription factors EB and E3 are mTORC1 substrates that do not contain a TOS motif (PMID: 32612235), analyzing their phosphorylation state as well as their nuclear/cytosolic localization can be useful to test the first hypothesis. However, both hypotheses should be exhaustively tested through biochemical and imaging analyses.

Moreover, the relevance of these findings needs to be demonstrated in the context of physiological regulation of mTORC1 activity, particularly in response to nutrient availability. Notably, a very recent paper (PMID: 38788716) demonstrates that mTORC1 inhibition promotes the degradation of the HMGCS1 enzyme via the CTLH complex, thus impacting

cell proliferation by affecting the mevalonate pathway. Importantly, they show that deletion of CTLH E3 activity, through MAEA depletion, did not alter mTOR signaling or autophagy in HEK293T cells, which contrasts with the main findings presented in this manuscript. Therefore, the evidence that CTLH genetic inactivation is responsible for the downregulation of mTORC1 signaling needs further corroboration.

Throughout the manuscript, mTORC1 signaling is monitored by analyzing the phosphorylation levels of P-S6K only (or P-S6, which is not clear—see specific comments). The authors should monitor and quantify the phosphorylation levels of both P-S6K (at Thr389) and P-4E-BP1 (at Ser-65) relative to the corresponding total proteins, in both fed and starved conditions, with torin-1 treatment added as an internal control. However, the relevance of mTORC1 inhibition in the context of EBVaGC is not demonstrated, as treatment with rapamycin seems to have a subtle effect on the growth of these cells (Fig. S3F), suggesting that other functions downstream of the CTLH complex could impair the growth of these cancer cells rather than the regulation of the mTORC1 pathway. Therefore, as it stands, the study does not support the claims and conclusions made by the authors.

>> Thanks for the excellent summary and points raised. We respond to the specific points below. However we wish to address here that we did find that MAEA knockout decreased S6K and 4E-BP1 phosphorylation in HEK-293T, as shown now in Supplementary Fig. 4e.

As discussed above in response to Reviewer 2's comments and presented in Fig. 3h, we also found that overexpression of MKLN1 and ZMYND19 by electroporation of in vitro transcribed mRNA into MKLN1/ZMYND19 double KO HEK-293T inhibited S6K and 4E-BP1 phosphorylation in 293T.

Our data therefore suggests that MAEA KO inhibits mTOR in 293T at an early timepoint post-CRISPR editing (day 4 after Cas9 RNP electroporation) and that a mechanism by which MKLN1/ZMYND19 overexpression blocks mTOR signaling is operative in HEK-293T. By contrast, PMID 38788716 performed experiments on 293T single cell clones that were sorted into 96-well plates using a limiting dilution method and then expanded. Such expansion would have taken weeks. We therefore speculate that because PMID 38788716 performed experiments in 293T at a much later timepoint following CTLH editing, that there was a selection in their system against mTOR blockade downstream of MAEA KO. As a result, PMID 38788716 may not have observed effects of MAEA KO on mTOR signaling at the late timepoint in which the experiment was performed. In agreement with 38788716, our proteomic analysis did find upregulation of HMGCS1 in MAEA KO YCCEL1. We added discussion of this point to our discussion section.

We now also present data to show that despite MAEA KO effects on mTOR, MAEA KO + alpelisib did not significantly decreased 293T cell viability beyond treatment with alpelisib alone, as shown in new **Supplementary Fig. 2f**.

We are not certain why synthetic lethality is not observed in MAEA KO and alpelisib treated 293T, but suspect that this related to an aspect that has been lost with 293T. 293T have lost many pathways, including loss of the necroptosis and pyroptosis cell death pathways for

example. It is also possible that the adenovirus E1A oncogene, expressed in 293T, could underlie this observation. We likewise suspect that PMID 38788716 did not find effects of MAEA KO on autophagy in 293T for similar reasons. We now provide additional evidence as described below that MAEA KO induced autophagy in gastric carcinoma cells, including through analysis of LC3 and by electron microscopy demonstration of increased numbers of multivesicular bodies suggestive of autophagy.

Specific Comments

1. Fig. 2C: Provide more detailed information in the figure legend, including sample comparisons and statistical methods applied.

>>We increased the figure legend detail as suggested and include statistical methods detail.

2. Fig. 3B: Clarify whether the authors are monitoring P-S6K phosphorylation or P-S6 phosphorylation. The panel indicates P-S6 (T389), but S6 phosphorylation is typically monitored at Ser235/236 or Ser240/244, whereas S6K phosphorylation is monitored at Thr389. If the Western blot refers to P-S6K and pan-S6K, this needs to be corrected. The same error recurs in all figures monitoring mTORC1 signaling (Figs. 3C, 3J, 7E, 7G, S3E, S3N, S6C, S6D, S7A). It should be correctly indicated whether P-S6K phosphorylation or P-S6 phosphorylation is being monitored.

>>thank you for this point. Yes, we monitored P-S6K phosphorylation. We corrected the error as suggested.

Additionally, S6 phosphorylation is not a good readout of mTORC1 activity since it is an indirect substrate of mTOR (phosphorylated by S6K). For all these panels, the authors should provide analysis and quantification of phosphorylation levels of both S6K and 4E-BP1 relative to corresponding total proteins. Both fed and starved conditions should be analyzed, with torin-1 treatment as an internal control for mTORC1 signaling inhibition.

>>We now provide quantitation throughout. As mentioned above, we monitored S6K and have corrected that. We now include fed/starved conditions and use Torin-1 treatment as an internal control for mTORC1 signaling inhibition in both YCCEL1 and SNU-719.

Fig.3b

3. Fig. S3F: The four conditions shown (sgControl and sgMAEA in DMSO or rapamycin) should be compared together and analyzed through ANOVA (Student's t-test cannot be used when comparing more than two samples or conditions). However, it seems that rapamycin treatment does not significantly impact cell survival, suggesting that mTORC1 signaling does not significantly support the growth of these gastric tumor cells. These results do not align with the manuscript's main claim.

>>Thanks for this point. We now use Anova analysis and state that in the legend of this figure, which is now S4h. As shown below, we did see a significant effect of MAEA KO + rapamycin on cell survival. The combination of MAEA KO + rapamycin significantly reduced live cell numbers below those of either MAEA KO or rapamycin treatment alone.

We found similar results with YCCCL1 treated with rapamycin and alpelisib, as shown to the right below, consistent with the manuscript's main claim.

4. Fig. 3D: Quantification of lipid droplets should be provided to support the claim that their number significantly increases upon MAEA depletion. Since autophagy activation would promote lipid droplet clearance (PMID: 19339967) rather than accumulation, this observation contradicts the authors' claim that MAEA depletion induces mTORC1 downregulation and hence autophagy activation. To assess autophagy activation, the authors should check levels of lipidated LC3 (by WB) and LC3 puncta (by IF) upon MAEA depletion.

>>As suggested, we now provide in new **Supplemental Fig.5**. We quantitated lipid droplets per cytoplasm area from groups of 25 cells in collaboration with our electron microscopy core and using ImageJ software. We observed significantly more LD with MAEA depletion, as shown below. We also observed more double membrane, autophagosome-like structures which we now also present in new **Supplementary Fig. 5**.

While we agree that PMID 19339967 showed that autophagy reduced lipid droplets in liver cells, this phenomenon appears to be cell-type specific. PMID 25512609 (Madeira et al Mol Cell Biol 2015 35(4):737-46) found that mTORC1 inhibition including by rapamycin induces lipid droplet replenishment in yeast. Furthermore, PMID 25512609 (Nguyen et al, Dev Cell 2017 10;42(1):9-21.e5) found that mTORC1-regulated autophagy including by Torin 1 is necessary and sufficient for starvation-induced lipid droplet biogenesis. Thus, while 19339967 found that an autophagic lipophagy pathway degrades lipid droplets and that inhibition of autophagy results in lipid droplet accumulation in hepatocytes, the opposite was observed in murine embryonic fibroblasts, HeLa, Huh7, and U2OS in PMID 28697336. Therefore, our results suggest that YCCEL1 also exhibit behavior similar to the cells studied in PMID 28697336. Alternatively, it is possible that given emerging CTLH roles in lipid metabolism (i.e. mevalonate metabolism), that CTLH may also have roles in regulation of lipid droplet metabolism.

As suggested, we also measured LC3B-II (lipidated LC3) levels by immunoblot, and found it to be increased by amino acid starvation, by MAEA KO, and nearly additively by amino acid starvation + MAEA KO. Torin 1 was included as a + control. This is now presented in Supplementary Fig. 5a.

We also now provide immunofluorescence data in Supplementary Fig. 5b-c that cytoplasmic LC3 intensity is increased to higher levels by MAEA KO + amino acid starvation than by amino acid starvation alone.

5. Fig. 4D, 4E: To prove that the proteins of interest are indeed specifically associated with lysosomes, the authors should also monitor markers for other cellular compartments in the same WB (ER, Golgi, mitochondria, similar to what is shown in PMID: 29074583) to demonstrate the purity of their lysosomal fractions.

>>Thanks for this suggestion. We now add additional markers as suggested. We now include blots for Golgi marker Golgin97, ER marker calreticulin and mitochondrial marker VDAC.

6. Fig. 4F: To prove that ZMYND19 and MKLN1 associate with the outer (cytoplasm-facing) leaflet of the lysosome, the authors should treat affinity-captured lysosomes with increasing concentrations of proteinase K and show that these two proteins are sensitive to proteinase K digestion, contrary to luminal proteins (e.g., LC3, hydrolases) that will be protected.

>>Thanks for this suggestion. We developed and performed a proteinase K assay, in which we used Dounce homogenization to gently disrupt plasma membranes (ie PMID 31262961). We used this approach instead of LysoIP, because despite multiple attempts, we found that Proteinase K degraded lysosome resident proteins such as Cathepsin K following the Lyso-IP approach. This new data is presented in Fig. 4h and shown below. We overexpressed HA-ZMYND19 in YCCEL1 (given low endogenous ZMYND19 levels). We then gently lysed cells by dounce homogenization. Supernatants were treated with proteinase K, which degraded endogenous MKLN1, HA-ZMYND19 and as a control GAPDH, but to a lesser extent lysosome resident Cathepsin L. As a further control, we also treated Dounce homogenized YCCEL1 with Triton X-100 to permeabilize lysosomes, together with Proteinase K. Combined Triton and proteinase K degraded Cathepsin L, as expected.

Additionally, the authors should analyze lysosomal localization of endogenous MKLN1 and ZMYND19 by imaging analysis in response to MAEA depletion and mTORC1 activity (nutrient/starvation conditions).

>>We now provide in Fig. 4f and Supplementary Fig. 8b immunofluorescence microscopy analysis of ZMYND19 and MKLN1 subcellular localization. We cross-compared their subcellular localization with lysosomal LAMP2 and observed a high degree of co-localization with LAMP2.

We also observed co-localization between a subpopulation of MKLN1 and LAMP2

Total levels of these proteins should also be checked by WB in response to nutrient availability.

>> we did not find amino acid starvation to alter total cellular levels of MKLN1 or ZMYND19. We do not think that CTLH activity is regulated by amino acid starvation.

7. Fig. 6C and Fig. S6B: The authors report that both figures indicate the zinc finger of ZMYND19 is important for binding with Raptor, but Fig. S6B demonstrates the opposite! The two figures are not consistent: the zinc finger of ZMYND19 seems not to be important for binding with Raptor in 293T cells but is important in YCCEL1 cells. Quantification should be reported for these data, and the authors should reconcile this discrepancy.

>>We do not agree that there was a discrepancy. In both cases, we found that the zinc-finger deletion mutant co-immunoprecipitated a lower amount of Raptor. The magnitude of the result was stronger with YCCEL1 in Fig. 6C, but the result in original Fig. S6B (now S10B) also shows the same effect. We now provide quantitation as suggested, in which we normalized the amount of IP signal by input.

Figure 6C:

Figure S10B

8. Fig. 6E: The data are insufficient to support the claim that ZMYND19/MKLN1 preferentially associate with inactive Rags. The slight difference observed in the WB seems to align with levels of expression of RagC constructs. This result needs to be validated through quantification of IP and imaging analysis. If the authors propose that this association occurs during starvation, this needs to be demonstrated in fed vs. starved cells

and following endogenous Rags.
 >> We now provide quantification.

We did not wish to propose that this happens during starvation, as we observed in cells +AA and have therefore modified the text.

9. Fig. 7G: Authors should test also starved versus fed condition and test whether Rheb overexpression shows the same phenotype.

>>As suggested, we now also present in Supplementary Fig. 11d that MAEA KO impairs mTORC1 activation in cells with Rheb overexpression. We included fed vs starved conditions and quantitation.

Comments on the Text

1. Row 63: RagA and RagB are functionally interchangeable, as are RagC and RagD, hence four heterodimer configurations are possible: RagA/RagC, RagA/RagD, RagB/RagC, and RagB/RagD. Their activation state depends only on amino acid loading. Please correct the sentence.

>>Thanks for pointing this out, we had meant to convey this concept in the original draft. We now write this more clearly.

2. Row 149: SESN2 is not a component of the GATOR1 complex but a regulator of its activity (it binds and inhibits GATOR2, which inhibits GATOR1). Please correct this.

>>We corrected this error, thanks for pointing it out.

3. Row 152: The complex name is Ragulator, not regulator. Please correct this.

>>Thanks for spotting this typo.